# High-density electrode recordings reveal strong and specific connections between retinal ganglion cells and midbrain neurons

Jérémie Sibille [1,2,3,4], Carolin Gehr [1,2,3,4], Jonathan I. Benichov [5,6], Hymavathy Balasubramanian[1,2,3,4], Kai Lun Teh [1,2,3,4], Tatiana Lupashina[1,2,3,4], Daniela Vallentin [5,6] & Jens Kremkow [1,2,3,4] ✉

The superior colliculus is a midbrain structure that plays important roles in visually guided behaviors in mammals. Neurons in the superior colliculus receive inputs from retinal ganglion cells but how these inputs are integrated in vivo is unknown. Here, we discovered that high-density electrodes simultaneously capture the activity of retinal axons and their postsynaptic target neurons in the superior colliculus, in vivo. We show that retinal ganglion cell axons in the mouse provide a single cell precise representation of the retina as input to superior colliculus. This isomorphic mapping builds the scaffold for precise retinotopic wiring and functionally specific connection strength. Our methods are broadly applicable, which we demonstrate by recording retinal inputs in the optic tectum in zebra finches. We find common wiring rules in mice and zebra finches that provide a precise representation of the visual world encoded in retinal ganglion cells connections to neurons in retinorecipient areas.

Retinal ganglion cells (RGCs) encode the visual world in over 30 parallel functional pathways[1] and send this information via axons along the optic nerve to multiple and distributed areas in the vertebrate brain (Fig. 1a)[2–9]. A major retinorecipient area in rodents is the superior colliculus (SC) in the midbrain[6,10], referred to as optic tectum (OT) in non-mammalian vertebrates. The SC is an evolutionary old brain structure that is part of the extrageniculate visual pathway[11] and is central for visually guided behaviors[12,13]. While we have learned much about how SC neurons process visual stimuli[14–26], how SC neurons integrate retinal activity on a functional level in vivo is still largely unknown[27].

There are multiple mechanisms that could possibly explain how SC neurons could integrate RGC inputs. SC neurons might be driven by sparse but strong RGC inputs (Fig. 1b, top) such that individual RGCs can drive SC spiking activity. Alternatively, SC neurons could receive numerous but weak RGC inputs, in which case simultaneous activation

of multiple pre-synaptic RGCs is required to drive SC spiking activity (Fig. 1b, bottom). These two distinct wiring schemes have implications for how SC neurons represent the visual world encoded in the diverse pathways of their retinal afferents. Strong but sparse inputs would indicate that SC neurons reliably represent the activity from a small part of the visual field (Fig. 1b, top) based on inputs from few retinal pathways, in a manner comparable to the retinogeniculate circuit[28,29] and the somatosensory system[30,31]. In contrast, if SC spiking activity is driven by the summation of numerous inputs, SC neurons could generate new representations by combining the activity of multiple and diverse RGC types from a larger part of the visual field (Fig. 1b, bottom), similar to what has been reported in thalamo-cortical visual circuits[10,32–35]. Anatomically, the spatial spread of RGC axonal arbors in SC[4] would support both of these wiring schemes and therefore it is still largely unresolved how SC neurons integrate RGC activity in vivo.

[1]Neuroscience Research Center, Charité-Universitätsmedizin Berlin, Charitéplatz 1, 10117 Berlin, Germany. [2]Bernstein Center for Computational Neuroscience Berlin, Philippstraße 13, 10115 Berlin, Germany. [3]Institute for Theoretical Biology, Humboldt-Universität zu Berlin, Philippstraße 13, 10115 Berlin, Germany. [4]Einstein Center for Neurosciences Berlin, Charitéplatz 1, 10117 Berlin, Germany. [5]Max Planck Institute for Ornithology, Eberhard-Gwinner Straße, 82319 Seewiesen, Germany. [6]Max Planck Institute for Biological Intelligence (in foundation), Eberhard-Gwinner Straße, 82319 Seewiesen, Germany. ✉e-mail: jens.kremkow@charite.de

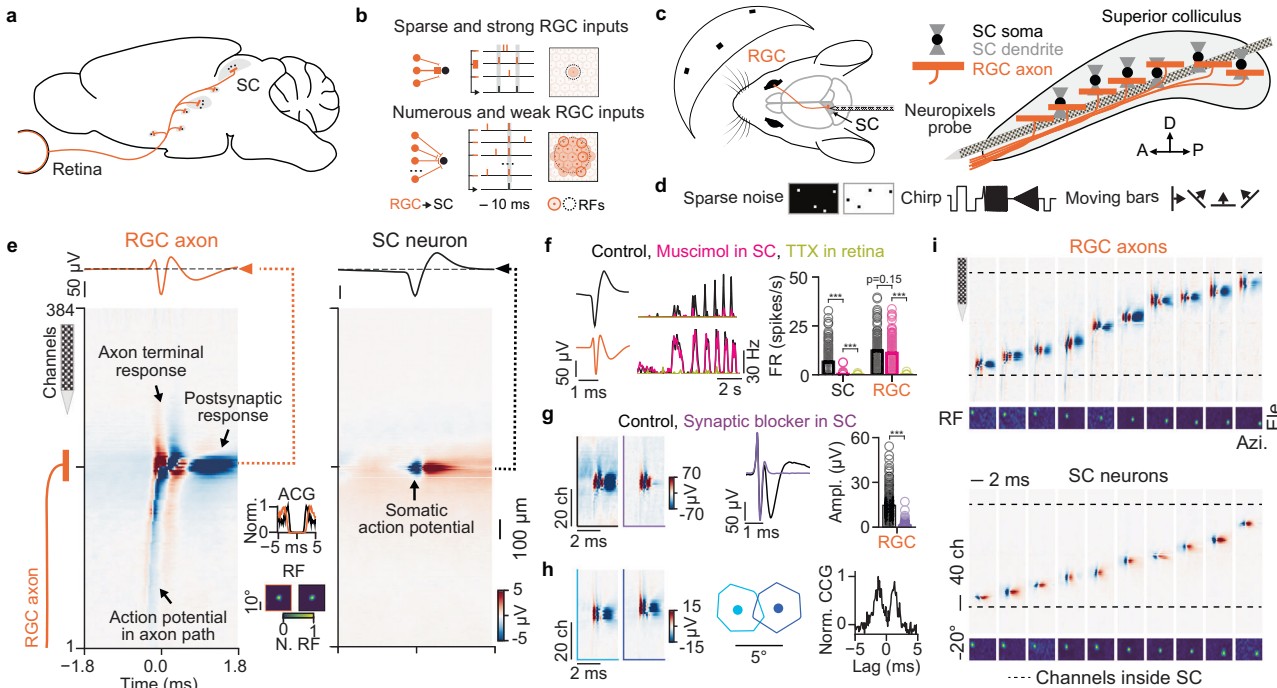

**Fig. 1 | Simultaneous recordings of RGC axons and SC neurons in the mouse in vivo. a** The superior colliculus (SC) receives inputs from retinal ganglion cells (RGC). **b** Possible functional wiring of the retinocollicular connections (left), synaptic integration (middle), and retinotopic precision (right, functional connections shown in darker color). SC neurons could be driven by strong but sparse RGC inputs (top) or by the synchronous activation of numerous weak RGC inputs (bottom). **c** Experimental setup for simultaneous recordings of RGC axons and SC neurons with high-density electrodes in the mouse SC in vivo. The visual dome (left) allows the presentation of stimuli in a large area of the visual field of the head-fixed mouse. Note: SC neurons have diverse dendritic morphologies[79] and only one is shown here. **d** Visual stimuli used to characterize the functional properties. **e** Spatiotemporal electrical signal of an RGC axonal action potential (AP) (left, RGC axon) and somatic SC AP (right, SC neuron). The AP propagating along the path of the RGC axon is visible in the multi-channel waveforms. ACG spike train auto-correlogram, RF receptive field. **f** Pharmacological confirmation

of axonal and somatic waveforms (left). Visually evoked activity of an SC neuron (top) and an RGC axon (bottom) during the different conditions shown as peristimulus time histograms: control (black), muscimol application in SC (magenta), and tetrodotoxin (TTX) injection in the eye (yellow). Firing rates during the different conditions (right). \*\*\*$p = 6.0 \times 10^{-32}$, $3.16 \times 10^{-22}$, 0.154, $1.5 \times 10^{-29}$, $n = 184$ SC neurons, $n = 169$ RGC axons, $n = 5$ mice. **g** Pharmacological confirmation that the second trough in RGC axonal waveforms is postsynaptic evoked activity. \*\*\*$p = 4.62 \times 10^{-35}$, $n = 203$ RGC axons, $n = 3$ mice. **h** Recording from neighboring RGC axons (left). Neighboring RGC axons in SC have close but non-overlapping RF centers (middle) and can show putative electrical coupling in the spike train cross-correlogram (CCG) (right). **i** RGC axons and SC neurons cover a large part of the SC circuit and different retinotopic positions. The SC borders were identified by a continuous retinotopic map within the visual driven channels. Comparisons with two-sided Wilcoxon signed-rank test. Source data are provided as a Source Data file.

For instance, one hallmark of the retina is its organization into mosaics[36–40] that are present even in species with poor visual acuity, like the mouse[9]. In these mosaics, RGCs of the same functional type tile the retina in a quasiregular lattice[36,39–41] which is thought to reflect optimal and efficient encoding of visual scenes[36,40,42]. Whether the retinal ganglion cell axonal arbors maintain the precision of the retina in the SC is unknown but important for understanding the functional wiring of the retinocollicular circuit. Therefore, revealing the functional organization of the retinocollicular circuit is central for advancing our mechanistic understanding of how SC neurons process visual stimuli and their role in mediating visually guided behaviors.

The primary obstacles in answering these questions are technical difficulties in recording RGC activity and axon locations simultaneously with their postsynaptic targets in vivo. The synaptic connectivity between progressive stages of sensory processing is typically assessed using topographically aligned recordings of somatic activity in the two regions of interest[28,32,33,43]. While this method has provided crucial information regarding the mechanisms underlying visual processing[28,32,33,44–48], it provides a low yield of synaptically connected neurons, often restricted to a few pairs recorded simultaneously[28,49], which ultimately limits our understanding of how populations of afferent inputs are integrated within target circuits. In summary, due to technical limitations the functional organization of the retinocollicular circuit is still largely elusive.

The aim of this study was to assess how neurons in the superior colliculus integrate retinal inputs in vivo. To that end we show that high-density electrodes overcome current technical limitations and that measuring the activity of RGC axons simultaneously with their postsynaptic targets in the midbrain at a large-scale in vivo is possible. Employing this method, we then investigate the fine-scale organization of RGC axons in the midbrain and elucidate how midbrain neurons functionally integrate those retinal inputs in vivo. In addition, we demonstrate that the observed wiring schemes and functional patterns are shared principles between mice (*Mus musculus*) and zebra finches *(Taeniopygia guttata)*.

## Results

### Recording RGC axons and SC neurons with high-density electrodes

To study the functional organization of the superior colliculus we used high-density electrodes (Neuropixels probes[50]) to record extracellular neuronal activity in the mouse SC in vivo. The mouse was head-fixed, inside a visual dome[51] that allowed us to present visual stimuli in a large part of the visual field[52] (Fig. 1c). To record neuronal activity in the SC we targeted the visual layers of SC with a tangential recording configuration that places hundreds of recording sites within the optical layer and superficial gray layers of SC[52] (Figs. 1c and S1n). To characterize the visual response properties of the recorded neurons,

we presented light and dark sparse noise, a full-field chirp stimulus and moving bars (Figs. 1d and S1c).

Using this recording approach, we discovered that the high spatiotemporal sampling of the high-density electrodes, together with their low noise level, allows one to distinguish the waveforms of somatic action potentials of SC neurons (Fig. 1e, right) from axonal action potentials of RGC axons (Fig. 1e, left; see Fig. S3 for waveform classification). Both types of waveforms can be sorted into well isolated single unit clusters with clear refractory periods (Fig. 1e, see spike train auto-correlogram "ACG") and corresponding good quality metrics such as action potential amplitude and isolation distance (Fig. S4). The majority of waveforms of somatic action potentials are biphasic and with a small spatial spread (Figs. 1e, right and S4a). In contrast, the waveforms of RGC axons have a larger spatial spread (Figs. 1e, left and S4a) and are composed of fast bi/triphasic components caused by the axonal action potential and the axons terminal responses[53] followed by a second slower trough corresponding to the synaptically induced dendritic activity in postsynaptic SC neurons (Fig. 1e, left, arrows; Fig. S2). We observed that action potentials propagate along an axonal path in the multi-channel waveform view (Fig. 1e, left), with conduction velocities in the range reported from retinal afferents to the SC[54] (Figs. S2a–c, conduction velocity = $3.5 \pm 1.3$ m/s, $n = 283$ RGC axons, $n = 14$ mice). Because RGC axons innervate the SC along the anterior-posterior axis[2,4,9,55] (Fig. 1c) we hypothesized that action potential propagation can only be observed in recordings aligned with the anterior-posterior axis (Figs. S1d and S1f) but not in recordings aligned with the medio-lateral axis (Figs. S1j). In 17 out of 20 recordings along the anterior-posterior axis and in 0 out of 7 recordings along the medio-lateral axis we observed action potential propagations, supporting the interpretation that the axonal waveforms in our recordings are retinal afferents making synaptic connections onto SC neurons. To further test this hypothesis, we performed a series of in vivo pharmacological experiments (Fig. 1f/g and S2d–i, see Methods) in mice in which we had removed most of visual cortex to ensure that the axonal signals do not originate from visual cortex. We injected muscimol, a $GABA_A$ receptor agonist, into the SC in vivo to silence SC neurons and to verify that the triphasic waveforms are signals from long-range axons[56] that innervate SC. As a result, the somatic waveforms were strongly suppressed by muscimol (SC neuron firing rate: control = $6.55 \pm 6.33$ spikes/s, muscimol = $0.29 \pm 0.94$ spikes/s, $p = 6.0 \times 10^{-32}$, two-sided Wilcoxon signed-rank test, $n = 184$ SC neurons, $n = 5$ mice), but the axonal waveforms remained (RGC axon firing rate: control = $12.25 \pm 7.74$ spikes/s, muscimol = $11.05 \pm 7.32$ spikes/s, $p = 0.154$, two-sided Wilcoxon signed-rank test, $n = 169$ RGC axons, $n = 5$ mice) (Fig. 1f, magenta). We then injected a synaptic blocker into the SC to confirm that the second negative waveform component originates from postsynaptic responses in SC neurons and, as predicted, the amplitude of this component was reduced (Amplitude: control = $14.57 \pm 9.04$ μV, synaptic blocker = $1.39 \pm 1.89$ μV, $p = 4.62 \times 10^{-35}$, two-sided Wilcoxon signed-rank test, $n = 203$ RGC axons, $n = 3$ mice) (Fig. 1g). Finally, we applied tetrodotoxin (TTX) to the eye of the mouse which silenced the activity of the axonal waveforms (RGC axon firing rate: TTX = $0.01 \pm 0.13$ spikes/s, $n = 169$ RGC axons, $n = 5$ mice) (Fig. 1f, light green). Together, these findings support the notion that the axonal waveforms originate from the retina and do not arise from other sources, e.g. cortex (Fig. 1f, light green). These results demonstrate that the triphasic waveforms recorded in the SC originate from RGC axons making synaptic contacts with SC neurons.

The small distance between recording sites allowed us to isolate activity from RGCs that are neighbors in the retina (Figs. 1h and S5e), which are characterized by adjacent but non-overlapping receptive field centers and often having similar functional responses to a visual chirp stimulus (Fig. S5c–e). In addition, in such neighboring RGC pairs we were able to occasionally observe putative electrical coupling

between RGCs. This was evident in the double peaks in the cross-correlograms (CCG), which is a defining characteristic of coupling between neighboring RGCs of the same[57] and different type[58] (Fig. 1h and S5e). Well-targeted recordings yielded a high number of simultaneously recorded RGC axons and SC neurons (average number of simultaneously recorded RGC axons = $48 \pm 34$ and SC neurons = $114 \pm 58$, total number RGC axons = 1199 and SC neurons = 1831, $n = 27$ recordings from 24 mice). As expected from anatomy[2], the majority of recorded RGC axons and SC neurons were located in the optical and superficial gray layers (Fig. S1, intermediate gray layer: $n = 37$ RGC axons $n = 86$ SC neurons; optical layer: $n = 641$ RGC axons $n = 891$ SC neurons; superficial gray layer: $n = 361$ RGC axons, $n = 628$ SC neurons; zona layer: $n = 26$ RGC axons $n = 45$ SC neurons. For RGC axons/SC neurons with reconstructed anatomical location using SHARP-track[59,60], see Methods for more details). Moreover, both RGC axons and SC neurons covered a large region across the visual field (Figs. 1i and S1d–k), RGC axons derived from a diversity of retinal pathways[1] (Figs. S5a/b) and SC neurons covered a broad range of functional response classes across the population (Fig. S6).

Taken together, our data demonstrate that high-density electrodes enable recordings of the activity of afferent axonal action potentials simultaneously with the action potentials of post-synaptic targets at a large scale in vivo, both in anesthetized ($n = 24$ mice) and awake mice ($n = 3$ mice) (Fig. S7h–j). Thereby, this method permits the study of fine scale organization of afferent axons and the resulting functional connectivity with their target neurons in vivo.

## Precise spatial mapping between RGC receptive fields and RGC axons

Next, we wanted to reveal the fine-scale spatial organization of multiple neighboring RGC axons in the SC. While previous anatomical work has demonstrated that axons from single RGCs form dense and stereotyped arbors in the SC[4], it remains unknown how the axons of neighboring RGCs are organized in relation to each other within the SC. The location of the RGC dendritic arbor in the retina can be estimated from the receptive field mapped with the sparse noise stimulus and thus can identify neighboring RGCs (Fig. 2a, see Methods). The anatomical location of the RGC axonal arbor can then be inferred from the RGC waveform on the high-density electrode (Fig. 1e/h). Specifically, the recording sites that contain the postsynaptic component of the triphasic RGC waveform identify the anatomical locations where the RGC axonal arbor makes synaptic contacts onto dendrites of SC neurons (Fig. 1e/g and 2a bottom-right). We defined this area on the high-density electrode probe as the RGC axonal synaptic contact field (AF) and used this in vivo measurement as a proxy for the anatomical location of the RGC axonal arbor within SC. Since the recording sites on the Neuropixels probe are organized in a checkerboard pattern with 480 rows (recording site distance = 20 μm) and 4 columns (recording site distance = 16 μm) it is possible to estimate the position and spatial extent of the axonal synaptic contact field along and, to some degree, across the probe (Fig. 2a, see Method for details on how the axonal contact field was fit with a 2d Gaussian function on the probe).

Having established a method for recording the receptive fields and axonal fields of multiple RGCs in vivo, we then investigated how the axons of simultaneously recorded RGCs organize within SC. Figure 2b shows a recording where we captured a large number of RGCs ($n = 76$ RGCs) which revealed that the axonal field positions of the RGCs gradually changed along the probe within SC (Fig. 2b, bottom) with the corresponding receptive field locations varying in elevation and azimuth (Fig. 2b, top). Remarkably, the spatial organization of receptive fields in the retina was preserved at the level of the RGC axonal fields within SC (Fig. 2b, compare top and bottom). This single cell precise spatial mapping between RGC receptive fields and RGC axons was even more apparent when separating RGCs into groups with

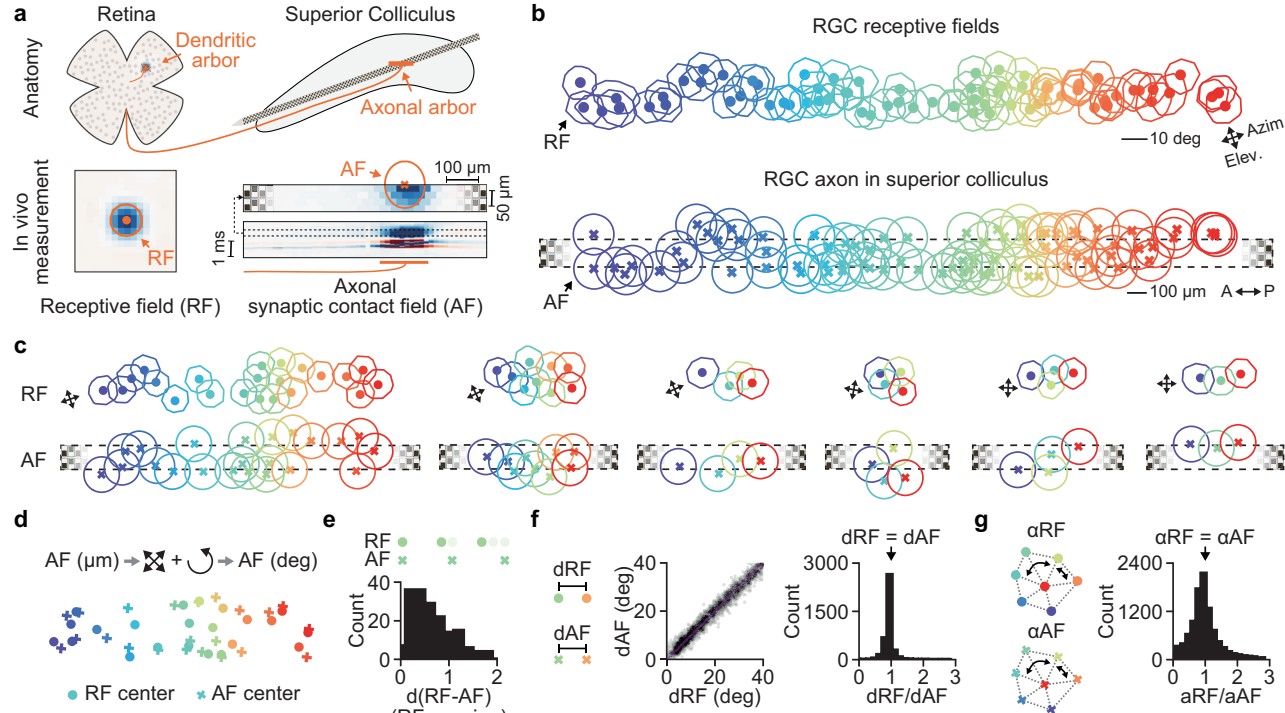

**Fig. 2 | Single cell precise spatial mapping between RGC receptive fields and RGC axons. a** Characterization of RGC dendritic and axonal arbors using in vivo measurements. The spatial location of the RGC dendrites can be estimated in vivo by mapping the visual receptive field (RF) (left). The spatial location of the RGC axonal arbors in SC can be measured in vivo by the axonal synaptic contact field (AF) (right). The AF is the area on the high-density electrode with evoked post-synaptic responses in SC. **b** RFs and AFs of simultaneously recorded RGCs. The color code identifies individual RGCs. Note the similarity between the spatial organization at the level of the receptive fields and axonal fields within SC. **c** Examples of simultaneously recorded RGCs with similar visually evoked activity. The single cell precise spatial mapping between RFs and AFs is evident. Note that the example on the left is a subset of RGCs shown in **b**. **d** Overlay of the RF/AF centers. For the overlay the AF centers was transformed from SC space (μm) into visual space (deg) by scaling and rotating the AF centers. This transformation preserves the geometrical properties of the AF centers. **e** Histogram of the distances between RF and AF centers in the unit of RF spacing. The RF spacing is the median RF distance between neighboring RGCs. The RF/AF above the histogram illustrate the distance at 0, 1 and 2 RF spacing. **f** Distances between RFs and AFs are similar. dRF plotted against dAF and the histogram of the ratio between dRF and dAF (n = 4028 RGC pairs). **g** Angles between RFs (αRF) and AFs (αAF) match shown in the histogram of the ratio αRF/αAF. Angles were measured between triples of RGCs (n = 10337 RGC triples). **e**–**g** from n = 5 mice. Source data are provided as a Source Data file.

similar response properties (Fig. 2c, note that the example on the left is a subset of RGCs shown in Fig. 2b while the other examples are from different recordings. See Methods and Fig. S5c–e for information).

To assess the geometrical similarity between the receptive field and axonal field organization we transformed the axonal field positions from the anatomical space in SC (μm) into the visual space of the receptive fields (deg). To do so, we linearly scaled and rotated the axonal field positions to match the size and orientation of the receptive field positions (Fig. 2d, see Methods). This transformation preserved the spatial relationship between the axonal field centers and allowed us to quantify the similarity between both patterns of organization. We measured the distances between receptive field and axonal field centers of individual RGCs and divided this distance by the RF spacing (RF spacing = median nearest neighbor's RF distance) to express this measurement independently of the spatial resolution of the recorded RGCs. A distance of 0 RF spacing reflects perfect overlap of the receptive field and axonal field centers while a distance of 1 indicates that both centers are non-overlapping and separated by one receptive field. Our results demonstrated that the receptive field and axonal field centers closely overlapped and that both mosaics were geometrically similar (Fig. 2e, median distance = 0.66 ± 0.43 RF spacing, n = 174 RGCs, n = 5 mice). To further characterize the similarity, we compared the receptive field and axonal field distances (dRF and dAF) between RGC pairs and the angles (αRF and αAF) between triplets of RGCs which revealed that the distances (Fig. 2f, median dRF/dAF = 1.0042 ± 0.80, n = 4028 RGC pairs, n = 5 mice) and angles (Fig. 2g, median αRF/

αAF = 1.01 ± 1.69, n = 10,337 angles, n = 5 mice) are similar. Taken together, our data show a close correspondence between the receptive field and axonal field centers and strongly suggest that RGC axons within SC maintain the fine-scale spatial organization of the RGC receptive fields in the retina. Thus, our data demonstrate that, on the level of single cells, RGC axons provide a precise representation of the retina as input to the SC.

## Monosynaptically connected RGC-SC pairs in vivo at a large scale

How do SC neurons sample from this precisely organized retinal input? To answer this question, we studied monosynaptically connected RGC-SC pairs. A key advantage of our method is the simultaneous recordings of RGC axons and SC neurons at sub-millisecond temporal resolution, which permits the identification of synaptically connected neuron pairs in vivo[28,33,44,47,49,61,62]. To assess synaptic connectivity between RGCs and SC neurons, we employed established cross-correlation analysis methods[33,43] (Figs. 3a/b and S7, see Methods). Connected RGC-SC pairs were identified in the spike train cross-correlograms by significant transient and short-latency increases in the spiking probability (Figs. 3a and S7; peak latency = 1.54 ± 0.39 ms, n = 1044 connected pairs, n = 22 mice), a hallmark of monosynaptic connectivity in vertebrate nervous systems[28,32,33,61,63]. Unconnected pairs do not show transient peaks (Figs. 3b and S7b/c). Depending on the number of recorded RGC axons, we could identify up to 229 monosynaptic connections in individual recordings (Figs. 3c and S7),

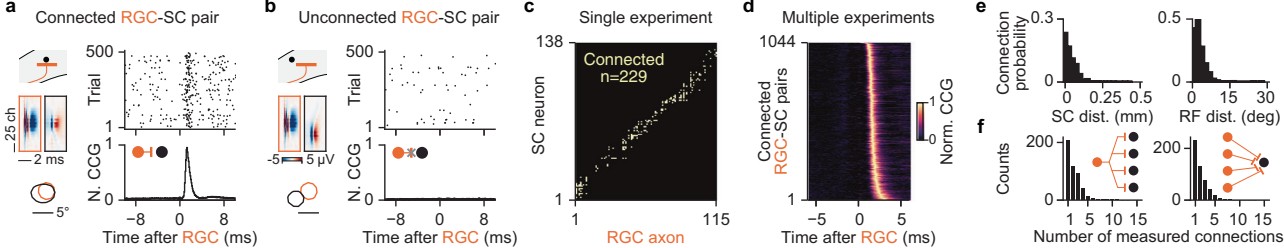

**Fig. 3 | Measuring afferent monosynaptic connections in vivo at a large scale. a** Examples of a monosynaptically connected RGC-SC pair. Raster plot of SC spiking activity triggered on RGC spike times (top). Cross-correlogram (CCG) between the RGC and SC spiking activity (bottom). The short latency peak in the CCG is a hallmark of synaptic connections in vertebrates. Note the close distance of the RGC axonal and SC neuronal waveforms on the probe (middle) and the overlapping RFs (bottom). **b** Unconnected RGC-SC pair. Unconnected pairs do not overlap in SC and visual space. **c** Connectivity matrix between RGC axons and SC neurons recorded in a single experiment. Yellow marks indicate connected pairs. RGC axons and SC neurons are sorted by their location in SC. **d** CCGs of connected pairs across multiple experiments, sorted by peak latency. **e** Connection probability as a function of SC distance and RF distance. **f** Number of measured RGC-SC connections per RGC axon (left) and SC neuron (right). Source data are provided as a Source Data file.

yielding more than one thousand measured connected RGC-SC pairs across multiple experiments in total (Fig. 3d, $n = 1044$ connections from $n =$ from 478 RGC axons and 504 SC neurons, $n = 24$ recordings, $n = 22$ mice). This high number of measured connected pairs resulted from the close proximity of RGC axons and SC neurons on the high-density electrode probe (Fig. 3e, left; median distance within SC = 51.22 μm, first quartile = 25.61 μm, third quartile = 80.00 μm, $n = 1044$ connected pairs) and from similar receptive field locations (Fig. 3e, right, median receptive field distance = 3.35 deg, first quartile = 2.19 deg, third quartile = 5.31 deg, $n = 389$ connected pairs with signal-to-noise of the receptive fields >20). Due to the large number of measured connections, we were able to identify diverging connections from single RGC onto multiple SC neurons (Fig. 3f, left) and converging connections from multiple RGCs onto single SC neurons (Fig. 3f, right). Thus, by recording RGC axons and SC neurons simultaneously on the same high-density electrode our method overcame current limitations and permitted to characterize the functional organization of the retinocollicular circuit in vivo.

## Synaptic organization of the retinocollicular circuit in vivo

Previous studies in cats have shown that single RGC spikes reliably trigger postsynaptic activity in neurons of the visual thalamus, dorsal lateral geniculate nucleus (dLGN)[28,64], and that the majority of dLGN spikes are driven by RGC activity[44]. It is unknown, however, whether this strong drive and coupling are common principles of RGC connections and are therefore also present in the retinocollicular pathway (Fig. 1b, top), or whether the retinocollicular circuit receives weak inputs from numerous RGCs (Fig. 1b, bottom). To differentiate between these distinct modes of signal transmission, we examined the activity of connected RGC-SC pairs (Fig. 4). We observed that individual RGC action potentials can trigger responses in the postsynaptic SC neuron (Fig. 4a, "1") or fail to be transmitted (Fig. 4a, "2"), and that SC APs can occur without input from that specific RGC (Fig. 4a, "3"). To quantify these observations, we estimated the strength of the connection by the efficacy measure and the coupling of the connection by the contribution measure[44] for each connected pair from the spike times of the entire recording. The efficacy is the probability that an RGC input triggers an action potential in the postsynaptic SC neuron (Fig. 4b, left). In the example shown in Fig. 4a–b, the efficacy was ~17% and across the population, we observed a log-normal distribution of connection efficacies, with a few very strong connections up to ~50% efficacy, but primarily weaker connections (Fig. 4c, median efficacy = 2.74%, first quartile = 1.53%, third quartile = 5.07%, maximum = 48.08%, $n = 1044$ connected pairs, $n = 478$ RGC axons, $n = 504$ SC neurons, $n = 22$ mice).

Next, we estimated the connection contribution, which characterizes the fraction of SC action potentials that are driven by the

activity of presynaptic RGCs and therefore provides a measure for how strong SC neurons are coupled to the activity of individual RGC inputs. High contribution values indicate that SC neurons are primarily driven by individual RGC afferent inputs while low contribution values reflect that SC neurons are driven by multiple RGC afferents or inputs from other sources. Our data revealed that SC neurons can be strongly coupled to retinal inputs, such that a large fraction of SC action potentials is preceded by action potentials of individual retinal ganglion cells (Fig. 4b, right, contribution = 45.1% in this example). However, as with efficacy, on the population level we observed many weakly coupled pairs (Fig. 4d, median contribution = 12.50%, first quartile = 7.54%, third quartile = 19.77%, maximum = 78.63%, $n = 1044$ connected pairs, $n = 478$ RGC axons, $n = 504$ SC neurons, $n = 22$ mice). The location of the majority of RGC-SC pairs could be assigned to the optic layer ($n = 633$ pairs) or to superficial gray layer ($n = 271$ pairs) of the SC (Fig. S1n, see Method). We observed statistically significant differences between RGC-SC connections in the two layers (optic layer: median efficacy = 2.46%, first quartile = 1.32%, third quartile = 4.69%; median contribution = 13.27%, first quartile = 8.19%, third quartile = 21.24%, $n = 633$ connected pairs; superficial gray layer: median efficacy = 3.24%, first quartile = 1.71%, third quartile = 5.34%; median contribution = 11.34, first quartile = 5.59%, third quartile = 16.94%; $n = 271$ connected pairs; $p = 0.002$ for efficacy and $p = 6.97 \times 10^{-6}$ for contribution, two-sided Wilcoxon rank-sum test). Since the effect size was small (Cohen's d = −0.09 for efficacy and Cohen's d = 0.33 for contribution) and the differences between the optic and superficial gray layers in SC are thus negligible. Therefore, we pooled the data across all SC layers in all further analyses.

Across the population, we discovered a log-normal distribution of connection efficacy ($p = 0.295$ for testing the hypothesis that the logarithm of the efficacies is not normally distributed using the D'Agostino's K2 test, n pairs = 1044), but not for connection contribution ($p = 7.78 \times 10^{-8}$, D'Agostino's K2 test). Log-normal distributions of connection strength are widely observed in the vertebrate brain[61,62,65,66], including human[67], which could be the result of circuit refinement during development[68] by which only a few strong connections remain after the refinement process. In our data, divergent RGC connections are characterized by only one or a few strong connections with SC neurons and multiple weaker connections (Fig. 4e/f and S6e, efficacy: 1st = 16.25 ± 7.68 %, 2nd = 8.48 ± 5.62%, $p = 2.51 \times 10^{-8}$, two-sided Wilcoxon rank-sum test, $n = 33$ divergent connections with at least three connections and efficacy 1st > 10%). Likewise, only a few RGCs contributed strongly to the spiking of single postsynaptic SC neurons (contribution: 1st = 45.16 ± 11.86%, 2nd = 27.05 ± 9.44%, $p = 1.02 \times 10^{-5}$, two-sided Wilcoxon rank-sum test; $n = 22$ divergent connections with at least three connections and contribution of 1st > 30%). We reasoned that this connectivity motif could be the result of

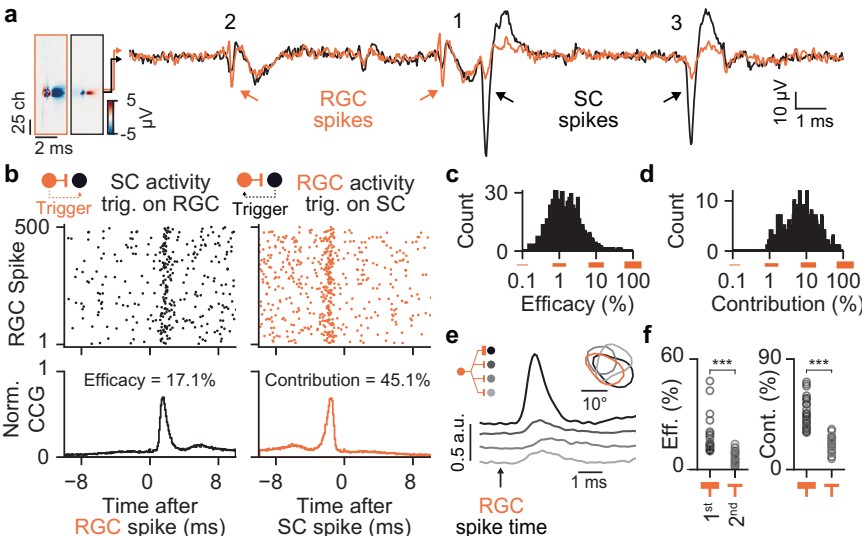

Fig. 4 | SC neurons are strongly driven by retinal inputs. a Example traces showing the electrical signals of a monosynaptically connected RGC axon (orange) and SC neuron (black) pair. (1) the RGC spike triggers a SC spike, (2) failed transmission, (3) SC spike without RGC input. b Strong RGC-SC coupling. SC spiking activity relative to the RGC spike times of the example pair shown in a (left). Note that the RGC input strongly drives spiking in the connected SC neuron, estimated by the connection efficacy. A large fraction of SC spikes is driven by RGC activity, estimated by the connection contribution (right). c Population histogram of efficacy values. Note the log-normal distribution, with few strong and many weak connections. d Contribution of all measured connections. e Example of a divergent connection with one strong and several weak connections. The gray lines show the cross-correlogram of the pairs and the inset shows the receptive field contours of the recorded neurons. f Efficacy and contribution measurements for the strongest and second strongest connections. Efficacy: $n = 33$ divergent connections, $p = 2.51 \times 10^{-8}$. Contribution: $n = 22$ divergent connections, $p = 1.02 \times 10^{-5}$. Two-sided Wilcoxon rank-sum test. Source data are provided as a Source Data file.

the non-Gaussian distributed connection strength. To test this prediction, we performed a permutation test by randomly sampling ($n = 1000$ repeats) connection efficacy and connection contribution of divergent connections from the measured distributions. This permutation test showed that the median of the data fell within the 2.5% and 97.5% percentile interval of the shuffled data for both efficacy (Fig. S7f) and contribution (Fig. S7g).

## Functional organization of the retinocollicular connections in vivo

We next sought to investigate the functional specificity of the retinocollicular connection strength to better understand the observed diversity of the connection efficacies and contributions. In particular we wanted to understand why some RGC-SC pairs were strongly connected, while many RGC-SC pairs showed moderate or weak connections. From work in cats, it is known that functionally similar RGC-dLGN pairs with overlapping receptive fields are strongly connected, while less similar pairs have weaker connections[44]. To assess whether similar wiring rules are at work in the retinocollicular circuit we compared the functional similarity of connected RGC-SC neurons to the connection strength. The functional similarity was measured by the correlation coefficients between the trial averaged visually evoked activity of the connected RGC and SC neurons during the dark and light sparse noise ($r_{SD}$ and $r_{SL}$), chirp ($r_{chirp}$), and moving bars ($r_{m. bar}$) (Fig. 5a). To describe the overall functional similarity by a single similarity value, we then averaged these four correlation measurements. A similarity value of 1 corresponds to visually driven responses that are perfectly correlated while a value of 0 reflects uncorrelated responses. Our data demonstrated that the RGC-SC connection strengths, as measured by the connection efficacy, was positively correlated with the functional similarity of the connected neurons ($r = 0.55$, $p = 1.36 \times 10^{-43}$, Pearson correlation coefficient test, $n = 526$ connected pairs), such that functionally similar RGC-SC pairs were most strongly connected (Fig. 5b, left). Likewise, RGC inputs highly contributed to the spiking activity of their postsynaptic SC neurons when the functional similarity of the RGC-SC pair was high and less when the

functional similarity was low ($r = 0.56$, $p = 5.59 \times 10^{-45}$, Pearson correlation coefficient test, $n = 526$ connected pairs) (Fig. 5b, right). To investigate whether this functional specificity of the connection strength was reflected in the similarity of important tuning properties of neurons in visual circuits, we estimated the preferred direction and preferred orientation of connected RGC-SC pairs using the responses to the moving bar stimulus. We found that connected and direction-selective RGC-SC pairs had similar preferred directions (mean preferred direction difference = $24.23 \pm 29.15°$, $n = 50$ connected pairs), confirming previous results[27], and that connected orientation-selective RGC-SC pairs had similar preferred orientations (mean preferred orientation difference = $10.50 \pm 8.22°$, $n = 7$ connected pairs).

Our results support the notion that retinocollicular connections are organized in a specific manner with functionally similar RGC-SC pairs being strongly connected, suggesting that a large fraction of SC neurons receives limited convergent input from the retina. However, we also noticed cases with relatively strong connections between RGC-SC pairs with low similarity, suggesting that some SC neurons receive convergent input from a functionally more diverse pool of RGC afferents. To clarify the diversity of the presynaptic RGC pools, we took advantage of the large number of available connections in our dataset and studied the convergent connections from multiple RGC axons to single SC neurons (SC neurons with at least three identified RGC connections were included in this analysis, $n = 57$ SC neurons). Our results revealed that a subset of SC neurons received convergent inputs from a functionally homogenous pool of RGCs, such that the evoked responses of the SC neurons were similar to the presynaptic RGC pool (Fig. 5c, note the similarity between receptive fields as well as evoked chirp responses of RGCs and the SC neuron). However, in another set of SC neurons the presynaptic RGC pool was functionally more diverse (Fig. 5d). To quantify this observation, we calculated the correlation of the responses to the chirp stimulus ($r_{chirp}$) among the RGCs of the presynaptic pools and used the average of these correlation values to characterize the functional diversity of the afferent RGC pools. Values close to 1 reflect a functionally homogenous pool of RGC afferents while lower values reflect a more diverse pool of RGC

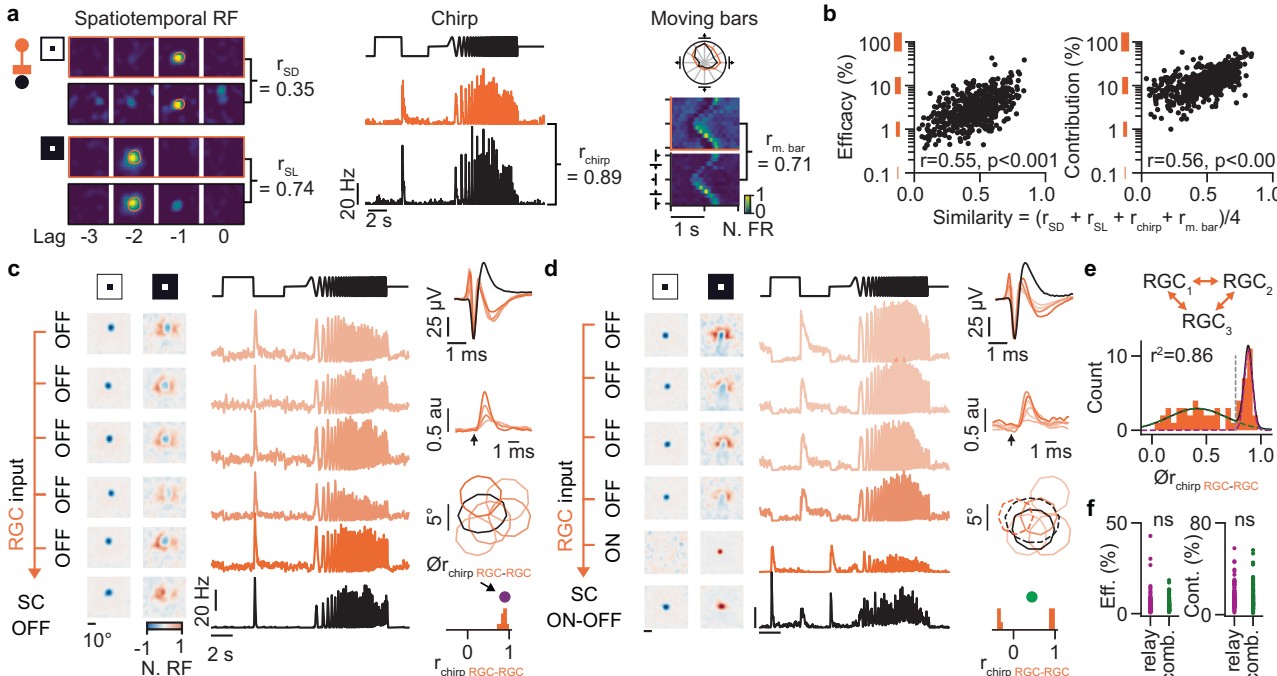

**Fig. 5 | Functional organization of the retinocollicular connections. a** Visually evoked activity in response to sparse noise (left), chirp (middle), and moving bars (right) of a connected RGC-SC pair. The functional similarity of the RGC axon (orange) and the postsynaptic SC neuron (black) is evident and characterized by the correlation coefficients $r_{SD}$, $r_{SL}$, $r_{chirp}$ and $r_{m. bar}$. **b** The connection efficacy and contribution are correlated with the overall functional similarity between the RGC-SC pair. The overall functional similarity is estimated by the average of $r_{SD}$, $r_{SL}$, $r_{chirp}$, and $r_{m. bar}$. $p = 1.36 \times 10^{-43}$ and $p = 5.59 \times 10^{-45}$, Pearson correlation coefficient test, $n = 526$ connected pairs. **c** Relay motif example of an SC neuron receiving convergent inputs from a pool of RGCs with similar functional responses. Receptive fields (left). Responses to the chirp stimulus (middle). Spike waveforms, CCGs, contours of RFs, and the histogram of $r_{chirp}$ between RGC-RGC (orange) (right). The

arrow in the CCG marks the RGC spike time. The magenta dot shows the average $r_{chirp}$ of the presynaptic RGC pool. **d** Combination motif example, same format as (**c**). Both ON and OFF RGCs converge onto the SC neuron. OFF-RFs shown as solid lines and ON-RFs as dashed lines. **e** Functional diversity of RGC convergent inputs to SC neurons. Histogram of the average $r_{chirp}$ between RGCs converging onto the same SC neuron. Note that some RGC input pools are very similar with $r_{chirp}$ values close to 1 while others convey a mixed input with lower $r_{chirp}$ values. Note that the distribution is bimodal with functionally similar pools (relay, magenta) and functionally diverse pools (combination, green) exist across the population of SC neurons ($n = 57$ SC neurons). **f** Connection efficacy and contribution are similar in relay ($n = 104$ pairs, $p = 0.73$) and combination ($n = 138$ pairs, $p = 0.33$). Two-sided Wilcoxon rank-sum test. Source data are provided as a Source Data file.

afferents (the similarity of the responses to the dark/light sparse noise and moving bars were not included in this analysis because those measures are sensitive to the precise retinotopic location of the RGCs and thus can mask the functional similarity measured with the full-field chirp stimulus). Across the population, both SC neurons with functionally homogenous presynaptic RGC pools and also SC neurons with heterogenous pools existed (Fig. 5e, note the bimodal shape in the distribution, $r^2$ bimodal = 0.86, $r^2$ unimodal gauss = 0.16, non-linear least square fit). Despite these differences in afferent inputs, we did not observe systematic differences in connection efficacy or contribution between these two types of RGC pools (efficacy: relay = 5.43 ± 5.84%, $n = 104$ connections, combination = 4.96 ± 3.43%, $n = 138$ connections, $p = 0.73$; contribution: relay = 16.00 ± 9.67%, $n = 104$ connections, combination = 15.13 ± 9.79%, $n = 138$ connections, $p = 0.33$, Wilcoxon rank-sum test) (Fig. 5f). Taken together, the functional similarity between RGC-SC pairs is an important factor in determining the strength of the connection, albeit with different modes of functional convergence.

**Comparing the retinotectal circuit in the mouse and zebra finch**
The retinofugal pathway to the midbrain is highly conserved across all classes of vertebrates[12]. We hypothesized that the wiring principles we discovered in the mouse might be of general nature and therefore likely to be found also in non-mammalian vertebrate species, e.g. birds. To test this hypothesis, we studied the synaptic and functional organization of retinal afferent inputs to neurons in the optic tectum (OT) of the zebra finch by employing our high-density electrode method to

simultaneously measure RGC axons and connected OT neurons in vivo (Fig. S8). Our zebra finch data confirmed several key observations from the mouse: RGC axons in the OT of the zebra finch precisely reflected the receptive field organization in the retina (Figs. 6b/c and S7g–k, finch: median dRF/dAF = 1.02 ± 0.40, $n = 471$ RGC pairs; median αRF/αAF = 1.03 ± 2.04, $n = 2913$ angles, $n = 2$ zebra finches) and zebra finch OT neurons received a small number of RGC afferents (Fig. 6d and S8f) with a log-normal distribution of RGC connection efficacy (Fig. 6e, $p = 0.376$ for testing the hypothesis that the logarithm of the efficacies is not normally distributed, $n$ pairs = 105, $n = 5$ zebra finches, D'Agostino's K2 test) but not RGC connection contribution (Fig. 6f, $p = 0.009$, D'Agostino's K2 test). As a consequence, OT neurons can be strongly driven and tightly coupled to their RGC afferent inputs (Fig. 6f and S8, finch: median efficacy = 4.49%, first quartile = 2.07%, third quartile = 10.16%, maximum = 39.26%; median contribution = 9.28%, first quartile = 4.89%, third quartile = 19.01%, maximum = 58.83%; $n = 105$ connected pairs), with connection strength being positively correlated with the functional similarity of connected RGC-OT pairs (Fig. 6g, similarity vs efficacy: r = 0.62, $p = 6.51 \times 10^{-06}$; similarity vs contribution: r = 0.71, $p = 7.18 \times 10^{-08}$, Pearson correlation coefficient test, $n = 43$ connected RGC-OT pairs). Moreover, both SC/OT neurons sample RGC inputs from a restricted retinotopic area (Fig. 6h). Interestingly, we noticed that there was a gap in the receptive fields' positions along the probe in a zebra finch recording (Fig. S8d), which could be related to a gap of RGC axons in the optic tectum around the representation of the optic nerve head[69]. Taken together, despite the higher spatial resolution of the avian visual system[70] and the large evolutionary distance

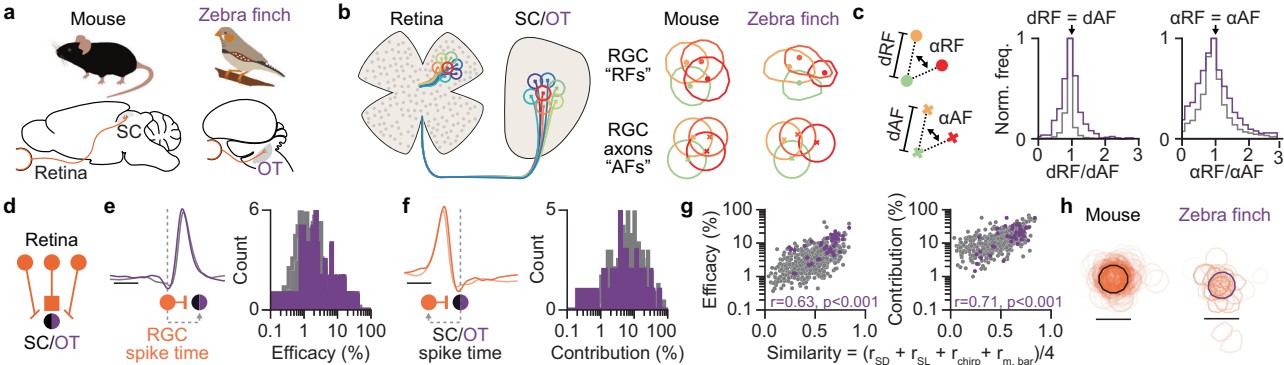

**Fig. 6 | Wiring principles of the retinotectal circuit in mice and zebra finches.** **a** Retinotectal pathway in mouse and zebra finch. The mouse SC and the zebra finch optic tectum (OT) receive inputs from RGCs. **b** RGC axons in SC/OT reflect the receptive field mosaic in the retina. Examples of RGC receptive fields and corresponding axonal fields from mouse and zebra finch are shown. **c** The RGC axonal mosaics provide a single cell precise isomorphic representation of the RGC receptive field mosaics as input to the SC/OT. Note, the mouse data is redrawn from Fig. 2 to allow a comparison to the zebra finch. **d** Limited convergence in the retinotectal circuit. **e** RGC-SC/OT connections efficacies are log-normally distributed in both species. Average RGC-SC/OT cross-correlograms of connected pairs ($n = 1044$ connected pairs) and of connected RGC-OT pairs ($n = 105$ connected pairs) (left). Distribution of connection efficacies in both species (right). The mouse data is replotted from Fig. 4c. **f** RGC-SC/OT connections contributions in both species. Spiking in SC/OT neurons is tightly coupled to the activity of RGC afferents. Average SC/OT-RGC cross-correlograms of connected pairs ($n = 1044$ connected pairs) and of connected RGC-OT pairs ($n = 105$ connected pairs) (left). Distribution of connection contribution in both species (right). The mouse data is replotted from Fig. 4d. **g** Functional specificity of connection efficacy and connection contribution. The plots show the functional similarity vs efficacy/contribution of connected RGC-OT pairs ($p = 6.51 \times 10^{-06}$ and $p = 7.18 \times 10^{-08}$, Pearson correlation coefficient test, $n = 43$ connected pairs). Gray dots replot the mouse data from Fig. 5b. **h** SC/OT neurons integrate RGC inputs from a restricted area of the visual field. Individual RGC receptive fields in orange and the average of the SC/OT receptive fields in black are shown ($n = 218$ RGC-SC pairs; $n = 43$ RGC-OT pairs). Receptive fields from different parts of the visual field were aligned by shifting the receptive fields based on SC/OT receptive field centers. To differentiate weak from strong connections, the receptive field outline transparency was scaled by the connection efficacy, with weaker connections being more transparent. Source data are provided as a Source Data file.

between mammals and birds, our results indicate that retinal afferents are integrated by zebra finch OT neurons according to principles similar to those followed by neurons in the mouse SC (Figs. 6 and S7/8). Therefore, our data strongly support the notion that retinotectal circuit follows similar wiring principles in mice and zebra finches.

## Discussion

### Recording afferent axons with single high-density extracellular electrodes in vivo

We discovered that high-density electrodes capture the electrical activity of RGC axons in the midbrain of mouse and zebra finch. The pharmacological experiments in the mouse revealed that the triphasic waveforms remained active after applying muscimol to the SC in vivo (Fig. 1f). Therefore, the triphasic waveforms cannot originate from neurons within the SC circuit but are signals from long-range afferent axons[56]. Furthermore, the triphasic waveforms resemble the local field potential signature of single thalamic axons in cortex measured via thalamic spike-triggered-averaging of cortical local field potentials using paired recordings[43,53,71]. Considering this data, we conclude that the triphasic waveforms originate from single afferent axons making synaptic connections onto midbrain neurons.

Both retina and cortex provide long-range axonal inputs to SC and thus potentially both structures could be the source of the axonal waveforms. We could observe the streak of the propagating action potential only in the antero-posterior recordings but not in the medio-lateral recordings (Fig. S1). This observation matches well with the anatomy of retinal axons innervating SC[9,72] but less to the anatomy of cortical axons innervating SC[73]. In addition, the spatial spread of the axonal contact field is in the range of the anatomical spread of RGC axonal arbors in SC[4], the visually evoked activity of the axonal waveforms resembles what is known about RGCs (Fig. S5a/b), and applying TTX into the mouse eye abolished the activity of the axonal signals (Fig. 1f). Taken together, we conclude that the triphasic waveforms measured with the high-density electrode in SC/OT are RGC axons making synaptic connections onto midbrain neurons.

Measuring the synaptic contact field of afferent axons using single high-density electrodes in vivo opens up new opportunities to investigate the organization and function of long-range axons in vivo. However, it is still unclear what axonal morphologies generate electrical signals with amplitudes large enough to be captured by high-density electrodes. RGC axons form dense arbors within SC and modeling work suggests that axonal branching plays an important role in generating axonal extracellular potentials[74]. Thus, this method could potentially be employed to study long-range axons with dense arborizations in vivo such as thalamo-cortical axons within cortex[10,75].

### Identifying afferent connections with high-density electrodes in vivo, at large scale

A key advantage of our approach is that the sub-millisecond temporal resolution of the high-density electrodes permit the detection of synaptically connected RGC-SC pairs in vivo[33,49,61] at large scale. Revealing the functional composition of presynaptic pools[76], including afferent connections, has been technically challenging, as it requires recording with multiple carefully aligned electrodes in the afferent and target brain region[32,43,49]. High-density electrodes solve this issue by recording the activity of afferent axons and their postsynaptic target neurons simultaneously on nearby channels on the same probe (Figs. 1d and 3a/e), thereby yielding an unprecedentedly large number of connected pairs in vivo (Fig. 3c/d), even in awake mice (Fig. S7j). This method opens the door for assessing the integration of retinal inputs to SC neurons in awake behaving mice which would allow addressing long-standing questions about the role of SC in visual perception[12] and attention[77]. Finally, this method could potentially also be employed to study the thalamo-cortical circuit in awake behaving conditions[78] with a higher yield of connected thalamo-cortical pairs.

Despite this high yield in identifying connected pairs, the method cannot capture the full constellation of inputs to individual neurons nor unambiguously reveal whether a connection is located on the proximal or distal part of the postsynaptic dendrites (the dendritic arbors of certain cell types can span several hundred μm[79]). The wide

range of physical distances between RGC axons and SC neurons on the probe (Fig. 3e) suggests that both proximal and distant connections were captured by our method. Furthermore, we could identify multiple (3–5) converging RGC inputs to SC neurons (Fig. 3f), which is in the range (~5) of the reported number of converging RGC neurons onto SC neurons estimated electrophysiologically in vitro[80]. Thus, based on this number our approach can sample a fair amount of the presynaptic RGC pool of individual SC neurons, although such a high sampling is achieved only in a subset of SC neurons (Fig. 3f). Since the anatomical evidence of the number of presynaptic RGCs of SC neurons is still an open question our numbers represent a lower bound and an under sampling is likely, in particular for weak connections that do not reliably evoke spiking activity in SC neurons.

Alternative approaches to study the functional organization of retinal inputs in SC could be RGC bouton imaging[81] or single-cell-initiated transsynaptic tracing[82]. While viral tracing would provide a more complete view on the presynaptic RGC pool of individual SC neurons, this method is limited in that it does not reveal the connection strength[83]. Future studies using these molecular methods could provide important insights into the cell-type specific wiring of the retinocollicular circuit, e.g. by combining the viral approaches with cell-type specific transgenic mouse lines[84].

### Revealing synaptic inputs from afferent connections in vivo

Our work also demonstrated that it is feasible to measure and characterize sub-compartments of afferent connections in vivo, including synaptically evoked dendritic responses in postsynaptic neurons[53], on a single high-density electrode (Fig. 1d). Previous work using spike triggered field potentials revealed important insights about the functional and synaptic properties of thalamo-cortical circuit[43,53,85–87]. However, these studies required technically challenging paired recordings with well aligned electrodes. Our method overcomes this technical challenge by measuring the action potential and location of the afferent axon together with the evoked synaptic field simultaneously on the same probe. Recent work using planar high-density electrodes showed that it is possible to characterize connectivity and function using extracellular electrodes in ex vivo preparations[88]. Our study now shows that high-density electrodes can be used for studying synaptic physiology of afferents in vivo, which opens up new avenues for how to probe both function and structure of neural circuits, including the plasticity of afferent synaptic inputs.

### Spatial precision of the retinotectal circuit

A key finding of our study is that the retinotectal circuit is organized in a highly spatial-precise manner, providing important insights into the role of SC in visually guided behaviors. It has long been known that retinal ganglion cells are organized in precise mosaics in the vertebrate eye[36,40,42] that are thought to be related to the efficient encoding of natural visual stimuli[40]. Whether this spatial precision is maintained at the output level of the retinal axons has remained an open question since the discovery of retinal mosaics more than 170 years ago[39,89]. Our experimental data show an extraordinary correspondence between the spatial organization of the retina at its input (receptive fields) and its output (axons). While such an isomorphic representation of the retinotopic map on a larger scale is a known hallmark of the visual system, including the superior colliculus in the mouse[55] and the optic tectum in the zebra finch[90], the single cell precision of this mapping at the level of the RGC axons in the midbrain has not been shown before. Thus, our data suggest that a key role of the retinotectal wiring is to provide a faithful representation of the visual world to neurons in the midbrain.

Although our dataset included a wide diversity of RGC types (Fig. S5a/b), we have grouped RGCs into putative functional types based on the similarity of the responses to the chirp stimulus and based on the sparse noise receptive field only (Fig. 2c). Therefore, it could be that not all RGCs were classified appropriately. The main result holds true when pooling across RGCs independent of their functional responses (Fig. 2b, e–g), supporting the conclusion that the precise axonal wiring is a general principle. It remains to be clarified whether all RGC types follow this precise organizing principle or whether differences across RGC types and location in the retina[91–95] exist. Moreover, the small width of the Neuropixels probe only provides a narrow sampling of neuronal tissue in two dimensions. While several important properties of neighboring RGC axons could be revealed using this method (Fig. 2 and S5), characterizing the full complexity of the three-dimensional organization of RGC axons within SC requires further investigations. Two-photon calcium imaging of RGC axons in SC would be well suited to further deepen our understanding of the functional organization of RGC axons in SC in 2D and potentially also 3D using multi-plane imaging[96], in particular when combined with transgenic mouse lines that label genetically identified single RGC types[2]. Finally, what developmental mechanisms underlie this single cell precise mapping from the retina to the midbrain and whether this precision is unique to vision or a general principle of sensory afferent organization in the midbrain[73,97] are both open yet important questions.

### Functional specific retinotectal connection strength

Previous work has shown that retinal inputs to dLGN neurons in the cat are strong and driving[28,44,64]. Our data in the mouse and zebra finch now provide evidence that midbrain neurons are also strongly driven by individual retinal afferents suggesting a general principle of retinofugal wiring. It is worth noting that while we observed many weaker RGC-SC connections, connection efficacy and connection contribution were correlated to the similarity of the connected RGC-SC pair (Fig. 5b), with efficacies in the range of up to 40-50% and with contribution values up to 70–80%. These values are similar to the range reported for RGC-dLGN connections in cat[44], which are considered to be strong driver connections[98]. Importantly, connection strength increases with the similarity of the connected RGC-dLGN pair[44], which is in line with our data of RGC-SC/OT pairs and thus supports the idea that the weaker connections most likely reflect non-optimal RGC inputs to SC/OT neurons. Alternatively, the various RGC types could have specific connection strength to the diversity of SC neuron types (Fig. S6). Although we observed a significant negative correlation between the RGC-SC connection contribution and orientation selectivity of the pre-synaptic RGC ($r = -0.28$, $p = 2.38 \times 10^{-08}$, Pearson correlation coefficient test, $n = 379$ connected pairs), more work is required to fully answer this question. Taken together, the efficient way SC/OT neurons integrate RGC inputs is reminiscent of the way neurons in the dLGN integrate retinal inputs[29,44] (but see[99]). Furthermore, this observation stands in contrast to the thalamocortical system in which thalamic afferent inputs to cortical excitatory neurons are weak[32,33,100,101] and the synchronous activation of multiple thalamic afferents is required to drive cortical spiking[32,100,102]. Thus, the main brain regions involved in visual processing, the midbrain and visual cortex, integrate their afferent inputs in different ways, suggesting potential distinct roles in sensory processing and visually guided behaviors.

### Different wiring modes of the retinotectal connections

While the functional similarity between connected RGC-SC/OT pairs was correlated to the connection strength, we also observed midbrain neurons that integrated inputs from various RGC types (Fig. 5). This is similar to the situation in the retinogeniculate circuit in which both relay and combination modes of integration have been reported[82,99]. It remains unresolved what determines whether an SC/OT neuron receives functionally specific or diverse retinal inputs. One possibility is the downstream targets of the postsynaptic SC/OT neuron[103]. Another possibility is the location within the SC/OT circuit, the retinal pathway[1,6] or cell type of the SC/OT neuron since it is known that

diverse populations of excitatory and inhibitory neurons exist in SC[12,14,16,19–23,79,104–106]. However, the full diversity of SC cell-types is still being investigated[104,105,107] and it remains to be shown whether SC cell-type specific wiring rules exist[84,103]. While our dataset contains a diversity of SC neuronal types (Fig. S6) we did not observe any obvious systematic differences between those neurons and more work is needed to clarify which retinal pathways are combined or relayed at the level of the diverse SC neuronal types. Answering this question is important for gaining a mechanistic understanding of the retinocollicular circuit in visually guided behaviors. This could be achieved by combining our tangential high-density recording method with optogenetic approaches to identify SC cell-types[52] or by using transgenic mouse lines in combination with transsynaptic viral tracing[103].

### Similar fine-scale organization of the midbrain across different species

Our results show that key observations in the mouse SC, e.g. the precise RGC axonal organization and the functional specificity of connection strength, are also found in the zebra finch optic tectum. This is interesting given that the spatial resolution of neurons in the optic tectum of zebra finch is higher compared to neurons in the mouse superior colliculus (Fig. S4). Moreover, while RGC axons innervate the visual layers of the mammalian SC from the ventral part[9,55], RGC axons in the avian visual system grow into the OT from the outside of the most superficial layers[70]. Despite these functional and anatomical differences on the macroscopic level, the fine scale organization of the RGC axons within the target layers of the midbrain and the resulting functional connectivity appear to be comparable between mammals and birds. This strongly suggests that the highly precise wiring of the retinotectal circuitry that we discovered is essential for visually guided behaviors in mouse and zebra finch and potentially in all vertebrates.

## Summary

In summary, we showed that the retinotectal circuit in both mouse and zebra finch is characterized by limited convergence and log-normally distributed connection strength, with connection strength being strongest for functional similar RGC-SC/OT pairs. This precise wiring extends the single cell precise isomorphic mapping of retinal mosaics to the axonal input level in the midbrain. Because the functional organization of the retinotectal circuit is similar in mouse and zebra finch and resembles the organizational principle of retinal inputs to visual thalamus in cat and mouse[28,29,64], we propose that retinofugal connections follow a canonical wiring pattern that provides a precise and reliable representation of the visual world to neurons across the different targeted regions in the vertebrate brain.

## Methods

### Animals, surgery, and preparation

All experiments were pursued in agreement with the local authorities upon defined procedures (Landesamt für Gesundheit und Soziales - LAGeSo Berlin - G0142/18 and Regierungspräsidium Oberbayern - ROB-55. 2-2532. VET_02-18-182). During all experiments, maximum care was taken to minimize the number of animals used and their discomfort. Mice: Adult male mice (C57BL/6 J) from the local breeding facility (Charité-Forschungseinrichtung für Experimentelle Medizin, $n = 20$) and Charles-River Germany ($n = 7$) were used. Induction was achieved with isoflurane (2.5% in oxygen Cp-Pharma G227L19A). Once anesthetized, the surgery was performed in a stereotactic frame (Narishige) with a closed-loop temperature controller (FHC-DC) for monitoring the animal's body temperature. The isoflurane level was gradually lowered during surgery (0.7–1.5%) while ensuring a complete absence of vibrissa twitching or responses to tactile stimulation. During surgery, the eyes were protected with eye ointment (Vidisic). For awake mouse recordings ($n = 3$), the head post was implanted two weeks before the recording day and protected with silicone elastomer sealant

Kwik-Cast (WPI Germany). The analgesic metamizole (200 mg/kg, Zentiva-Novaminsulfon) was administered in drinking water after head post implantation for a recovery period of 3 days. After recovery, the animals were gradually habituated to the recording setup. The craniotomy was performed on the day of recording for anesthetized ($n = 24$) and awake mice ($n = 3$). Zebra finches: Adult male zebra finches (>180 days post-hatching) were obtained from the local breeding facility at the Max Planck Institute for Ornithology in Seewiesen ($n = 7$). Birds were anesthetized with isoflurane (1–3% in $O_2$) and head-fixed in a stereotactic instrument (Kopf) while the body temperature was maintained at 40 °C with a homeothermic monitoring system (Harvard Apparatus) with the head tilted by 45 deg to the azimuthal plane. For all experiments, a dental cement-based crown (Paladur, Kuzler) was used to fix the head post and grounding.

**Pupil tracking in the awake mouse.** To monitor pupil position and dilation in awake recordings, we captured the contralateral eye on a camera (Basler acA 1300) equipped with a zooming lens (850 nm bandpass filter, ThorLabs) using a custom written pupil tracking software. The eye was illuminated with an infrared light source (ThorLabs LZ1-10R602). To avoid interference between the camera with the visual stimulus, eye tracking was performed via a dichroic mirror (Semrock, FF750-SDi02-25×36) that was placed between the eye of the animal and the stimulus screen. The pupil size and position were extracted via DeepLabCut (2.1)[108] and analyzed using custom-written scripts in Python (Fig. S7h).

### Visual stimulation

Visual stimuli were generated in Python using the PsychoPy[109] toolbox. The onsets of the visual stimuli were marked by a TTL signal that was generated and time-locked to the screen update on the stimulus computer and recorded together with the neuronal signals from the Neuropixels probe. Visual stimuli were presented in a spherical visual dome (EBrilliantAG, IP44, diam = 600 mm)[51] using a projector (NEC ME331W, refresh rate = 60 Hz, mean luminance = 110 cd/m², Gamma corrected) to cover a large part of the visual field. The image was projected into the dome using a plexiglass reflecting half bowl (Modulor, 0260248). A layer of broad-spectrum reflecting paint (Twilightlabs) was applied inside the dome to improve the brightness of the reflected image[51]. To accurately project the image from the projector via the spherical mirror onto the domes surface we applied a warping in PsychoPy that was estimated using the meshmapper software (http://paulbourke.net/dome/meshmapper). The resulting projector image covered an area of around 180 deg azimuth and 110 deg elevation. The head-fixed mice and zebra finches were positioned on a platform inside the dome such that the eye, contralateral to the recorded SC/OT, was facing the projector image on the dome surface (Fig. 1). In a subset of experiments, we used an LCD display (Dell S2716DG, refresh rate = 120 Hz, mean luminance = 120 cd/m², Gamma corrected but without sphere mapping) instead of the visual dome because additional equipment required more space, e.g. the injector during the pharmacological experiments or the camera for pupil tracking in the awake experiments. The center of the LCD screen was aligned to the pupil resting position employing a semi-online receptive field analysis[52], 64 deg lateral to the nose of the mouse[110] and 62 deg lateral to the beak of the finch[111].

**Sparse noise for receptive field mapping.** To characterize receptive fields, we presented sparse noise targets of varying size and contrast polarity for 100 ms in a pseudo random manner on a grid of 36 × 22 positions. The grid spacing was 5 deg and the grid covered 180 × 110 deg of the visual field. The sparse noise targets were either dark (on light background) or light (on dark background) to characterize the ON and OFF receptive fields. Because the number of possible grid positions was very high, we presented multiple sparse noise

targets simultaneously but in non-overlapping positions at a given time to increase the number of repeats per grid position[112]. We used three different target sizes presented in separate sequences with varying number of targets per frame and trials per position (5 deg targets = 6 targets per frame and 50 trials per position; 10 deg targets = 4 targets per frame and 30 trials per position; 15 deg targets = 2 targets per frame and 20 trials per position). The sparse noise sequences were generated once, saved and the same sequences reused across the different experiments.

**Full-field chirp.** To characterize the contrast polarity, temporal frequency as well as contrast response properties we presented a full-field chirp stimulus[1]. The full-field stimulus varies in brightness: it starts with a gray background and several light decrement and increment steps (-2.18 s black, -3.28 s white, -3.28 s black, 2.18 s gray) followed by sinusoidal intensity modulations with increasing frequency (0.5 Hz to 11 Hz) at full contrast (8.75 s) and increasing contrast (0 to 100%) at 0.4 Hz (8.75 s) and ending with 2.18 s gray background.

**Moving bars.** To measure the orientation and direction tuning, we presented moving white bars on a dark background. The bars moved in 1 out of 12 directions (30 deg spacing between directions) on every trial at a fixed speed of 90 deg/s. The bars were 10 deg in width and with a length that covered the entire projector image/screen.

**Visual stimuli in the zebra finch experiments.** In the zebra finch experiments, the same stimulus set as in the mouse was used. Due to the higher spatiotemporal resolution of the zebra finch visual system, we had to adjust several stimulus parameters. Sparse noise: the sparse noise target and grid size was reduced to 2.5 deg. Full-field chirp: the chirp stimulus was presented at a four times higher update rate (8.75 s instead of 35 s).

## Electrophysiological recordings

Neuronal activity in the mouse superior colliculus and the zebra finch optic tectum was recorded using high-density electrodes. In this study, Neuropixels probes (Phase 3a and Phase 3B1[50]) were used together with the Open Ephys software (www.open-ephys.org, 0.5.5.2) for data acquisition. Phase 3a probes were used with the phase 3 A hardware and Phase 3B1 probes were used with the PXIe system (National Instrument NI-PXIe-1071). The extracellular signals were amplified on the Neuropixels probe and stored in the local field potential band (0.5–500 Hz) and the action potential band (0.3–10 kHz). To target the superior colliculus in the mouse and the optic tectum in the zebra finch, the Neuropixels probe were inserted stereotactically and in a tangential manner using a micromanipulator system (NewScale, MPM-M3-LS3.4-15-XYZ Upright). All stereotactic coordinates were defined according to their distance to lambda, either in the medio-lateral (ML), dorso-ventral (DV), or antero-posterior (AP) axis. All angles and coordinates were recorded in reference to the azimuthal plane at lambda (Paxinos and Franklin, Nixdorf 2007 stereotaxic atlas). The Neuropixels probe was inserted either tangentially in the superior colliculus from the back (Figs. S1b/d, antero-posterior insertion: 15 to 25 deg, 500 to 1200 μm ML, −100 to −500 μm DV, −100 to −300 μm AP from lambda) or from the side (Figs. S1b/h, medio-lateral insertion: 20 deg to 30 deg, −100 to −500 μm DV, 0 to 900 μm AP). The angles in the antero-posterior insertions were measured in reference to the azimuthal plane, with the probe initially aligned to the brain midline so that it remained within a sagittal plane. Similarly, the angles in the medio-lateral insertion were measured in reference to the azimuthal plane, with the probe being perpendicular to the brain midline in order to stay within a coronal plane. In the zebra finch, the insertion was performed along the antero-posterior axis (within sagittal planes) at 40 deg from the azimuthal plan (Fig. S8a/b, in reference to lambda: 3000 to 3800 μm ML, −4250 to −5000 μm DV, 4000 to 4800 μm AP).

In all recordings, the Neuropixels probe was lowered >4 mm into the target region followed by a small withdrawal of 20 to 50 μm to release accumulated mechanical pressure. The probe was allowed to settle for ~10–20 min before visual receptive fields of the multi-unit-activity was mapped using a sparse noise stimulus to confirm that the visual stimulus covered the retinotopic positions of the recorded neurons[52]. If visually driven activity was obtained on at least 50 channels, the data acquisition was started and visual stimulus set was presented. Otherwise the probe was relocated to a different position.

## Pharmacological experiments

The pharmacological experiments were designed in multiple stages: (1) control, (2) muscimol injection, (3) synaptic-blocker injection and (4) tetrodotoxin (TTX) injection into the eye. In each of these stages, a reduced test stimulus set (15 deg dark/light sparse noise sequences, chirp and moving bar) was presented to assess the visually driven activity. During the pharmacological experiments, the visual cortex was removed during surgery to avoid cortically driven visually activity in the SC[15,113]. To that end, the skull was open above visual cortex (1 mm to 3 mm lateral from midline and −2 mm to −4 mm from Bregma) and the underlying cortex was manually aspirated via a pipette. To inject muscimol and the synaptic-blockers into the superior colliculus, the injector (Drumond, Nanoject II) was inserted vertically in the top part of the superior colliculus (lambda: 500 to 1000 μm ML, 200 to −200 μm AP, 1100 to 1400 μm DV) before inserting the Neuropixels probes to decrease movement-related artifacts. Once the Neuropixels probe was properly placed approximately 250 nL of the pharmacological cocktails were injected in the different stages of the experiment (Fig. S2f). In the muscimol mixture, Cholera Toxin subunit B and Alexa 488 Conjugate (C22841, Invitrogen) were added to the muscimol solution (Abcam, ab120094, 2.5 mM in PBS) and injected into the superior colliculus and allowed to diffuse for about 5 min before the test stimulus set was presented. After the muscimol stage of the experiment the injector was slowly removed and the solution in the injector replaced with the of synaptic blocker mixture (Dextran (Fluorescein, 10,000 MW, anionic, D1821), muscimol, NBQX Biozol-HB0443, and D-AP5 Biozol-HB0225 at 2.5 mM, 2.5 mM, 5 mM, respectively, in PBS). The injector was then lowered again into the superior colliculus, using the coordinates of the first injection, and the synaptic-blockers were injected and allowed to diffuse for about 5 min before the test stimulus set was presented again. Experiments with synaptic blocker mixture were considered unsuccessful when we encountered a problem in the second injection (n = 3 successful double-injections). Finally, at the end of all pharmacological experiments, a small volume of TTX (~15 μL, Biozol-HB1034, 100 μM in PBS) was applied in the contralateral eye to abolish all remaining visually-driven retinal activity, and a final chirp stimulus was presented to confirm an absence of visually driven activity.

## Histology and probe localization

**Histology.** For histological reconstruction of the electrode track, the probe was removed and re-inserted in the same location coated with DiI (Abcam-ab145311) diluted in ethanol. The animal was then sacrificed either with isoflurane (>4%) or a subcutaneous injection of a Ketamine-Xylazine mix (Ketamidor 1 g/mL, Rompun 2%). Cardiac perfusion was performed with phosphate buffer saline solution (PBS) followed by 4% paraformaldehyde (PFA) in PBS. The brains were post-fixed overnight in 4% PFA and stored in PBS until histological slicing was performed using a vibratome (Leica VT1200 S). The brain slices were mounted in DAPI-Fluoromount-G (70–100 μm slices, Biozol Cat. 0100-20). Perfused zebra finch brains were transferred to 15% sucrose in PBS for 24 h post-fixation, and then they were moved into 30% sucrose for at least 12 h prior to sectioning with a cryostat microtome. The optic tectum was sliced into 90 μm sagittal sections, mounted

using DAKO (Agilent), and the recording location was confirmed by visually inspecting the recording track post hoc in brain slices.

**SHARP-track analysis in the mouse.** To identify the Neuropixels electrode track in 3D and to localize recording sites to brain regions, we used SHARP-track[60]. SHARP-track allows reconstructing the location of the Neuropixels probe in 3D within the Allen Mouse Brain Common Coordinate Framework based on the histology and physiological landmarks[59]. To align and scale the estimated recording site positions, we used two physiological landmarks that demarcate the SC circuitry, i.e. the first and the last recording site exhibiting visually driven multi-unit activity (Fig. S1m). An additional criterion was a continuous retinotopic map between these two landmarks (Fig. S1m, right). This additional constraint was necessary because antero-posterior insertions can contain visually driven activity in regions anterior to SC such as the pre-tectal areas or the pre-optic nucleus. In this case, the lower SC landmark is chosen as the channel which exhibits a discontinuous change in retinotopy, indicating a boundary between different visual circuits[114] and in our case between the SC and more anterior circuits. From the SHARP-track analysis, we extracted the brain regions of each recording site and assigned each RGC axon and SC neuron based on their best channel to the corresponding brain region.

## Data analysis
Data analysis was performed in Python 2 & 3 (www.anaconda.com) and Kilosort2 and 2.5[115], a MATLAB 2018 & 2019 (www.mathworks.com) package for spike sorting electrophysiological data.

## Statistics & reproducibility
Statistical tests were performed with the Wilcoxon rank-sum test (two-sided) for unpaired samples and with the Wilcoxon signed-rank test (two-sided) for paired samples using the scipy.stats module, unless stated otherwise. Population results are indicated as mean ± standard deviation unless stated otherwise. Reproducibility was obtained as indicated in the figure legends and main text. Data were included based on signal-to-noise and goodness-of-fit criteria, as described in the individual Methods sections. The sample size was limited to reduce the number of used animals and keep enough statistical power as predefined in ethical approvals. The experiments were not randomized.

## Multi-unit activity extraction and spike sorting
**Multi-unit activity.** For the multi-unit activity (MUA) analysis, the raw action potential band signals from the Neuropixels probe were median subtracted (across channels and time) and band bandpass filtered (Butterworth filter order 2, 0.3 to 3 kHz). Action potentials were then detected in each channel at a threshold of 4 standard deviations.

**Single-unit spike sorting.** Kilosort2 and 2.5[115] (https://github.com/MouseLand/Kilosort) were used for sorting detected spikes into isolated single-unit clusters followed by manual curation using Phy2

(https://github.com/cortex-lab/phy). Double-counted spikes within ±0.16 ms were removed from each cluster[47]. In case overlapping spikes between clusters were observed above chance in sharp zero-lag peaks in the cross-correlograms (CCG, peak windows ±0.5 ms) we re-evaluated the cluster in Phy2 and either refined the cluster assignment or removed the problematic cluster(s) from the dataset. Inter-spike-interval (ISI) violations were calculated as the ratio of the spikes within the refractory period (±1.5 ms) to the total number of spikes. Clusters with ISI > 0.05% were removed. Isolation distance was used as another quality metric to identify well isolated clusters (isolation distance > 10 a.u.). Moreover, only single-unit cluster that were stable over the entire recording duration were included in the dataset. Other quality metrics, e.g. silhouette score, were calculated using the ecephys spike sorting modules (https://github.com/AllenInstitute/ecephys_spike_sorting) to compare the quality of the single-unit clusters between RGC axons and SC/OT neurons.

## Waveform analysis of RGC axons and SC/OT neurons
**Waveform classification.** To distinguish action potential waveforms from RGC axons and SC neurons a classification approach was employed. For each single-unit cluster, we calculated the multi-channel waveform (MCW) by spike-triggered averaging (STA) all 384 raw Neuropixels action potential band channels using the spike times for each cluster (up to 50,000 spike times, ±10 ms STA window), following an offset correction for each channel. The multi-channel waveform thus represents the spatiotemporal profile of the action potentials and RGC axons and SC signals could be classified based on their distinct waveforms (Figs. 1e and S3). We used a two-step approach for this classification. First, a custom-written graphical-user-interface (GUI) was used to manually label the cluster. This GUI was based on (1) the characteristic presence of axonal and dendritic negative peaks within 3 ms (Fig. S3), and (2) the possible presence of the axonal path in antero-posterior recordings (Figs. 1e/i, S1f/g and S3). In a second step, we compared our manual classification to an automatized classification using a Gaussian mixture model (GMM) on a principal component projection of classical waveform features (Fig. S3a, Table 1). The optimal number of principal components (PC) that capture sufficient variance in the dataset was estimated heuristically, using the elbow method[116] illustrated by the scree plot representation (Fig. S3b). A scree plot represents the percentage of the variance contained in each PC, ordered by descending values (Fig. S3b). The "elbow" point in such a graph is identified as the PC number where the curve changes from a steep slope descent, to a linear, gradually descending slope – defining thus an optimal balance between the lowest number of components used and the cumulative variance explained between. In our case, beyond $n = 2$ components, the curve resorts to a linear slope descent, thus, the lowest number of components that could explain the maximum variance of the dataset was chosen as 2. Using the MCW, the spatial spread of interest (Σ) was estimated by the number of neighboring channels with amplitudes >15% of the cluster maximum at the best channel (BC), which is the channel with the

## Table 1 | Criteria for waveform analysis

| Waveform Amplitude | A1 | A2 | A3 | A4 |
|---|---|---|---|---|
| Criterion for selection | peak between t = −0.6:−0.05 ms | trough between t = −0.25:0.25 ms | peak between t = 0.25:1 ms | trough between t = 1:3 ms |
| **Waveform Duration** | W1*/W2*/W3* | D1 | D2 | D3 |
| Criterion for selection | A2/A3/A4 | time in ms between A2 and A3 | time in ms between A3 and A4 | time in ms between A4 and return to baseline value A0** after A4 |
| **Waveform Slope** | S1*** | S2*** | S3*** | S4*** |
| Criterion for selection | A1 + (0.8 x A1) to A2 − (0.2 x A2) | A2 + (0.2 x A2) to A3 − (0.2 x A3) | A3 + (0.8 x A3) to A4 − (0.2 x A4) | A4 + (0.2 x A4) to Aend# − (0.2 x Aend) |

Top: Criteria for waveform amplitudes. Middle: Criteria for waveform durations. *Peak widths for W1, W2, W3 measured using scipy.signal.peak_widths with rel_height = 0.5. **A0—Baseline amplitude value—Amplitude value at time t = −0.6 ms. Bottom: Criteria for waveform slope. ***Slope of the line fit when waveform reaches amplitude value. #Aend is the amplitude value at time t = 3 ms.

largest amplitude. This window was interpolated in time (10 times) and subsequently smoothed in time and space using a Gaussian filter (sigma time = 0.1 ms, sigma space = 2 recording sites along the probe). The interpolated and smoothed waveform was trough-aligned for more reliable characterizations keeping a pre-trough period of 0.6 ms and post-trough period of 3 ms. For further quantification the waveforms were re-normalized. 14 features were measured on each channel individually (Table 1), and averaged across the channels of the previously defined spatial spread (Fig. S3a). For example, all four slope measurements (S1–S4 in Fig. S3a) were computed between two concurrent peaks/troughs and were calculated from time points where the waveform crosses peak/trough1 (0.8 x peak/trough1) to peak/trough2 (0.2 x peak/trough2). Additionally, a smaller portion of the obtained axonal clusters from KS2 was detected based on their putative dendritic responses (Fig. S3e, right). These clusters were discarded as their detection occurred based on the postsynaptic dendritic responses of SC neurons and not on the action potential of the retinal axons.

**Detecting axonal contact field waveforms in Neuropixels datasets.** The standard Kilosort2 parameters are sufficient to detect axonal contact field waveforms in Neuropixels datasets. Importantly, during the curation in Phy2, the rejection criteria such as "multiple spatial peaks" and "too large spread"[47] should be minimized to increase the number of identified axonal contact field waveforms in the dataset. A key factor for recording axonal signals is a well-placed Neuropixels probe in the SC/OT tissue. To optimize the targeting and the yield of axonal signals, we adapted a semi-online approach that allows the assessment of whether a given insertion contains axonal contact field waveforms. To that end, we recorded ~5 min of neuronal activity and spike-sorted this short dataset with Kilosort2. During the sorting process, Kilosort plots the detected waveforms using the function "make_fig.m", which allows visually inspection of the waveform types in the dataset. To facilitate the identification of axonal contact field waveforms in this plot, we modified the "make_fig.m" code such that the waveforms are sorted by the value around 1.5 ms (which is the time of the second trough in the RGC waveforms). This semi-online analysis allows assessment of whether axonal contact field waveforms are in the dataset, within a few minutes. It can thus be used during a recording session such that if no axonal waveforms are identified the Neuropixels probe can be relocated to a different position. The modified "make_fig.m" is available on our GitHub repository (https://github.com/KremkowLab/Axon-on-Neuropixels-in-Kilosort)[117].

**Functional diversity of RGC axons and SC neurons**
**Diversity of RGC axons.** To characterize the diversity of the RGC axons we adapted a correlation analysis approach from Rosón et al.[29] and correlated the visually evoked RGC axon responses to the chirp stimulus to the 32 RGC types published by Baden et al.[1]. The Baden data was obtained from https://doi.org/10.5061/dryad.d9v38. We estimated the correlation coefficient between the chirp responses of the RGC axon to each of the 32 classes in the Baden dataset and the RGC axon was assigned to the class in the Baden dataset with the highest correlation value (Fig. S5). To compensate for the different sampling rates and signal timescales between the two-photon calcium imaging in the Baden dataset and our recording technique, we down-sampled and smoothed the chirp PSTHs of the RGC axons (time constant = 0.5 s). RGC axons with no modulation to the chirp stimulus were excluded from this analysis (RGC axons with a correlation value of their chirp PSTH to the chirp stimulus below 0.02 were removed, 10% drop out).

**Electrically coupled neighboring RGCs.** Putative electrically coupled RGCs can be identified based on the presence of characteristic double peaks in the spike train cross-correlograms[57]. To identify significant double peaked cross-correlograms, we estimated the baseline between −10 to −5 ms and the peaks on both sides of the zero-lag (−2.5

to −0.5 ms and 0.5 to 2.5 ms). RGC-RGC pairs were considered couple when both peaks were significantly different (>3 x standard deviation) from baseline (Fig. S5e).

**Diversity of SC neurons.** To characterize the functional diversity of the SC neurons we employed an unsupervised clustering approach. To that end, for each SC neuron the visually evoked responses to the chirp, dark sparse noise, light sparse noise and the moving bar stimuli were concatenated. The responses to the sparse noise were extracted from the receptive field peak pixel. The evoked responses to the moving bars (light bar on dark background, 12 directions) were calculated following the method described in[1]. Briefly, in the first step the times at which the bar entered the receptive field (onset response) and the moment when the bar left the receptive field (offset response) were calculated. The trial averaged PSTHs for each direction were then aligned and centered around the onset-response, with a 0.1 ms pre-stimuli, and 0.7 ms post-stimuli time window. The final response array [12 (directions) x 2700 (time points in ms)] was decomposed using singular value decomposition, to obtain a temporal component that represents an averaged response of all directions over time, and an orientation component that represents its tuning preference. This temporal component obtained for each neuron, which could uncover its polarity preference (ON/OFF/ON-OFF), and kinetics preference (sustained/transient) to the bar, was concatenated with its corresponding responses to the chirp and the sparse noise stimuli. This concatenated response vector thus captures the functional responses of individual SC neurons. To estimate the functional diversity among the population of recorded SC neurons we used a Uniform Manifold Approximation and Projection (UMAP[118]) together with a gaussian mixture model (GMM). The response vectors were normalized using the fit_transform function from the UMAP toolbox (https://github.com/lmcinnes/umap) and projected into a two-dimensional UMAP (n_neighbors = 0.25% of N_cells). The gaussian mixture model (scikit-learn library) was applied to the resulting UMAP projection and the number of clusters systematically varied between 1 and 60. For each number of clusters, the Bayesian Information Criterion was estimated and the number that minimized the Bayesian Information Criterion was used as the optimal number of clusters in the diversity analysis.

**RGC axonal synaptic contact fields**
The high-density of recording sites enables the identification of the spatial location of the electrical signals of RGC axons on the probe and hence their anatomical location within SC. Importantly, the waveforms of RGC axons also contain the post-synaptic response of SC dendrites (Fig. 1e/g) which we used as a proxy for the anatomical location where the RGC axonal arbors make synaptic contacts with SC neurons. Since the Neuropixels probes is organized in four columns of electrodes, we could measure the spatial location both along and across the probe (Fig. 2a, bottom right). Thus, we defined the area on the probe that contains the post-synaptic response of the SC dendrites the axonal synaptic contact field (AF). To characterize the spatial position of the axonal synaptic contact field, we fitted a two-dimensional Gaussian function to the two-dimensional representation on the probe (Fig. 2a, bottom-right). This Gaussian fit was necessary because some of the RGC axonal contact fields were only partially covered by the recording sites on the probe, e.g. the example in Fig. 2a. To fit the axonal contact field, we assumed a constant sigma of the Gaussian functions and only optimized the x and y position of the Gaussian by least squares fitting (scipy.optimize.least_squares). The axonal contact field center position was estimated from the Gaussian fit and could be located close to the electrode border or even outside of the recording sites (Fig. 2a and S5).

**Receptive fields of RGC axons and SC/OT neurons**
The spatial receptive fields were estimated via spike-triggered averaging (STA) and by using the receptive field at lag −1 frame as the

corresponding onset receptive field[34]. Receptive fields were interpolated by a factor of two using a 2D-cubic-interpolation (scipy.interpolate.interp2d). Receptive fields were mapped with 5 deg, 10 deg and 15 deg sparse noise targets (Fig. S1c). Receptive fields mapped with 10/15 deg targets overestimate the receptive field size mapped with 5 deg targets (Fig. S1c) and therefore we scaled the threshold for the contour lines by a factor of 1.4 when plotting receptive fields mapped with 10 deg or 15 deg (the factor 1.4 was estimated from the data). The receptive field distance was calculated by the Euclidean distance between the receptive field centers. The receptive fields were measured at an estimated average position of +5.25 deg in elevation and +38.45 deg in the azimuthal plane from the nose position. However, due to the tangential insertion angle, the receptive fields covered a large area of the visual field. Within each mouse the receptive field coverage was on average 100 deg in the azimuthal axis and 88 deg in the elevation axis.

## Comparing the spatial organization of retinal ganglion cell receptive fields and axonal fields

To compare the spatial organization of the RGC receptive fields within the retina with the organization of the RGC axons within the SC/OT we estimated the receptive field (RF) and axonal field (AF) centers using the center-of-mass measurement (Fig. 2b, RF centers = circles, AF centers = crosses). The ensemble of receptive and axonal field center positions was subsequently used to characterize the spatial RGC organization with the following measurements. We calculated the Euclidean distance between the RF centers of RGC pairs (dRF) in the visual space (deg) and the distance between the AF centers (dAF) in the SC space (μm) (Fig. 2d). To test for hexagonality of the example shown in Fig. S5c we used the Delaunay and Voronoi tessellations[119] and estimated the angles of the Delaunay triangles (Fig. S5c). To directly compare the similarity between the receptive field and axonal field organization we first transformed the axonal field positions from the SC space (μm) into the visual space of the receptive fields (deg). This was achieved by linearly scaling and rotating the positions of the axonal fields such that the summed distances between receptive field and axonal field positions were minimized. Important to note: this alignment step ideally requires a population of RGCs to avoid underestimating the distances between RFs and AFs ($n > = 20$ mouse RGCs in this study) but otherwise does not change the geometric organization of the axonal field mosaic, it only scales and rotates their positions.

From these transformed axonal field positions, we calculated the distance between receptive and axonal field centers of individual RGCs. We divided the distance between the receptive and axonal field centers by the mosaic spacing factor, which was estimated as the median receptive field distances between nearest RGC neighbors. A mosaic spacing of 1 is the distance between two neighboring RGCs and a value of 0 when the centers are overlapped (Fig. 2e). To assess the similarity in the geometrically organization of the receptive fields and axonal fields we calculated the Euclidean distances between receptive fields (dRF) and axonal fields (dAF) of pairs of RGCs (Fig. 2f) for RGC with high signal-to-noise receptive fields (SNR > 10) and good fit of the axonal fields (R2 > 0.8). The enclosed angle within triplets of RGCs (Fig. 2g) was calculated for RGCs belonging to the local neighborhood (within a radius of 5 x receptive field diameters). This analysis was performed for RGCs with similar response properties (Fig. 2) and independent of their functional type (Fig. S5). Functional similarity was assessed by comparing the evoked chirp responses and the receptive field properties (ON or OFF) to light and dark sparse noise.

## In vivo connectivity analysis

Monosynaptic connections between RGC axons and SC neurons were detected using established methods[28,33,43,48] on the jitter corrected cross-correlograms (CCGs) (Fig. S7a–c) based on statistically significant peaks at synaptic delays (+0.5 to 3.5 ms, purple) above the

baseline (Fig. S7a, −3.5 to 0 ms, green). CCG peaks had to extend over the threshold (baseline + 4 x standard deviation) for at least 5 consecutive time bins (0.1 ms resolution). The cross correlations were calculated using the pycorrelate package (https://github.com/tritemio/pycorrelate). Spike times over the entire recording were used in the CCG analysis to avoid biases inherited from a particular tuning following the exposure of a particular protocol. The jitter correction was required to remove stimulus-evoked common input. To estimate the jitter correction, we followed established approaches[48,120]. Briefly, we calculated a jittered version of each spike train by randomizing all spike times within consecutive 10–15 ms windows[120]. We then calculated the cross-correlation between a pair of neurons both for the original (raw CCG) and the jittered spike train (jittered CCG). Subtracting the jittered CCG from the raw CCG resulted in a jitter-corrected CCG (Fig. S7a, compare left and right).

## Efficacy and contribution analysis of connected RGC-SC/OT pairs

Synaptic efficacy and contribution were estimated from spikes during the entire recording duration following established approaches[33,44,121,122]. Briefly, efficacy was estimated from the jitter corrected CCGs by dividing the area of the CCG peak (peak window: 0.5 to 3.5 ms, Fig. S7a, purple) by the total number of presynaptic spikes. Thus, an efficacy measure of 1 (100%) would reflect that for each presynaptic spike, a postsynaptic spike could be detected. To estimate the contribution, we counted the number of SC spikes that were preceded by spikes from individual retinal afferents, in a time window between −3 to −0.5 ms, and divided this number by the total number of SC spikes. A contribution of 1 (100%) indicates that all spikes of an SC neuron are preceded by spikes from individual RGC afferents. To characterize the diversity of efficacy and contribution values of divergent connections, i.e. between individual RGC and multiple SC neurons, we restricted the analysis to divergent connections in which we identified at least three postsynaptic partners for individual RGCs. To characterize whether individual RGCs make multiple strong connections we further restricted the analysis to divergent connections for at least one strong connection was found (efficacy > 10% and contribution >30% for the analysis shown in Fig. 4f).

## Functional similarity analysis of connected RGC-SC/OT pairs

**Functional similarity index.** To characterize the functional similarity between connected RGC-SC/OT pairs we estimated the average of the correlation coefficients between the trial averaged visually evoked activity of during the dark and light sparse noise ($r_{SD}$ and $r_{SL}$), chirp ($r_{chirp}$) and moving bars ($r_{m. bar}$); functional similarity index = ($r_{SD}$ + $r_{SL}$ + $r_{chirp}$ + $r_{m. bar}$)/4. The similarity of the sparse noise responses ($r_{SD}$ and $r_{SL}$) was estimated from the spatiotemporal receptive fields which were calculated using the STA. The similarity of the chirp responses ($r_{chirp}$) was calculated from the chirp PSTHs. The similarity of the responses to the moving bars were like-wise estimated from the moving bar PSTHs. Only RGC axons and SC/OT neurons with high signal-to-noise in the visually evoked activity were included in this analysis (SNR > 10 for the receptive fields).

**Direction and orientation tuning.** To determine orientation tuning, we quantified the responses to moving bars as the maximum response of the PSTH in each exposed direction. The obtained tuning curve (Figs. 5a and S1c) was interpolated on 30 points and then fitted with von Mises functions[34] using the least square optimization function from SciPy. The von Mises function is a circular normal distribution and the sum of two von Mises functions allows fitting direction and orientation tuning curves and extracting preferred direction (PD) or orientation (PO)[123]. From this fit we extracted the firing rate at the preferred direction ($FR_{PD}$), the firing rate at the opposite direction ($FR_{180}$) and orthogonal direction ($FR_{90}$) to calculate the direction selectivity index (DSI = [$FR_{PD}$ − $FR_{180}$] / [$FR_{PD}$ + $FR_{180}$]) and orientation

selectivity index (OSI = [FR$_{PD}$−FR$_{90}$] / [FR$_{PD}$ + FR$_{90}$]). Only RGC axons and SC/OT neurons with high signal-to-noise in the visually evoked activity in responses to the moving bars (SNR > 2) and good R2 values for the fits (R2 > 0.8) were included in this analysis. Neurons with DSI > 0.2 were included in the pool of direction selective neurons and neurons with OSI > 0.2 and DSI < 0.2 in the pool of orientation selective neurons.

## Reporting summary

Further information on research design is available in the Nature Research Reporting Summary linked to this article.

## Data availability

The data shown in the figures are provided in the Source Data file. A minimum dataset for illustrating RGC axons in Neuropixels recordings in mouse SC has been deposited in a repository on Zenodo[124]. The raw Neuropixels datasets generated in this study are too large to made available online and are available upon request from the correspondence author (jens.kremkow@charite.de). Source data are provided with this paper.

## Code availability

The python code for the semi-online receptive field analysis and RGC axons identification is provided in a Git-Hub repository[117]. Analysis code is available upon reasonable request from the correspondence author (jens.kremkow@charite.de).

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

## Acknowledgements

We thank J-M. Alonso, J. Poulet, B. Judkewitz for helpful discussions and materials exchanges during the project; P. Wisinski-Bokiniec, C.J. Whitmire and J-M. Alonso for comments on the manuscript; T. Leva, and P. Schnepel for help with the Neuropixels recordings; Fabian Heim for zebra finch histology; J. Siegle and D. Denman for help with software

(OpenEphys) and setup; and J. Colonell for Neuropixels hardware explanations. We thank the Neuropixels community for their equipment and support and the Allen Institute for Brain Science for fostering high quality databases. The mouse head schema used in Figs. 1c, S1b, S2d, S7h was adapted from https://doi.org/10.5281/zenodo.3925903. The mouse and zebra finch drawings were provided by Adele Costalunga and the zebra finch sagittal schema by Giacomo Costalunga. This work was supported by the DFG Emmy-Noether grants KR 4062/4–1 (JK) and VA 742/2 (DV), the ERC-2017-StG - 757459 MIDNIGHT (DV) and Project number 327654276 – SFB 1315 (DV).

## Author contributions

J.S. and J.K. conceived and designed the study; J.S., C.G., J.B., D.V. collected the data; J.S., C.G., H.B., T.L., K.-L.T. and J.K. analyzed the data and J.S., D.V. and J.K. wrote the manuscript with inputs from all authors.

## Funding

## Competing interests

The authors declare no competing interests.
