## [Peer Review File · Nature Communications]

High-density electrode recordings reveal strong and specific connections between retinal ganglion cells and midbrain neuronsREVIEWER COMMENTS

Reviewer #1 (Remarks to the Author):

Summary

The manuscript of Sibille et al. makes creative use of Neuropixels probes to simultaneously record from a large population of retinal ganglion cells and neurons in the superior colliculus. Using a strategic angle of insertion of high-density, high channel count silicon probes they can simultaneously record from neurons in the superior colliculus and their retinal inputs. Using this large set of paired recordings, the authors claim that the retinal ganglion cells make strong and specific connections to their targets in the superior colliculus. The experiments are performed predominantly in mice, with a nice demonstration of the technique's flexibility by performing analogous recordings in zebra finches. The experiments and analysis are impressive and exhibit some of the promising results one could achieve using this technique. For, example detail how the axons of retinal mosaics are distributed in the super colliculus. However, the claims made by the authors about the nature of connectivity between retinal ganglion cells and their targets in the superior colliculus are not adequately supported by the presented data. This is predominantly a consequence of the mismatch between the anatomy of the superior colliculus and the recording technique, which leads to three important weaknesses:

1. The full constellation of inputs to any one neuron is very unlikely to be recorded using this technique as the dendritic trees are often large and run perpendicular to the insertion of the probe.
2. The recording technique appears to be undersampling the different neuronal types of retino-recipient neurons in the superior colliculus. There appears to be an oversampling of neurons that have cell bodies lying near the optic layers, and an undersampling of more superficial neurons.
3. The recording technique appears to be undersampling the inputs from the retina. Of the possible ~30 cell-types innervating the superior colliculus, only 5 response types are shown. This corresponds with the known depth distribution of where retinal axons of different types terminate within the superior colliculus.

Below we provide a more detailed critique of these issues.

Major issues

1. There is a fundamental mismatch between the recording technique and anatomy of the superior colliculus. While the recording technique samples from a single depth (at least locally) of the superior colliculus (estimated to be near the optic layer), both, the dendrites of neurons in the superior colliculus and innervating retinal ganglion cells, are organized across depths. In mice, axons of different retinal ganglion cell types stratify at different depths of the superior colliculus starting at its surface down to the depth of the optic layers. In addition, the dendrites of individual neurons in the superior colliculus are often quite extensive, extending across many depths. For example, wide-field and horizontal neurons have particularly large dendritic trees. It is not clear how local extracellular recordings from the optic layer (or any single depth) of the superior colliculus can adequately sample both the action potentials of a single neuron and its retinal inputs. How did the authors determine they could see both distal and proximal inputs to a single neuron? What evidence do the authors have that they are regularly sampling a large majority of the inputs to any single neuron they are recording from? We find it unlikely to be the case and request additional evidence to demonstrate that for an individual collicular neuron an adequate distribution of retinal ganglion cell inputs can be recorded to support the claims of the paper.
2. A catalogue of the depths of recordings is not adequately reported. It remains unclear at which depth of the superior colliculus Neuropixels probes penetrated relative to internal landmarks. This information is indispensable in estimating what cell-types were likely sampled (both collicular and retinal). We recommend showing an analysis of the location (in particular depth) in relation to known anatomical landmarks (both mouse and zebra finch). For example, using eye injections of CTB to see where the probe track lies in relation to the optic layers. Other solutions are of course possible.
3. The population of collicular neurons appears undersampled. We request an analysis of which collicular cell-types are being sampled (in particular across depth) to demonstrate that enough cell-types are being recorded to support the broad claims of the paper. Or identify reduced population and narrow the claims made by the paper.

4. The population of retinal ganglion cell types appears undersampled. We request an analysis of which retinal cell-types are being sampled (in particular across depth) to demonstrate that enough cell-types are being sampled to support the broad claims of the paper.
5. We have concerns that the method of how RGC-SC pairs are determined might create a selection bias that leads to the conclusion that retinal ganglion cells in general strongly drive the downstream partners in the colliculus. One potential issue is that connection efficacy and contribution were determined by inspecting the spiking activity in response to a full-field “chirp” stimulus which induces strong stimulus correlations in retinal ganglion cells of the same type. A stimulus that induced sparse responses in both the retinal ganglion cells and collicular populations might be more appropriate.
6. Isometric mapping of retinal ganglion cell mosaics appears to have been determined for just one putative cell type, transient Off cells. Either demonstrate that this generalized to a larger set of retinal cell types, or make clear that this claim can only be made for this cell type.
7. It is not clear how many retinal ganglion cell axon terminals and collicular neurons the analyzed 1048 RGC-SC pairs consist of. It is important to know whether Figures 4F and 5G contain all recorded RGCs and SC neurons, respectively, and if not, which exclusion criteria were applied.
8. A number of important references are missing, in particular regarding cell types of mouse superior colliculus and how the superior colliculus is innervated by different retinal ganglion cell types. While some of this information is referred to in reviews cited, the paper appears to skip over important anatomical details of the how different cell types (retinal and collicular) are distributed across the depth of the super colliculus. For examples paper by labs of Gabe Murphy, Joshua Sanes and Marta Bickford. In addition, a couple of important papers that directly demonstrate how retinal ganglion cells innervate either the thalamus or superior colliculus are also missing. For example, from the labs of Botond Roska and JC Cang.

Minor issues

1. More details about the coordinates (AP, ML), angle and depth of probe insertion would be helpful for the SC/OT community.
2. The picture of RGC axon terminals targeting SC somas is misleading. Retinal ganglion cell terminals will rather target the dendrites at different depths, including far from the probe.
3. Line 104: In the mouse retina, gap junctions are also present between retinal ganglion cells of different types (Cooler & Schwartz, 2020), hence, it is not valid to include gap junctions into the criterion of a single type based on cat data.
4. When referring to the retinothalamic connectivity please be explicit about species. On line 273 most papers were from experiments in cats and monkeys, not mice or zebra finches.
5. Line. 146: there are enough papers that demonstrate that a high-resolution map does not necessarily require a one-on-one mapping. Also, it is not clear why this would be the case in the superior colliculus.
6. Line. 320: Not clear whether Figure 4F includes all previously determined connections.
7. Line. 823: Only direction tuning was analyzed in this paper but the section in the Methods refers to orientation tuning. Was orientation tuning analyzed as well? What was the similarity between collicular neurons and ganglion cell afferent inputs in orientation tuning?

Reviewer #2 (Remarks to the Author):

Summary

The authors devise a novel approach to simultaneously record from retinal axons and postsynaptic neurons in the superior colliculus of mice and the optic tectum of zebra finches. They provide compelling evidence for the validity of their approach. The authors then use this novel technique to show that the retinotopy of retinal axonal projections in superior colliculus is preserved at a high level of precision. Moreover, they use the high temporal resolution of their technique to identify directly connected pairs of retinal axons and collicular/tectal neurons to quantify connection strength and functional similarity between retinal inputs and collicular/tectal neurons.

Major comments

1. The recorded data are of very high quality, and the claims that some axonal potentials could be identified as from retinal axons while others from collicular neurons are very well substantiated.
2. The authors claim that the mosaic organisation in the retina was preserved at the level of retinal ganglion cell (RGC) axons in superior colliculus (SC), however the evidence that they measured retinal mosaics is very limited. (A) The visual stimulation used to classify RGC types was reduced to a subset of stimuli that was used to distinguish the currently accepted number of RGC types (see Ref. 1: Baden et al., Nature, 2016). Consequently, the authors may have pooled several RGC types into the same functional mosaic. Regarding the RGC classification, it is not clear how this was achieved. Please, add more details. (B) The overlap of receptive fields (RFs) in the mosaic in Fig 2C seems very high, higher than the often-observed coverage factor of 2 (see Ref. 1), which would speak for a pooling of several RGC types. (C) The Neuropixels probes are very long but very thin (maximum channel distance across probe width is 75 μm). The possible coverage of 2D visual space by RFs of simultaneously recorded neurons or axons is therefore very limited. This also limits the possibilities to quantify retinal mosaics (see Major point 4(A) for a related comment). (D) The authors only show quantification of the retinal mosaics for a single RGC type (OFF cells for mice, ON-OFF cells for birds). How do these results compare to other RGC types? Do they generalize? Are the numbers of recorded RGC types in agreement with previously reported frequencies?
Despite these issues regarding retinal mosaics, we feel that the main result of the paper, namely that retinotopy is preserved in retinal axons, still holds. In fact, the quantifications in Fig 2F-I (and Fig S7H-K) showing that RFs and axonal positions in the brain match very well would be even stronger if all simultaneously recorded retinal axons were included instead of only the axons from a single RGC type.
We think that if the evidence for retinal mosaics cannot be strengthened by the authors, the title of the paper should be changed accordingly.
3. Given that the authors distinguish different RGC types, it would be very interesting to see whether they find differences between those types, e.g. in terms of efficacy and contribution, also in terms of numbers (do recorded numbers match those reported in the literature?).
4. Several results are presented without testing them for significance or comparing them to a null hypothesis:
 - (A) How do the results in Fig 2D+E (distances and angles between RFs and AFs) of a single RGC type compare to results when a comparable number of axons is randomly sampled from all simultaneously recorded axons? Are these significantly different?
 - (B) Are the log normal distributions in Fig 4C+D and Fig S8B significant?
 - (C) Are the distributions of efficacy and contribution for 1st and 2nd strongest connections (Fig 4F) direct consequences of the log-normal distributions (Fig 4C+D)? By sampling pairs of efficacies/contributions randomly from the log-normal distributions, the authors could compare the resulting "surrogate" distributions to the measured ones. Are they different?
5. The authors state that log-normal distributions for efficacy and contribution are commonly found in many brain areas and species. This would mean that the retino-collicular connections are not special in any way. This does not come across when the authors then state that "the retinocollicular circuitry is optimally wired for transmitting retinal activity in a functional specific manner", unless the authors think that all circuits (at least those with log-normal distributions of efficacy and contribution) are optimally wired for functionally specific transmission. Please, clarify.
6. Related to the ubiquity of the log-normal distributions, what is the significance for finding these distributions in mice and zebra finches? This finding then does not speak for an evolutionarily conserved circuit because most likely also non-conserved circuits show this distribution.
7. The authors collected unique datasets quantifying connection strengths between retinal axons and SC neurons on one hand and functional connection motifs (relay or combination) on the other hand. It would be very interesting if the authors showed how these are related to each other. E.g. do relay motifs have stronger connections than combination motifs? Related to this, the authors use various similarity measures (correlation of spatiotemporal RFs, difference between preferred directions, correlation of responses to chirp stimuli) to quantify function similarity (Fig 5). It would be preferable if they combined all of them to determine a single similarity measure and compare this to efficacy/contribution and to determine whether motifs are relay or combination.
Coming back to the different RGC types, are there differences in efficacy/contribution between RGC

types?

8. What is the evidence for 2 distinct groups of relay and combination motifs? The scatter plot in Fig 5G seems to show a continuous range of motifs.

9. Although it is very interesting that the reported results are similar for mice and zebra finches, it seems unfounded to us to generalize these findings to mammals and birds and even to all vertebrates (lines 405-7). Given that only about 10% of RGCs in the primate project project to the SC, while 80-90% of RGCs in the mouse project project to the SC, there may be substantial wiring differences even across mammalian species.

10. The authors state that their novel approach “opens up opportunities to investigate the principles of how afferent inputs organize in other parts of the brain” (l. 445). This statement may need a more cautious formulation as no attempt was made to show the feasibility of the approach in other brain areas. One argument against a generalization would be that the authors did not find any axons from other areas projecting to the superficial SC (for example from V1).

11. The authors suggest that their new approach is particularly suited to investigate functional maps of synaptic inputs. It seems to us that imaging approaches, e.g. two-photon imaging of axon/synapses, are superior as they provide a far better coverage of a 2D plane while Neuropixels probes provide a very limited sampling of brain space (max distance of channels across width of the probe is 75 μ m). The far greater advantage of the presented approach seems to be its excellent temporal resolution and the ability to detect direct connections. The authors may wish to highlight this instead.

Minor comments

- L. 7: “strong connections and limited functional convergence” does not reflect the results. The authors found mostly weak and only few strong connections (Fig 4C,D), and a range of connection motifs from functionally similar to dissimilar (Fig 5G). This statement is repeated in l. 490.
- Introduction: the authors should cite their Ref. 50 here as it is one of the first attempts to investigate the functional connectivity between RGCs and SC neurons in mouse, which is one of the major topics of the present paper.
- L. 108: how many animals were used? Please also add this information wherever appropriate.
- How many axons and neurons were recorded simultaneously on average across sessions?
- Report where RFs were measured in each mouse (elevation, azimuth).
- L. 141: how were borders of SC determined?
- L. 190: Reference S7C is probably wrong
- Fig 2A: The term “dendritic RF” is very confusing as it suggests that there is also an axonal RF.
- Fig 2B: caption says that RGCs were identified using a chirp stimulus but the cartoon (and responses) only show ON/OFF stimuli.
- Fig 2D+E: specify what black vs gray bars show
- Fig 2G (and Fig S7I): please report RF-AF distance in visual degrees; meaning of symbols above the histogram is unclear
- Line 295: Forgot the % sign
- Fig 4B: “Spike” would be more suitable for y-label (instead of “Trial”)
- Fig 4E: state what the lines mean (CCGs presumably)
- L. 337: Ref. 50 used intracellular recordings to show that direction preferences of retinal inputs and the connected SC neuron are similar. Why is confirmation by monosynaptically connected pairs pending?
- Fig 5E,F,G: what similarity measure was used in histograms and scatter plot?
- L. 396: “zebra finch OT neurons receive a limited pool of RGC afferents”. Unclear what the authors want to express here as the pool must be limited rather than infinite.
- Fig 6D+G: it would be more appropriate to show actual data rather than only cartoons/fits in this main figure. The fits can be plotted on top of histograms/the scatterplot.
- Fig 6F: to show that there is little scatter in retinal input one needs to see the outlines of single retinal RFs compared to the outline of the RF of the postsynaptic neuron. Also, it is difficult to compare a contour line to a RF.
- L. 495: Liang et al (Cell, 2018, A Fine-Scale Functional Logic to Convergence from Retina to Thalamus.) shows a different picture and should be mentioned here. Another important paper on thalamo-cortical connections is da Costa et al (J Neurosci, 2011, How thalamus connects to spiny

stellate cells in the cat's visual cortex)

- L. 610: what is the insertion point in AP? Are the reported ranges the different insertion points of the coverage of the probe within the brain? For reproducibility, insertion point is more important. Also, it is unclear what is meant by the angle. As the probe can be rotated and tilted in 3D, please specify in which plane the angle is measured and what is 0 degrees?
- L. 641: what's the length and width of the bars?
- L. 642: provide details about the chirp stimulus (starting and ending frequency, speed of modulation, ...)
- L.642: "The timings..." " This statement is unclear. What synchronizing signals? Marked where?
- L.646: what was the extent of the removal of visual cortex? Are there histological records? We suggest to mention the cortical removal in the main text as it is a major influencing factor.
- L.655: under which conditions were the experiments deemed as not successful?
- L.705: "The window was interpolated (101 times)". Please clarify. How is the window defined (space and time)? What does 101 times interpolating mean? In space and time? Was smoothing done in space also?
- L.707: "All slope measurements...". Unclear
- L.709: Fig S2B is probably the wrong reference
- L.755: how is synaptic contact field defined?
- L.791: do the CCG results depend on the stimulation protocol? Can it be measured during spontaneous activity to exclude possible drive by visual stimuli?
- L.805-808: Clarify how contribution was determined: from retinal spikes of a single axon or from retinal spikes of all recorded axons to a specific neuron? If the latter, is it somehow normalised to the number of detected synapses?
- L.814: lag of -1 ms or -1 frame?
- Fig S1: Please use fewer abbreviations. The text is currently very hard to understand.
- Fig S1G: Left and middle panel look very different. Are they not the same example? What is meant by "shank"?
- Fig S1J: Right panel not clear. Are these sagittal sections? If so, what are the ML coordinates of each section? What are we supposed to see? Please adapt brain atlas images to reflect histological images.
- Fig S1K: The green frame in Fig S1D shows channels in PrA. Where are the channels here? In the opposite SC?
- Fig S2A: Would considering more PCs improve classification?
- Fig S3A+C: under what conditions/stimulation were firing rates measured? Does it make a difference? What is spontaneous rate?
- Fig S3B+D: how are these measures defined?
- Fig S4C: What was the stimulation protocol?
- L.944: should relate to panel (F). What is the correlation between non-coupled RGCs for comparison?
- Fig S7: Please check spelling and references to various panels (top, left, ...)
- Fig S7B: The left slice looks very different from the right slice. Is this really the correct match in the brain atlas? Please mark OT.
- Fig S7D: What is the reason for the gap in the RFs?
- Fig S8C: similarity measurements are not reported for these examples
- State more clearly that most of the results were collected in anaesthetized mice, only Fig S1N,O are from awake mice

Writing style (suggestions):

- Some paragraphs start with a conclusion of the previous paragraph. Other paragraphs start with a statement on what this paragraph is about to show. The authors may wish to stick to one style, preferably the latter.
- "Paired recordings" in the title is misleading as only a single probe in one brain area is used for simultaneous axonal and neuronal recordings, which is the great advantage of this new approach.

Conclusion

Sibille et al. present a novel recording technique that can be used in vivo, and is highly useful to the

community, enabling to record synaptic input and the postsynaptic neuronal response of the retinocollicular circuit simultaneously. Using this technique, they shed new insights on a long-standing question: What is the connection pattern between RGCs and the SC/OT? While some of the claims need to be further substantiated, and more clarification is needed in parts of the work, the work itself is impressive and adds both to the visual neuroscience field, but also to neuroscience in general due to the novel technique. Accordingly, we highly recommend publishing this work once the issues we raised are addressed.

Reviewer #3 (Remarks to the Author):

Comments to the authors

Manuscript number: NCOMMS-21-38786-T

Title: Strong and specific connections between retinal axon mosaics and midbrain neurons revealed by large scale paired recordings

This is an interesting study investigating how the activity of retinal axons is paired with the activity of their target neurons in the superficial layers of the superior colliculus of mice and optic tectum in zebra finches. The results are based on extracellular recordings of single unit activities using pixel probes with many channels. Using thus a very modern electrophysiological approach several experiments have been performed in anaesthetised and awake animals. Many valuable findings are presented:

1. A method is presented on how of axonal waveforms, which are coming from the retinal ganglion cells (RGCs) can be separated from the activity of superior colliculus neurons (SC) using extracellular recording. This section includes additional validation experiments with pharmacological treatments.
2. Retinotopic organization of visual inputs in the superior colliculus is confirmed.
3. Monosynaptic connectivity between RGC axons and SC neurons is validated
4. How SC neurons integrate the inputs from RGC axons is investigated
5. The representation of spatiotemporal receptive fields in SC are investigated
6. The mammalian RGC-SC circuit is compared with the birds optic tectum recorded in zebra finches using a similar approach.

Overall, I am truly impressed about the amount of work that the authors have done and I am thrilled by the abilities of the authors to use different types of sophisticated analysis of a very large dataset. However, I also have some serious concerns on the manuscript which need to be addressed. Given that the results contain several types of valuable information my recommendation is a major revision. However, the manuscript should be really revised and partly rewritten and not be published in its current state.

My major concern is that the paper lacks to define a main research question. Many experiments have been put together according to the principle "more is more".

I believe that the presented results can be separated in at least three sophisticated research papers. This would allow to describe properly what has been done in a way that a broader readership would understand. This would also allow to address each research question separately, to present the findings accordingly and to discuss all crucial details of these findings in light of existent literature.

At present many details are not explained and require a lot of thinking and scrolling back and forth through the manuscript. The discussion of many aspects is short and only superficial. Please don't get me wrong, I truly believe that everything that is presented in this manuscript is logical and that everything makes sense. However, I find that the experiments are not presented efficiently.

This is already apparent after a brief look at the figures (there are overall 6 Figures, which contains up to 7 subfigures in the main manuscript. Furthermore, almost each sub-figure is divided in 2-3 additional sub-sub figures. Further 8 Figures with sub- and sub-sub-figures are in the supplement). This is an enormous amount of information. At the same time the results and methods are not explained and discussed to the needed extend in light of the already existent literature. The methods need to be reproducible. This is not given at present.

One suggestion could be to focus this manuscript mainly on the question how SC neurons integrate RGC inputs. In my view this is the most novel and most interesting part of the study. Two very interesting hypothesis are proposed on how superior colliculus neurons could integrate retinal ganglion cell inputs (Lines 21-25, summarised in Figure 1B). The findings need to be discussed in light of these hypothesis. All the rest of the manuscript should be constructed around this major question only.

In order to address this question, it was of course needed first to validate methodologically that the activity in the retinal ganglia cell axons can indeed be separated from the activity of superior colliculus neurons. This can be itself either a purely methodological paper, that needs to be published first, or it can remain in the current paper as "experiment one". However, it needs to be discussed very clearly and in light of existent literature to what degree such a separation of waveforms based on extracellular recordings, without any morphological validation can be used to undoubtedly identify RGC axonal responses. An alternative interpretation would be that such separated waveforms are coming from axonal activity of other SC internal neurons. They all would be visual and may respond faster than other SC neurons. This interpretation needs to be excluded. At the moment I am not fully convinced that the used approach is reliable. The pharmacological treatment for validation is also not very convenient to me. The pharmacological effects are clear, but the interpretation is vague. The authors are welcome for a rebuttal :)! Explain please all your arguments against my interpretation in the discussion section of your revised manuscript.

The section investigating the monosynaptic connectivity between RGC axons and SC neurons should also remain, because it's an additional part dealing with connectivity of RGC axons and SC neurons. Thus it fits to the main story line.

The remaining parts of the manuscript should be left out of the present manuscript. They all can become other more valuable papers. In the present manuscript there is not enough space for presenting and discussing all the findings. Presenting them only superficially as they are present for now, is not a good solution in my view.

For instance, the finding of retinotopic organisation in optic tectum is nothing novel per se. This has been demonstrated using optic imaging of intrinsic signals even in zebra finches (Keary et al., 2010, PlosOne). I agree of course that the electrophysiological validation is needed. However, this is not so important for a high impact manuscript. Keep these results for another solid paper in a decent journal, where all aspects and details would be discussed.

The visual field analysis, directional tuning etc. can be also left out of this manuscript (btw. what about orientation selectivity?). At present this is a very superficial presentation of the findings. This part has a lot of potential in particular for a comparative study of mice and zebra finches. I am a big fan of comparative study of brain functions and evolution of visual processing. I believe that your data has a lot of potential for comparing between zebra finches and mice in proper manuscript addressing only this issue. Here you could also consider, that there are some substantial differences in the organisation of the retinas and optic tecta (e.g. in finches are more layers in the tectum compared to mice, finches have more photoreceptors etc.). Thus, some differences in the activity in

the optic tectum in these two vertebrate models should be extractable from your data. Take a look at your data considering this and make a great separate paper out of the data in the end.

Minor comments:

Abstract: please don't use abbreviations in the abstract.

Overall: please reduce the amount of abbreviation to a minimum. There is already so much information in the result section, don't make it harder for the reader by adding additional abstraction level coded in abbreviations.

Introduction: there are too many aims. Remove paragraph two and specify the main aim in end of the manuscript

Results:

My suggestion as already mentioned above would be to remove all sections and leave only three following the order:

1. Recording afferent axons and local neurons simultaneously using high-density electrodes.
2. Synaptic organization of the retinocollicular circuit in vivo.
3. Measuring monosynaptic connectivity in vivo at a large scale

However, in any case, since the results are following the introduction, it should be made sure that the reader can understand the basic methodological approach without reading the methods first. A simple claim "see methods" is thus of little use for the reader here. A methodological figure, showing how the stimuli were presented and what kind of stimuli were used would be helpful. All needed details that would allow to understand the results should be provided. This would make the manuscript better accessible for a larger public.

Overall to many graphs, and even more are in supplement as I already mentioned above. Moreover, some sub-sub figures are very small. See e.g. figure 1B, 1C or figure 5A. I am glad that I have a PDF and can zoom in on my computer monitor. I would not be able to see anything in a printed version. If you have less results sections, you would have more space for larger images.

Line 328: Please explain (or show in a figure) what kind of a sparse noise stimulus was used. I don't want to read the Paper "15" to extract this information which would allow me to understand your paper.

Line 348: What is a chirp stimulus? Please explain.

Discussion:

Is the methodological validation your main finding? Then it should be a methodological paper. But then it would not be a suitable paper for nature communications.

I would not put this part in front of the discussion and I would also not limit the discussion to only advertise your method so much here (Btw. pixel probes are commercially available, at least this part is not so novel). Instead I would suggest to discuss properly the validity of the method for measuring "axonal synaptic contact fields" in your extra cellular recording approach. I am not sure though, if such a conclusion can be made at all without a morphological validation study using e.g. calcium imaging and viral tracing. But you can try to convince also readers like me with a proper discussion.

Methods:

It should be clarified why 95 mice were needed but only 7 zebra finches. I still don't fully understand which setup was used for which experiments. For which of the experiments awake animals were needed and how many. Was the same setup used for presenting visual stimuli to zebra finches and mice?

I think the methods should be organised in a more efficient way, presenting each experiment independently in a concise and clear way.

Lines 570-571: I suppose this is an analgetic? Please add this information

Lines 580-589: "Recordings..." this part should go in the part "Electrophysiological recordings" starting from Line 601

Lines 589-599: "Histology..." this part should be after the pharmacological application section before data analysis.

Lines 626-643: "Visual stimulation" this whole section needs to be overworked. Crucial details are missing. Was the same setup used for anaesthetised zebra finches and mice? What does it mean either a calibrated screen or projector? For which of the experiments did you use a screen and for which a projector? You need to be more specific. It is not clear to me what kind of stimulation was presented for which species and under which conditions. A figure of the setup/setups including images of the used visual stimuli would be very useful. Please keep in mind that the crucial parts of the experiments have to be reproducible based on the information provided in the methods section.

Line 656: What do you mean by $(n=3/6)$? Is it 3 or 6?

Line 714-721: The logic of this approach for detection of axonal efferents needs to be explained better.

Line 746-760: This sounds really fascinating and I am really trying hard to understand how it is possible to separate signals coming from RGC axons from those of SC neurons. Are you sure that these are RGC and SC neurons without any morphological confirmation? I don't doubt that there is a reasonable logic behind this approach. However, this part needs to be described in a way, that also other people can understand.

Lines 762-786: this section is very hard to read because too many abbreviations were used. I would suggest in general to avoid abbreviation whenever it is possible through the whole manuscript. It is possible to write axonal field instead of AF and receptive field instead of RF etc. Your paper will become more readable.

Lines 810-822: I think that this "Receptive fields" section should be better placed before "...retinal ganglion cells mosaics..." section in line 761. Moreover,... (you already probably know what I will say now :))... also this section needs a better explanation to make it understandable for more general public and to be reproducible.

Lines 826-827: What is a "Mises function" ?

Supplements:

Figure S7B: consider that you penetrated several layers of optic tecta in zebra finches. While the outer layers are retinotopically organized, the deeper layers, especially the output layers should be less retinotopic. Instead, several types of functionally separated units should be more abundant in the deeper layers.

We are grateful for the comments from the reviewers, which helped us to strengthen our analyses and communicate more effectively the details of our findings and its significance. To address their criticisms, we have performed new analyses and revised the text and figures extensively. We believe the manuscript was significantly improved.

Because the manuscript has been extensively re-written, including new figures and new discussion and methods sections we have not highlighted individual changes in the manuscript. We do provide the line numbers and important parts of the text that were changed, here in this letter. Below, we address the specific comments from the reviewers in detail.

Reviewer #1 (Remarks to the Author):

Summary

The manuscript of Sibille et al. makes creative use of Neuropixels probes to simultaneously record from a large population of retinal ganglion cells and neurons in the superior colliculus. Using a strategic angle of insertion of high-density, high channel count silicon probes they can simultaneously record from neurons in the superior colliculus and their retinal inputs. Using this large set of paired recordings, the authors claim that the retinal ganglion cells make strong and specific connections to their targets in the superior colliculus. The experiments are performed predominantly in mice, with a nice demonstration of the technique’s flexibility by performing analogous recordings in zebra finches.

The experiments and analysis are impressive and exhibit some of the promising results one could achieve using this technique. For, example, detail how the axons of retinal mosaics are distributed in the super colliculus. However, the claims made by the authors about the nature of connectivity between retinal ganglion cells and their targets in the superior colliculus are not adequately supported by the presented data. This is predominantly a consequence of the mismatch between the anatomy of the superior colliculus and the recording technique, which leads to three important weaknesses:

We are excited about the overall positive assessment of reviewer #1. We are grateful for the detailed and valuable comments based on which we refined the analysis, rewrote the manuscript and updated figures. We hope that our changes adequately address the concerns of reviewer #1.

1. The full constellation of inputs to any one neuron is very unlikely to be recorded using this technique as the dendritic trees are often large and run perpendicular to the insertion of the probe.

We thank the reviewer for bringing up this important question and we acknowledge the validity of this critique. We did not want to give the impression that we are able to capture the full constellation of all inputs to a given SC neuron. We apologize if we conveyed this message. Our method allows the characterization of functional synaptic connections between RGC axons and SC neurons in vivo by means of cross-correlation analysis. This method cannot capture the full constellation of all inputs nor the anatomical location of these inputs to a given SC neuron. However, this method does provide a unique and novel characterization on how SC neurons integrate RGC inputs in vivo. Direct functional connections between RGC and SC neurons in vivo were not reported before and identifying divergent and convergent connections between RGCs and SC neurons is, even without capturing all connections, a significant methodological achievement. In order to improve our manuscript, we have modified the text in the discussion of the revised manuscript as follows:

Line 501: “Despite this high yield in identifying connected pairs, the method cannot capture the full constellation of inputs to individual neurons nor unambiguously reveal whether a connection is located on the proximal or distal part of the postsynaptic dendrites (the dendritic arbors of certain cell types can span several hundred μm^{60}).”

2. The recording technique appears to be undersampling the different neuronal types of retino-recipient neurons
in the superior colliculus. There appears to be an oversampling of neurons that have cell bodies lying near the optic
layers, and an undersampling of more superficial neurons.

We thank the reviewer for raising this point. In the revised manuscript we now include a detailed analysis on the location
of the neurons and axons within the SC. To do so we used the method “SHARP-track^{1,2}” that allows reconstructing the
location of the Neuropixels recording sites within the Allen Mouse Brain Common Coordinate Framework, based on
histology and physiological landmarks.

This new analysis revealed that the majority of waveforms from RGC axons (94%) and SC neurons (92%) are located
in the optical layer as well as in the superficial gray layer (see new Fig. S1l-m), as expected from anatomy. This analysis
further showed that our dataset contains ~1.7 times more RGC axons in the optical layer (n = 641 RGC axons)
compared to the superficial gray layer (n = 361 RGC axons), as predicted by the reviewer. However, the number of
RGC axons and SC neurons in the superficial gray layer is still high and therefore we think that our dataset includes
the superficial gray layers in a representative manner. In the revised manuscript we now present the results of the
SHARP-track analysis and the resulting locations of RGC axons and SC neurons in Figure S1l-n. We also added
detailed information in the results, methods and discussion sections.

Line 64: “To record neuronal activity in the SC we targeted the visual layers of SC with a tangential recording
configuration that places hundreds of recording sites within the optical layer and superficial gray layers of SC⁵² (Figs.
1c and S1n).”

Line 119: “As expected from anatomy², the majority of recorded RGC axons and SC neurons were located in the optical
and superficial gray layers (Fig. S1, intermediate gray layer: n = 37 RGC axons n = 86 SC neurons; optical layer: n =
641 RGC axons n = 891 SC neurons; superficial gray layer: n = 361 RGC axons, n = 628 SC neurons; zona layer: n =
26 RGC axons n = 45 SC neurons. For RGC axons/SC neurons with reconstructed anatomical location using SHARP-
track^{58,59}, see Methods for more details).”

Line 797: “SHARP-track analysis in the mouse: To identify the Neuropixels electrode track in 3D and to localize
recording sites to brain regions, we used SHARP-track⁵⁹. SHARP-track allows reconstructing the location of the
Neuropixels probe in 3D within the Allen Mouse Brain Common Coordinate Framework based on the histology and
physiological landmarks⁵⁸.”

3. The recording technique appears to be undersampling the inputs from the retina. Of the possible ~30 cell-
types innervating the superior colliculus, only 5 response types are shown. This corresponds with the known depth
distribution of where retinal axons of different types terminate within the superior colliculus.

We thank the reviewer for this comments and apologize for representing the functional diversity of the RGC axon in
such an oversimplified manner in Figure 1. The diversity in the dataset is indeed richer and goes beyond the five types
shown in Figure 1. The rationale for showing only those five types was to highlight that our method is capable of capturing
different functional RGC types and not to show the complete diversity of the RGCs in the dataset. However, we agree
with reviewer #1 that this has to be clarified to address potential over-/undersampling biases. To uncover the diversity
of recorded RGC types we have performed new analyses, added subpanels to Figure S5a/b and removed the
misleading panel I from Figure 1. We also added a section about the diversity of the RGC types in the methods and
results section.

In brief, to characterize the diversity of the RGC axons we adopted an approach introduced by Roson et al.³; we
correlated the RGC axon responses to the chirp stimulus in our dataset to the 32 RGC types published by Baden et
al.⁴ (dataset obtained from: <https://doi.org/10.5061/dryad.d9v38>). This analysis showed that the diversity of the RGCs
in our dataset covered the RGC types reported in Baden et al. (Figure S5a/b). However, since we did not include UV
stimuli in our stimulus set this classification is limited and therefore we only show the overlap with the Baden dataset
to highlight the diversity of RGC types in our dataset. We did not use this classification for further analysis in the study.
We now added the section “Functional diversity of RGC axons and SC neurons” in the methods and modified the main
results text.

Line 125: "... RGC axons derived from a diversity of retinal pathways¹ (Fig. S5a/b)."

Line 892: "Diversity of RGC axons: To characterize the diversity of the RGC axons we adapted a correlation analysis approach from Rosón et al.²⁹ and correlated the visually evoked RGC axon responses to the chirp stimulus to the 32 RGC types published by Baden et al.¹. The Baden data was obtained from <https://doi.org/10.5061/dryad.d9v38>. We estimated the correlation coefficient between the chirp responses of the RGC axon to each of the 32 classes in the Baden dataset and the RGC axon was assigned to the class in the Baden dataset with the highest correlation value (Fig. S6)."

Below we provide a more detailed critique of these issues.

Major issues

1. *There is a fundamental mismatch between the recording technique and anatomy of the superior colliculus. While the recording technique samples from a single depth (at least locally) of the superior colliculus (estimated to be near the optic layer), both the dendrites of neurons in the superior colliculus and innervating retinal ganglion cells, are organized across depths. In mice, axons of different retinal ganglion cell types stratify at different depths of the superior colliculus starting at its surface down to the depth of the optic layers. In addition, the dendrites of individual neurons in the superior colliculus are often quite extensive, extending across many depths. For example, wide-field and horizontal neurons have particularly large dendritic trees. It is not clear how local extracellular recordings from the optic layer (or any single depth) of the superior colliculus can adequately sample both the action potentials of a single neuron and its retinal inputs. How did the authors determine they could see both distal and proximal inputs to a single neuron?*

We thank the reviewer for this important question. We apologize if we conveyed the message that we can sample from both proximal and distal inputs to single neurons. While our approach allows the identification of the connected pairs using spike-train cross-correlation and measurement of the physical distance between the waveform centers of the RGC axons and SC neurons on the probe, it cannot reveal the location of the synaptic contact. The distribution of the physical distance between the centers of the RGC axons and SC neurons covers a wider range (Fig. 3e) with some connected RG-SC pairs being > 200 μm apart. Thus, on the population level, our data likely represents both proximal and distal RGC inputs to SC neurons. We now discuss this issue in detail:

Line 503: "Despite this high yield in identifying connected pairs, the method cannot capture the full constellation of inputs to individual neurons nor unambiguously reveal whether a connection is located on the proximal or distal part of the postsynaptic dendrites (the dendritic arbors of certain cell types can span several hundred μm ⁶⁰). However, the wide range of physical distances between RGC axons and SC neurons on the probe (Fig. 3e) suggests that both proximal and distant connections were captured by our method."

What evidence do the authors have that they are regularly sampling a large majority of the inputs to any single neuron they are recording from? We find it unlikely to be the case and request additional evidence to demonstrate that for an individual collicular neuron an adequate distribution of retinal ganglion cell inputs can be recorded to support the claims of the paper.

We thank the reviewer for highlighting this important point. Previous work estimated that SC neurons receive converging inputs from a small number of retinal inputs, e.g. Chandrasekaran et al.⁵ estimated that around five RGCs connect onto individual SC neurons. In our dataset, we do identify a considerable number of convergent connections with 3 or more connections (3 converging connections = 73 pairs, 4 converging connections = 39 pairs, 5 converging connections = 19 pairs, data taken from Fig. 3f). Therefore, we conclude that our method allows the identification of an adequate distribution of RGC inputs to individual SC neurons. Furthermore, the contribution of individual RGCs to the spiking of their postsynaptic SC neurons can reach values of up to 70-80%. Usrey et al.⁶ has demonstrated that similar high contribution values likely reflect a small number of connections. If SC neurons received on the order of 100-200 RGC inputs the contribution values should be considerably smaller, as is the case for thalamic input to cortical neurons^{7,8}. We do acknowledge the important issue raised by the reviewers and we have now discussed this point in the revised manuscript.

Line 508: "Furthermore, we could identify multiple (3-5) converging RGC inputs to SC neurons (Fig. 3f), which is in the range (~5) of the reported number of converging RGC neurons onto SC neurons⁷⁹. Thus, our approach can adequately

sample the presynaptic RGC pool of individual SC neurons, although such a high sampling is achieved only in a subset
of SC neurons (Fig. 3f).”

2. A catalogue of the depths of recordings is not adequately reported. It remains unclear at which depth of the
superior colliculus Neuropixels probes penetrated relative to internal landmarks. This information is indispensable in
estimating what cell-types were likely sampled (both collicular and retinal). We recommend showing an analysis of the
location (in particular depth) in relation to known anatomical landmarks (both mouse and zebra finch). For example,
using eye injections of CTB to see where the probe track lies in relation to the optic layers. Other solutions are of course
possible.

We now provide more information on the depths of the recordings using the 3D reconstruction SHARP-track method^{1,2}.
This method allows reconstructing the electrode track and assigning the location of the probe within the Allen Mouse
Brain Common Coordinate Framework. We used the visually evoked multi-unit activity as physiological landmarks to
determine the start and end of the SC circuit on the probe. Furthermore, we ensured that the retinotopic map changed
continuously and systematically within the channels assigned to the SC, with sudden jumps in retinotopy indicating the
channel with the visual circuit border⁹, in our case where the probe was leaving the SC tissue. By assigning the
recording sites to the brain region, this analysis showed that the majority (~93%) of RGC axons and SC neurons were
located in the optic layer and the superficial gray layer. For the zebra finches we used similar physiological landmarks,
i.e. visually driven activity and a smooth retinotopic map, to identify recording sites within the optic tectum. Moreover,
the recording location was confirmed by visually inspecting the recording track post hoc in brain slices (Fig. S8). The
new results and the method are shown in Figure S11-n and discussed in the main text in the revised manuscript.

Line 805: “SHARP-track analysis in the mouse: To identify the Neuropixels electrode track in 3D and to localize
recording sites to brain regions, we used SHARP-track⁵⁹. SHARP-track allows reconstructing the location of the
Neuropixels probe in 3D within the Allen Mouse Brain Common Coordinate Framework based on the histology and
physiological landmarks⁵⁸.”

Line 801: “The optic tectum was sliced into 90 μm sagittal sections, mounted using DAKO (Agilent), and the recording
location was confirmed by visually inspecting the recording track post hoc in brain slices.”

Moreover, for the mouse we have added a new analysis of the connection efficacy and contribution as a function of the
SC target layer, focusing on SC neurons located in the optical and superficial gray layer. While we did observe
statistically significant difference between connections to neurons in both layers the effect size was small. Therefore,
we pooled connections across layers together.

Line 299: “The location of the majority of RGC-SC pairs could be assigned to the optic layer ($n = 633$ pairs) or to
superficial gray layer ($n = 271$ pairs) of the SC (Fig. S1n, see Method). We observed statistically significant differences
between the connection strengths to neurons in the two layers (optic layer: efficacy = $3.78 \pm 4.16\%$, contribution =
$16.59 \pm 12.21\%$ $n = 633$ connected pairs; superficial gray layer: efficacy = $4.20 \pm 4.16\%$, contribution = $12.76 \pm 9.37\%$, $n =$
271 connected pairs; $p = 0.002$ for efficacy and $p < 0.001$ for contribution). However, the effect size was small (Cohen’s
$d = -0.09$ for efficacy and Cohen’s $d = 0.33$ for contribution) and the differences between the optic and superficial gray
layers in SC are thus negligible. Therefore, we pooled the data across all SC layers in all further analyses.”

3. The population of collicular neurons appears undersampled. We request an analysis of which collicular cell-
types are being sampled (in particular across depth) to demonstrate that enough cell-types are being recorded to
support the broad claims of the paper. Or identify a reduced population and narrow the claims made by the paper.

We agree with the reviewer and now integrate a new state-of-the-art analysis to investigate the diversity of the recorded
SC neurons. To estimate the diversity of SC neurons we analyzed and clustered the visually evoked responses related
to the light sparse noise, dark sparse noise, chirp and moving bars by means of Uniform Manifold Approximation and
Projection (UMAP) projection and gaussian mixture modeling¹⁰. We identified 19 different SC neuron classes indicating
that a majority of SC neuron types were sampled. In the revised manuscript we added the results of this new analysis
in Figure S6, which we describe in the methods, results and discussion sections.

Line 125: "RGC axons derived from a diversity of retinal pathways¹ (Figs. S5a/b) and SC neurons covered a broad
range of functional response classes across the population (Fig. S6)."

Line 595: "While our dataset contains a diversity of SC neuronal types (Fig. S6) we did not observe any obvious
systematic differences between those neurons and more work is needed to clarify which retinal pathways are combined
or relayed at the level of the diverse SC neuronal types."

Line 909: "Diversity of SC neurons: To characterize the functional diversity of the SC neurons we employed an
unsupervised clustering approach. To that end, for each SC neuron the visually evoked responses to the chirp, dark
sparse noise, light sparse noise and the moving bar were concatenated."

4. The population of retinal ganglion cell types appears undersampled. We request an analysis of which retinal
cell-types are being sampled (in particular across depth) to demonstrate that enough cell-types are being sampled to
support the broad claims of the paper.

We now provide a detailed analysis on the diversity of RGC types (see response to question 3 line 91 in this
document).

5. We have concerns that the method of how RGC-SC pairs are determined might create a selection bias that
leads to the conclusion that retinal ganglion cells in general strongly drive the downstream partners in the colliculus.
One potential issue is that connection efficacy and contribution were determined by inspecting the spiking activity in
response to a full-field "chirp" stimulus which induces strong stimulus correlations in retinal ganglion cells of the same
type. A stimulus that induced sparse responses in both the retinal ganglion cells and collicular populations might be
more appropriate.

We thank the reviewer for bringing up this issue and we apologize that we have not communicated how we calculated
the efficacy and contribution in the main text. The efficacy and contribution were calculated using spikes across the
entire recording duration. This information was provided in the method (line 791 of the original manuscript) but not in
the main text. We now provide this information also in the main text to make this point clear.

Line 283: "To quantify these observations, we estimated the connection efficacy and connection contribution⁴⁴ for each
connected pair from the spike times of the entire recording."

6. Isometric mapping of retinal ganglion cell mosaics appears to have been determined for just one putative cell
type, transient Off cells. Either demonstrate that this generalized to a larger set of retinal cell types, or make clear that
this claim can only be made for this cell type.

We now provide the results of a new analysis that includes RGCs irrespective of their functional responses when
estimating the match between the receptive field and axonal field positions. The only selection criteria were a high
signal-to-noise receptive field (SNR > 10) and a good fit of the axonal contact field ($R^2 > 0.8$). This analysis supports
the conclusion that the match between the receptive and axonal fields position is a general property. We have now
included the results of this analysis in figure S5 and mention it in the main text. However, we do also agree that more
work is needed to fully answer whether all retinal ganglion cell types follow this principle and we added text regarding
this point in the discussion.

Line 211: "To determine if retinal ganglion cell axons are generally organized in this spatially precise manner, we
analyzed the receptive fields and axonal fields of simultaneously recorded RGCs irrespective of their functional
responses..."

Line 547: "Although our dataset included a wide diversity of RGC types (Figs. S5a/b), we have grouped RGCs into
putative functional types based on the similarity of the responses to the chirp stimulus and based on the sparse noise
receptive field only (Fig. 2). Therefore, it could be that not all RGCs were classified appropriately. However, the main
result holds true when pooling across RGCs independent of their functional responses (Fig. S5h), supporting the
conclusion that the precise axonal wiring is a general principle. Nonetheless, it remains to be clarified whether all RGC

types follow this precise organizing principle or whether differences across RGC types and location in the retina⁹¹⁻⁹⁵
exist. ”

7. It is not clear how many retinal ganglion cell axon terminals and collicular neurons the analyzed 1048 RGC-
SC pairs consist of.

We now provide this information in the main text on line 251.

It is important to know whether Figures 4F and 5G contain all recorded RGCs and SC neurons, respectively, and if not,
which exclusion criteria were applied.

We thank the reviewer for pointing to this lack of detail. Figures 4f and 5g contain only a subset of the measured RGC-
SC pairs. The main reason is that the analysis shown in 4f and 5g are based on divergent connections (4f, n pairs \geq
3) and convergent connections (5g, n pairs \geq 3), which was only possible for a subset of RGCs (4f) and SC neurons
(5g). Moreover, in this analysis we aimed at investigating whether multiple strong connections could be found within a
given pool of divergent connections. Therefore, the divergent connections had to contain at least one pair with strong
connection (efficacy > 0.1) or high connection contribution (contribution > 30). This exclusion criterion was introduced
to allow interpreting the results but does not affect the significance of the finding (without this criterion: efficacy 1st vs
2nd: $p < 0.0001$, $n = 120$; contribution 1st vs 2nd: $p < 0.0001$, $n = 120$). The analysis shown in 5G required that the responses
to the chirp stimulus had a signal-to-noise > 2 . We now provide the exclusion criteria in the manuscript.

Line 313: “In our data, divergent RGC connections are characterized by only one or a few strong connections with SC
neurons and multiple weaker connections (Figs. 4e/f and S6e, efficacy: 1st = $16 \pm 9\%$, 2nd = $6 \pm 3\%$, $p < 0.001$, $n = 30$
divergent connections with at least three connections and efficacy 1st $> 10\%$). Likewise, RGCs contributed most
strongly to the spiking activity of only a few postsynaptic SC neurons (contribution: 1st = $45 \pm 11\%$, 2nd = $21 \pm 6\%$, $p <$
0.001 ; $n = 30$ divergent connections with at least three connections and contribution of 1st $> 30\%$).”

8. A number of important references are missing, in particular regarding cell types of mouse superior colliculus
and how the superior colliculus is innervated by different retinal ganglion cell types. While some of this information is
referred to in reviews cited, the paper appears to skip over important anatomical details of how different cell types
(retinal and collicular) are distributed across the depth of SC. For example, papers by labs of Gabe Murphy, Joshua
Sanes and Marta Bickford. In addition, a couple of important papers that directly demonstrate how retinal ganglion cells
innervate either the thalamus or superior colliculus are also missing. For example, from the labs of Botond Roska and
JC Cang.

We now provide more anatomical details on the SC and the innervation of the retinal ganglion cells in LGN and SC in
the revised version of the manuscript and we cite a series of original studies in addition to review articles on this topic.

**Minor issues**

1. More details about the coordinates (AP, ML), angle and depth of probe insertion would be helpful for the
SC/OT community.

We now provide these details in the revised version of the manuscript.

Lines 739: “The Neuropixels probe was inserted either tangentially in the superior colliculus from the back (Figures
S1b/d, antero-posterior insertion: 15 to 25 deg, 500 to 1200 μm ML, -100 to -500 μm DV, -100 to -300 μm AP from
lambda) or from the side (Figs. S1b/h, medio-lateral insertion: 20 deg to 30 deg, -100 to -500 μm DV, 0 to 900 μm AP).
The angles in the antero-posterior insertions were measured in reference to the azimuthal plane, with the probe initially
aligned to the brain midline so that it remained within a sagittal plane. Similarly, the angles in the medio-lateral insertion
were measured in reference to the azimuthal plane, with the probe being perpendicular to the brain midline in order to
stay within a coronal plane. In the zebra finch, the insertion was performed along the antero-posterior axis (within
sagittal planes) at 40 deg from the azimuthal plan (Figs. S8a/b, in reference to lambda: 3000 to 3800 μm ML, -4250 to
322 -5000 μm DV, 4000 to 4800 μm AP).”

2. The picture of RGC axon terminals targeting SC somas is misleading. Retinal ganglion cell terminals will rather
target the dendrites at different depths, including far from the probe.

We now include a schematic of the SC dendrites in Fig 1c. Please note, we only showed one possible dendritic shape
to keep the schematic of the recording configuration as simple and concise as possible in Fig. 1c, however a reference
for further details was added in the figure legends.

Line 141: "Note: SC neurons have diverse dendritic morphologies⁶⁰ and only one stereotypical morphology is shown
here."

3. Line 104: In the mouse retina, gap junctions are also present between retinal ganglion cells of different types
(Cooler & Schwartz, 2020), hence, it is not valid to include gap junctions into the criterion of a single type based on cat
data.

In Figure 1h we used the coupling between RGCs to show that the spatial resolution of the Neuropixel probes is
sufficient to sample from axons of neighboring RGCs. In our dataset most coupled pairs have almost identical
responses to the chirp stimulus (see new Fig. S5e). Based on this finding and previous work, we used coupling as an
indicator that RGCs are from the same functional type. However, we agree with the reviewer that this is an
oversimplification. Therefore, we now only use coupling to support that we can record from neighboring RGCs (Fig. 1h
and Fig. S5e) and do not use coupling as a criterion to identify RGCs belonging to the same functional type. We
modified the sentence in the revised manuscript and cite the Cooler & Schwartz 2020 paper in this context.

Line 111: "In addition, in such neighboring RGC pairs we were able to occasionally observe putative electrical coupling
between RGCs. This was evident in the double peaks in the cross-correlograms (CCG), which is a defining
characteristic of coupling between neighboring RGCs of the same⁵⁶ and different type⁵⁷ (Figs. 1h and S5e)."

4. When referring to the retinothalamic connectivity please be explicit about species. On line 273 most papers
were from experiments in cats and monkeys, not mice or zebra finches.

We now explicitly state the species in this sentence.

Line 274: "Previous studies in cats have shown..."

5. Line. 146: there are enough papers that demonstrate that a high-resolution map does not necessarily
require a one-on-one mapping. Also, it is not clear why this would be the case in the superior colliculus.

We agree that a high-resolution map does not necessarily require a one-on-one mapping. In the revised manuscript
we have modified the motivation of this section.

Line 159: "Next, we wanted to reveal the fine-scale spatial organization of multiple neighboring RGC axons in the SC.
While previous anatomical work has demonstrated that axons from single RGCs form dense and stereotyped arbors in
the SC⁴, it remains unknown how the axons of neighboring RGCs are organized in relation to each other within the
SC."

6. Line. 320: Not clear whether Figure 4F includes all previously determined connections.

We now define the exclusion criteria in the main text as well as in the method section (see also: response to question
7 line 386 in this document).

7. Line. 823: Only direction tuning was analyzed in this paper but the section in the Methods refers to orientation
tuning. Was orientation tuning analyzed as well? What was the similarity between collicular neurons and ganglion cell
afferent inputs in orientation tuning?

We apologize for this mistake. It was meant to be called direction tuning. However, based on this suggestion we now
analyzed the orientation tuning of connected RGC-SC pairs as well. Our results support that the preferred orientation
of connected pairs is similar (mean preferred orientation difference = $10.50 \pm 8.2^\circ$, $n = 7$ connected pairs). We renamed
the method section to "Direction and orientation tuning" and provide the information on the orientation tuning analysis
in the results section.

Please note, in response to reviewer #2, we now characterize the similarity between connected pairs more generally
based on the responses to the dark and light sparse noises, chirp and moving bars. Because this general similarity
measure captured what we aimed at conveying in a more concise manner, we removed the comparison between the
preferred direction from Figure 5 and report these results only in the results text.

Line 361: "We found that connected and direction-selective RGC-SC pairs had similar preferred directions (mean
preferred direction difference = $24.23 \pm 29.15^\circ$, $n = 50$ connected pairs), confirming previous results²⁷, and that
connected orientation-selective RGC-SC pairs had similar preferred orientations (mean preferred orientation difference
= $10.50 \pm 8.22^\circ$, $n = 7$ connected pairs)."

**Reviewer #2 (Remarks to the Author):**

*Summary*

*The authors devise a novel approach to simultaneously record from retinal axons and postsynaptic neurons in the*
*superior colliculus of mice and the optic tectum of zebra finches. They provide compelling evidence for the validity of*
*their approach. The authors then use this novel technique to show that the retinotopy of retinal axonal projections in*
*superior colliculus is preserved at a high level of precision. Moreover, they use the high temporal resolution of their*
*technique to identify directly connected pairs of retinal axons and collicular/tectal neurons to quantify connection*
*strength and functional similarity between retinal inputs and collicular/tectal neurons.*

*Major comments*

1. *The recorded data are of very high quality, and the claims that some actional potentials could be identified*
*as from retinal axons while others from collicular neurons are very well substantiated.*

We warmly thank the reviewer for this compliment.

2. *The authors claim that the mosaic organisation in the retina was preserved at the level of retinal ganglion cell*
*(RGC) axons in superior colliculus (SC), however the evidence that they measured retinal mosaics is very limited.*

We now provide further evidence that the RGC receptive field organization is maintained at the level of the axons in
SC by performing the same analysis on RGCs irrespective of their functional type. For details please refer to response
to the similar question by reviewer #1 (point 6 on line 334 of this letter).

(A) *The visual stimulation used to classify RGC types was reduced to a subset of stimuli that was used to distinguish*
*the currently accepted number of RGC types (see Ref. 1: Baden et al., Nature, 2016). Consequently, the authors may*
*have pooled several RGC types into the same functional mosaic. Regarding the RGC classification, it is not clear how*
*this was achieved. Please, add more details.*

We thank the reviewer for pointing to this lack of detail. We now provide more details on the classification of the RGCs
in the results section (see also line 97 in this rebuttal). Despite classifying the RGC types as rigorously as possible, we
cannot rule out that some of the RGC types were misclassified. We think that this potential misclassification will not
affect the results, since the new analysis showed that the results hold true when pooling RGCs independent of their
type (see below).

Line 582: "Although our dataset included a wide diversity of RGC types (Figs. S5a/b), we have grouped RGCs into
putative functional types based on the similarity of the responses to the chirp stimulus and based on the sparse noise
receptive field only (Fig. 2). Therefore, it could be that not all RGCs were classified appropriately. However, the main
result holds true when pooling across RGCs independent of their functional responses (Fig. S5h), supporting the
conclusion that the precise axonal wiring is a general principle. Nonetheless, it remains to be clarified whether all RGC
types follow this precise organizing principle or whether differences across RGC types and location in the retina⁹¹⁻⁹⁵
exist."

Line 983: "This analysis was performed for RGCs with similar response properties (Fig. 2) and independent of their
functional type (Fig. S5). Functional similarity was assessed by comparing the evoked chirp responses and the
receptive field properties (ON or OFF) to light and dark sparse noise."

(B) *The overlap of receptive fields (RFs) in the mosaic in Fig 2C seems very high, higher than the often-observed*
*coverage factor of 2 (see Ref. 1), which would speak for a pooling of several RGC types.*

We thank the reviewer to pointing to this important detail and apologize that we did not notice this issue in the initial
submission. In the analysis shown in Figure 2 we mainly focused on the center of the receptive fields and therefore did
not pay attention to the large receptive field overlap. Revising the experimental paradigm revealed that the large overlap
is an artifact from the size of the sparse noise stimulus. To map receptive fields, we used sparse noise targets of three
different sizes (5 deg, 10 deg and 15 deg) that were presented on a grid of 36x22 positions, grid spacing 5 deg. While

a 5 deg target was shown in only one grid position, the 10/15d eg targets covered multiple grid positions and therefore
 overestimated the receptive field size. To illustrate this, Figure R1 in this letter shows the receptive fields of the same
 recorded neuron measured with the three sparse noise sizes.

In the original manuscript we used the receptive fields measured with the 10 deg sparse noise because the signal-to-
 noise of the receptive fields was, on average, higher as compared to the receptive fields mapped with the 5 deg sparse
 noise targets. When plotting the receptive fields as contour lines we did not consider that 10 deg sparse noise
 overestimates the receptive field size (Figure R1A-C, please compare the black and red contour lines), resulting in the
 large overlap that the reviewer identified. To compensate for this measurement artifact, we estimated a scaling factor
 for the contour line threshold for which the contour of the 10 deg matches the contour of the 5 deg receptive field size
 (Figure R1C). Applying this scaling factor when plotting the RGC receptive fields shown in Figure 2 reduces the
 receptive field overlap considerably. We now provide this detail on the analysis in Figure S1c where we show the three
 different sizes of sparse noise and the resulting different sizes of the measured receptive fields.

Line 948: "Receptive fields mapped with 10/15 deg targets overestimate the receptive field size mapped with 5 deg
 targets (Fig. S1c) and therefore we scaled the threshold for the contour lines by a factor of 1.4 when plotting receptive
 fields mapped with 10 deg or 15 deg (the factor 1.4 was estimated from the data)."

**Figure R1. Receptive field size estimation with the different sparse noise stimuli.** (A) Three sizes of sparse noise
 targets were used to characterize the receptive field in this study (5, 10 and 15 deg). The targets were presented on the
 same 5 deg grid and the 5 deg target covered one grid position (left), the 10 deg target 2x2 grid positions (middle) and
 the 15 deg target 3x3 grid positions (right). While the 5 deg target provides the most precise characterization of the
 spatial receptive field size (left), the 10 and 15 deg targets usually resulted in a higher signal-to-noise receptive field.
 Therefore, presenting the different size targets on the same 5 deg grid was done to measure the receptive field center
 location at a resolution of 5 deg even with the 10 and 15 deg stimuli. While the receptive field center position is
 accurately characterized with all three target sizes, the estimated receptive field size is overestimated by the 10/15 deg
 stimuli due to the spatial blurring of the 10 and 15 deg stimuli (middle and right). (B) Left, shown are the contour lines
 of the receptive fields with the same threshold shown in A and measured with the different target sizes. Right,
 compensating for the spatial blurring by increasing the threshold of the contour line for the 10 deg target size.
 (C) Analysis of the receptive field size using the 5 and 10 deg sparse noise targets. The 10 deg target size overestimates
 the receptive field size measured with the 5 deg targets. Note that all black data points are all above the unit line.
 Increasing the threshold for receptive field size estimation compensated for the spatial blurring by the 10 deg stimulus.
 Green data points lie now on the unity line. (D) RGC receptive field mosaic as shown in the original version (left) with
 the overlap of the receptive fields. Compensating for the spatial blurring by the 10deg stimulus reduces the overlap of
 neighboring RGCs (right). The version on the right is now shown in the revised manuscript.

 (C) The Neuropixels probes are very long but very thin (maximum channel distance across probe width is 75 μ m).
 The possible coverage of 2D visual space by RFs of simultaneously recorded neurons or axons is therefore very
 limited. This also limits the possibilities to quantify retinal mosaics (see Major point 4(A) for a related comment).

We agree that the spatial extent of the Neuropixels probe is limiting the 2D coverage and we hope that future studies,
e.g. using two-photon calcium imaging of RGC axons within SC, will characterize the 2D properties in more detail. The
key observation in our study, i.e. that RGCs with neighboring receptive fields have precisely located neighboring axons
in the SC, can be shown without the additional coverage. The estimation of the axon field center is only affected in
cases when the axon field is located on the side of the probe. To address this, we estimated the axon field centers from
the 2D Gaussian fits. This fit allowed us to extract axon field centers at the border of the probe or slightly outside of the
probe, e.g. as shown in the example in Fig. 2a. We now revised the text in the discussion regarding this point.

Line 934: "To characterize the spatial position of the axonal synaptic contact field, we fitted a two-dimensional Gaussian
function to the two-dimensional representation on the probe (Fig. 2a, bottom-right). This Gaussian fit was necessary
because some of the RGC axonal contact fields were only partially covered by the recording sites on the probe, e.g.
the example in Fig. 2a."

(D) The authors only show quantification of the retinal mosaics for a single RGC type (OFF cells for mice, ON-OFF
cells for birds). How do these results compare to other RGC types? Do they generalize? Are the numbers of recorded
RGC types in agreement with previously reported frequencies?

We now provide more information regarding the RGC types for the mosaics in Figs. S5 and S8. Furthermore, we have
also performed new analysis to support the generality of this finding, i.e. we studied the relationship of the RGC
receptive fields and axonal fields independent of the functional type (see comment below). However, as our dataset
may not cover all possible RGC types we now added a sentence in the discussion mentioning that future work could
further investigate the axon mosaics of specific RGC types. Please refer to line 592 in this document.

Despite these issues regarding retinal mosaics, we feel that the main result of the paper, namely that retinotopy is
preserved in retinal axons, still holds. In fact, the quantifications in Fig 2F-I (and Fig S7H-K) showing that RFs and
axonal positions in the brain match very well would be even stronger if all simultaneously recorded retinal axons were
included instead of only the axons from a single RGC type.

We thank the reviewer for this suggestion. We performed the new analysis on RGCs irrespective of the functional type
and the main results hold true. Please see new Fig. S5 and also comment above to reviewer #1 (point 6 on line 334 in
this document).

We think that if the evidence for retinal mosaics cannot be strengthened by the authors, the title of the paper should
be changed accordingly.

We now provide more evidence to strengthen our conclusion about the retinal axon mosaics (see above) and we
hope reviewer #2 agrees. We therefore would like to keep the axonal mosaic aspect in the title.

3. Given that the authors distinguish different RGC types, it would be very interesting to see whether they find
differences between those types, e.g. in terms of efficacy and contribution, also in terms of numbers (do recorded
numbers match those reported in the literature?).

We agree with the reviewer that an in-depth investigation of the various RGC types and their efficacy/contribution is an
important question. We performed new analyses to start addressing this question and we observed that orientation
selectivity of the RGC is negatively correlated to the connection contribution, with the strongest connection being only
weakly orientation selective (see Figure R2 in this letter). While these are promising new results, we feel that fully
answering this question requires new and tailored experiments which are beyond the scope of this study. We added a
note regarding this point in the discussion.

Line 573: "Alternatively, the various RGC types could have specific connection strength to the diversity of SC neuron
types (Fig. S6). Although we observed a significant negative correlation between the RGC-SC connection contribution
and orientation selectivity of the pre-synaptic RGC ($r = -0.28$, $p < 0.001$, $n = 379$ connected pairs), more work is required
to fully answer this question."

Figure R2: Correlation between efficacy/contribution and the direction/orientation selectivity of the presynaptic RGCs. (A) Shown is the correlation between the efficacy and contribution to the direction selectivity index and orientation selectivity index of the presynaptic RGC.

4. Several results are presented without testing them for significance or comparing them to a null hypothesis:

(A) How do the results in Fig 2D+E (distances and angles between RFs and AFs) of a single RGC type compare to results when a comparable number of axons is randomly sampled from all simultaneously recorded axons? Are these significantly different?

The results shown in Figure 2d/e were mainly presented to illustrate a known classical feature of retinal mosaics. To reduce the number of panels and graphs (see comment reviewer #3) we have now removed this panel from the revised version of the Fig. 2 and only show it in the example in Fig. 5d.

(B) Are the log normal distributions in Fig 4C+D and Fig S8B significant?

We now provide the p-values for the test of the log-normality of the efficacy and contribution distributions. To estimate the significance of the distributions we tested whether the log of the values differ from a normal distribution using the `scipy.stats.normaltest` function. This test shows that the distribution of the efficacies is indeed not different from a log-normal distribution for both mice and finch. However, this test revealed that the distribution of the contribution values is different from a log-normal distribution. We now report these values in the manuscript.

Line 308: "Across the population, we discovered a log-normal distribution of connection strength ($p = 0.295$ for testing the hypothesis that the log of the efficacies is not normally distributed, n pairs = 1044, D'Agostino's K2 test), but not for coupling strength ($p < 0.001$).

Line 420: "Similar to the mouse SC, zebra finch OT neurons received a small number of RGC afferents (Figs. 6d and S8f) with a log-normal distribution of RGC connection efficacy (Fig. 6e, $p = 0.376$ for testing the hypothesis that the log of the efficacies is not normally distributed, n pairs = 105, $n = 5$ zebra finches, D'Agostino's K2 test) but not RGC connection contribution (Figs. 6f, $p = 0.009$).

(C) Are the distributions of efficacy and contribution for 1st and 2nd strongest connections (Fig 4F) direct consequences of the log-normal distributions (Fig 4C+D)? By sampling pairs of efficacies/contributions randomly from the log-normal distributions, the authors could compare the resulting "surrogate" distributions to the measured ones. Are they different?

We thank the reviewer for this suggestion. We have performed this analysis and found that sampling from shuffled surrogate data, and analyzing it in the same way as the original data, reproduced the results. We included this result in the manuscript (see result section, Fig S7f/g).

Line 318: "We reasoned that this connectivity motif could be the result of the non-Gaussian distributed connection strength. To test this prediction, we performed a permutation test by randomly sampling connection strengths of divergent connections from the measured distributions and analyzed those randomly generated divergent connections in the same way as the real data. Random sampling could produce similar divergent connection motifs (Fig. S7f) that were statistically similar to the real data (Fig. S7g, 1st data vs 1st shuffled: $p < 0.05$ in less than 0.5% of repeats, 2nd data vs 2nd shuffled: $p < 0.05$ in less than 4% of repeats, $n = 10000$ repeats).

5. The authors state that log-normal distributions for efficacy and contribution are commonly found in many brain
areas and species. This would mean that the retino-collicular connections are not special in any way. This does not
come across when the authors then state that “the retinocollicular circuitry is optimally wired for transmitting retinal
activity in a functional specific manner”, unless the authors think that all circuits (at least those with log-normal
distributions of efficacy and contribution) are optimally wired for functionally specific transmission. Please, clarify.

Considering the reviewers' comment we agree that one cannot answer whether the wiring of the retino-collicular
connections is special or whether all circuits with log-normal distributions are optimally wired for functionally specific
transmission. Therefore, we have removed this sentence in the revised manuscript.

6. Related to the ubiquity of the log-normal distributions, what is the significance for finding these distributions in
mice and zebra finches? This finding then does not speak for an evolutionarily conserved circuit because most likely
also non-conserved circuits show this distribution.

We thank the reviewer for pointing to this misattribution. We did not intend to convey that log-normality is a feeding
mechanism in the observed synaptic connectivity but rather a common observed feature. Consequently, we removed
the usage of this adjective in the revised manuscript.

Linen 420: “Similar to the mouse SC, zebra finch OT neurons received a small ...”

Line 617: “In summary, we showed that the retinotectal circuit in both mouse and zebra finch is characterized by limited
convergence and log-normally distributed connection strength, with connection strength being strongest for functional
similar RGC-SC/OT pairs.”

7. The authors collected unique datasets quantifying connection strengths between retinal axons and SC
neurons on one hand and functional connection motifs (relay or combination) on the other hand. It would be very
interesting if the authors showed how these are related to each other. E.g. do relay motifs have stronger connections
than combination motifs?

We agree with the reviewer that investigating whether relay and combination motifs have distinct connection strengths
is interesting and relevant. With the current dataset we could not observe any significant differences between relay and
combination motifs in regard to the connection strength (see new Fig. 5f). We believe more work is required to be able
to obtain a conclusive answer. While our dataset contained a large number of connected RGC-SC pairs, we were able
to identify convergent connections with more than three RGC afferents only for a subset of SC neurons (n=53). We
have now included the results from the analysis in the new Fig. 5f and added corresponding text in the results.

Line 388: “Despite these differences in afferent inputs, we did not observe systematic differences in connection efficacy
or contribution between these two types of RGC pools (efficacy: relay = $5.43 \pm 5.81\%$, n = 104 connections, combination
= $4.96 \pm 3.42\%$, n = 138 connections, p = 0.73; contribution: relay = $16.00 \pm 9.62\%$, n = 104 connections, combination =
$15.13 \pm 9.76\%$, n = 138 connections, p = 0.33) (Fig. 5f).”

Related to this, the authors use various similarity measures (correlation of spatiotemporal RFs, difference between
preferred directions, correlation of responses to chirp stimuli) to quantify function similarity (Fig 5). It would be preferable
if they combined all of them to determine a single similarity measure and compare this to efficacy/contribution and to
determine whether motifs are relay or combination.

Based on this suggestion we significantly improved the manuscript and Figure 5. We now combine the various similarity
measures into a single similarity measure. To that end we computed the correlation coefficient between the
spatiotemporal receptive fields (both for the light (r_{SL}) and dark (r_{SD} sparse noise), the responses to chirp (r_{chirp}) and the
responses to the moving bars ($r_{m.bar}$). The single similarity measure is then given by: $\text{similarity} = (r_{SD} + r_{SL} + r_{chirp} + r_{m.bar})/4$. We show an example of how we estimated these four measures (r_{SD} , r_{SL} , r_{chirp} , $r_{m.bar}$) in Figure 5a and how the
single similarity measure is calculated in Figure 5b. In Figure 5b we then show the correlation between this similarity

measure and the connection efficacy (Figure 5b, left, $r=0.55$, $p<0.001$, $n=526$ connected pairs) and correlation between
the similarity and the connection contribution (Figure 5B, right, $r=0.56$, $p < 0.001$, $n=526$ connected pairs).

While the single similarity measure captured the link between the connection strength and the functional similarity of
RGC-SC pairs, using this measure to characterize the diversity of the convergent RGC pool was difficult. The jitter in
the retinotopic locations of the RGCs complicates the identification of RGCs with similar functional type using the
correlation of the spatiotemporal receptive fields (r_{SD} , r_{SL}) and the correlation of the moving bar responses ($r_{m.bar}$).
Therefore, we used only the r_{chirp} as the measure of functional similarity of the RGCs as this measure is independent
of the retinotopic position of the RGC receptive fields.

Please refer to the section “Functional organization of the retinocollicular connections in vivo” starting on line 338 for
details.

Coming back to the different RGC types, are there differences in efficacy/contribution between RGC types?

We now analyzed the efficacy/contribution in relation to the orientation and direction tuning, as a first proxy for different
RGC types. While this preliminary analysis showed interesting trends in the data, addressing this question in detail
would require a more extensive stimulus set, e.g. including UV stimuli (see response to question 3 lines 738). Therefore,
we feel that answering this question is beyond the scope of this study.

8. What is the evidence for 2 distinct groups of relay and combination motifs? The scatter plot in Fig 5G seems
to show a continuous range of motifs.

We are grateful for this question which led us to optimize our analysis to demonstrate a bimodal distribution. We now
tested the bimodal shape of the RGC-RGC similarity distribution shown on top of the scatter plot of Figure 5G (r_2
bimodal = 0.86, r_2 = unimodal gauss = 0.16, non-linear least square fit). The RGC-RGC similarity characterizes the
functional diversity of the RGC input pool and we found that a population of SC neurons receives inputs from very
similar RGC types (relay) while for another SC population the afferent RGC pool is more diverse (combination); This
holds true on the level of the responses to the chirp stimulus. We agree with the reviewer that the scatter plot in Figure
5G appears to be more a continuum, which is due the more uniform distribution of the RGC-SC similarity values.
Because our main conclusion is based on the RGC-RGC distribution, and to avoid confusion for the reader, we have
removed the RGC-SC similarity aspect from Figure 5, including the scatter plot, and only show the distribution of the
RGC-RGC similarity in the revised manuscript. We have modified the corresponding results, method and discussion
text.

Line 378: “To quantify this observation, we calculated the correlation of the responses to the chirp stimulus (r_{chirp}) among
the RGCs of the presynaptic pools and used the average of these correlation values to characterize the functional
diversity of the afferent RGC pools.”

Line 585: “Different wiring modes of the retinotectal connections. ...”

9. Although it is very interesting that the reported results are similar for mice and zebra finches, it seems
unfounded to us to generalize these findings to mammals and birds and even to all vertebrates (lines 405-7). Given
that only about 10% of RGCs in the primate project to the SC, while 80-90% of RGCs in the mouse project to the SC,
there may be substantial wiring differences even across mammalian species.

We agree that there are fundamental differences between the visual systems of even different mammalian species and
we have changed the conclusion to specifically focus on mice and zebra finches.

Line 436: “Therefore, our data strongly support the notion that retinotectal circuit follows similar wiring principles in
mice and zebra finches.”

10. The authors state that their novel approach “opens up opportunities to investigate the principles of how afferent
inputs organize in other parts of the brain” (l. 445). This statement may need a more cautious formulation as no attempt

was made to show the feasibility of the approach in other brain areas. One argument against a generalization would
be that the authors did not find any axons from other areas projecting to the superficial SC (for example from V1).

We agree that this sentence was unspecific and we have reworded it. We point out that this method might be only
applicable for areas receiving axons with dense axonal arbors which generate a large electrical signal.

Line 481: "Measuring the synaptic contact field of afferent axons using single high-density electrodes in vivo opens up
new opportunities to investigate the organization and function of long-range axons in vivo. However, it is still unclear
what axonal morphologies generate electrical signals with amplitudes large enough to be captured by high-density
electrodes. RGC axons form dense arbors within SC and modeling work suggests that axonal branching plays an
important role in generating axonal extracellular potentials⁷⁴. Thus, this method could potentially be employed generally
to study long-range axons with dense arborizations in vivo such as thalamo-cortical axons within cortex^{10,75}."

11. The authors suggest that their new approach is particularly suited to investigate functional maps of synaptic
inputs. It seems to us that imaging approaches, e.g. two-photon imaging of axon/synapses, are superior as they provide
a far better coverage of a 2D plane while Neuropixels probes provide a very limited sampling of brain space (max
distance of channels across width of the probe is 75 μ m). The far greater advantage of the presented approach seems
to be its excellent temporal resolution and the ability to detect direct connections. The authors may wish to highlight
this instead.

We agree that two-photon imaging is superior for studying neuronal activity in a 2D plane. However, the ability to detect
connected RGC-SC pairs is the key advantage of our approach. We now emphasize these points more clearly in the
discussion.

Line 554: "Moreover, the small width of the Neuropixels probe only provides a narrow sampling of neuronal tissue in
two dimensions. Two-photon calcium imaging of RGC axons in SC would be well suited to further deepen our
understanding of the functional organization of RGC axons in SC in 2D and potentially also 3D using multi-plane
imaging⁹⁶. Finally, what developmental mechanisms underlie this single cell precise mapping from the retina to the
midbrain and whether this precision is unique to vision or a general principle of sensory afferent organization in the
midbrain^{73,97} are both open yet important questions."

Line 490: "A key advantage of our approach is that the sub-millisecond temporal resolution of the high-density
electrodes permit the detection of synaptically connected RGC-SC pairs in vivo^{33,49,61} at large scale."

Minor comments

• L. 7: "strong connections and limited functional convergence" does not reflect the results. The authors found
mostly weak and only few strong connections (Fig 4C,D), and a range of connection motifs from functionally similar to
dissimilar (Fig 5G). This statement is repeated in l. 490.

We respectfully disagree with the reviewer on this point. We observed weaker connections because connection efficacy
and connection contribution are correlated to the similarity of the connected RGC-SC pair (Figure 5b). The optimal
RGC inputs to SC neurons are strong with efficacies in the range of up to 40-50% and with contribution values up to
70-80%. These values are similar to the range reported for RGC-LGN connections in the cat (e.g. ⁶), which are
considered to be strong driver connections¹¹. It was reported that the efficacy and contribution values for RGC-LGN
connections increased with the similarity (receptive field overlap and ON/OFF polarity) of the connected RGC-LGN
pair⁶. We found a similar phenomenon in our dataset and conclude that the weaker connections most likely reflect the
non-optimal RGC inputs to SC neurons. Moreover, the median value of the connection contribution is around 15%, with
many connections reaching contribution values above 50%. This shows that SC neurons are strongly coupled to
individual RGC inputs. In comparison, the coupling strength between LGN and V1 is typically between 2-4%^{12,13}. We
hope these are convincing arguments that will lead reviewer #2 toward agreeing with our conclusion that SC neurons
are strongly driven by their optimal RGC inputs.

Regarding the “limited functional convergence”. We agree that refereeing to limited functional convergence may not be
the optimal way to convey our findings as we see a range of motifs in our data. We therefore have revised this part of
the abstract.

Line 7: “This isomorphic mapping builds the scaffold for precise retinotopic wiring and functionally specific connection
strength.”

• Introduction: the authors should cite their Ref. 50 here as it is one of the first attempts to investigate the
functional connectivity between RGCs and SC neurons in mouse, which is one the major topics of the present paper.

We thank the reviewer for pointing out this lack of context in our introductions. We now cite this paper in the introduction
(Ref. 27).

Line 17: “While we have learned much about how SC neurons process visual stimuli^{14–26}, how SC neurons integrate
retinal activity on a functional level in vivo is still largely unknown²⁷.”

• L. 108: how many animals were used? Please also add this information wherever appropriate.

We now provide the number of animals and additional information wherever appropriate.

• How many axons and neurons were recorded simultaneously on average across sessions?

The exact number of axons and neurons varies depending on multiple parameters of the recording, e.g. the insertion
angle, recording depth etc.. On average we identify around ~30% of recorded waveforms as axons. We have added
more details about the number of recorded axons and neurons in the main text.

Line 116: “Well-targeted recordings yielded a high number of simultaneously recorded RGC axons and SC neurons
(average number of simultaneously recorded RGC axons = 48 ± 34 and SC neurons = 114 ± 58 , total number RGC axons
= 1199 and SC neurons = 1831, $n = 27$ recordings from 24 mice).”

To increase the number of recorded axons we employed a semi-online analysis that allows assessment of whether
axonal contact field waveforms are present in the dataset within a few minutes. In the method section we now provide
a link to a GitHub repository that contains the necessary code and information.

Line 878: “To optimize the targeting and the yield of axonal signals, we adapted a semi-online approach that allows the
assessment of whether a given insertion contains axonal contact field waveforms. To that end, we recorded ~5 minutes
of neuronal activity and spike-sorted this short dataset with Kilosort2...”

Line 889: “... (<https://github.com/KremkowLab/Axon-on-Neuropixels-in-Kilosort>).”

• Report where RFs were measured in each mouse (elevation, azimuth).

We added this information.

Line 953: “The receptive fields were measured at an estimated average position of +5.25 deg in elevation and +38.45
deg in the azimuthal plane from the nose position. However, due to the tangential insertion angle, the receptive fields
covered a large area of the visual field. Within each mouse the receptive field coverage was on average 100 deg in the
azimuthal axis and 88 deg in the elevation axis.”

• L. 141: how were borders of SC determined?

We now provide this information in the revised manuscript.

Line 155: “The SC borders were identified by a continuous retinotopic map within the visual driven channels.”

- L. 190: Reference S7C is probably wrong

Corrected.

- Fig 2A: The term “dendritic RF” is very confusing as it suggests that there is also an axonal RF.

We thank the reviewer for pointing out this misleading choice of words and we now refer simply to the “Receptive field” in Fig 2a.

- Fig 2B: caption says that RGCs were identified using a chirp stimulus but the cartoon (and responses) only show ON/OFF stimuli.

We now show a longer interval for the responses to the chirp stimulus.

- Fig 2D+E: specify what black vs gray bars show

The black bar showed the data from the receptive field mosaics and the gray bars from the axon mosaic shown in Figure 2c. Note, in the revised version of the manuscript we removed panels 2d and 2e to reduce the density of presented data (comment from reviewer #3).

- Fig 2G (and Fig S7I): please report RF-AF distance in visual degrees; meaning of symbols above the histogram is unclear

We thank the reviewer for this suggestion. However, reporting the distances in the fraction of mosaics spacing is important for normalizing for the different RGC receptive field sizes in our dataset. Therefore, if possible, we would like to keep this unit of distance. The symbols above the histogram were intended to graphically show the distance between the receptive field and axonal field. We have modified those symbols and provide an explanation within the figure legend in the revised version.

Line 233: “The RF/AF above the histogram illustrate the distance at 0, 1 and 2 mosaic spacing.”

- Line 295: Forgot the % sign

Corrected.

- Fig 4B: “Spike” would be more suitable for y-label (instead of “Trial”)

We have changed the label accordingly.

- Fig 4E: state what the lines mean (CCGs presumably)

We now include this information in the legend.

Line 334: “Example of a divergent connection with one strong and several weak connections. The gray lines show the cross-correlogram of the pairs and the inset shows the receptive field contours of the recorded neurons.”

- L. 337: Ref. 50 used intracellular recordings to show that direction preferences of retinal inputs and the connected SC neuron are similar. Why is confirmation by monosynaptically connected pairs pending?

We thank the reviewer for this question and we apologize for the unclear wording. The method used in Ref. 50 captures the retinal input to SC neurons on the population level. What we meant to say is that connected direction selective RGCs and direction selective SC neuron pairs were not measured before and that the aim of showing the data in Figure 5C/D was to confirm the results of Ref. 50. Please note, in response to a recommendation by reviewer #1 we now

estimate a more general measure using the sparse noises, chirp and the moving bars to characterize the functional
similarity between connected RGC-SC pairs. Therefore, we removed the panel with the direction tuning from Figure 5
and only report this information in the rewritten text of that results section.

Line 361: “We found that connected and direction-selective RGC-SC pairs had similar preferred directions (mean
preferred direction difference = $24.23 \pm 29.15^\circ$, $n = 50$ connected pairs), confirming previous results²⁷, and that
connected orientation-selective RGC-SC pairs had similar preferred orientations (mean preferred orientation difference
= $10.50 \pm 8.22^\circ$, $n = 7$ connected pairs).”

• Fig 5E,F,G: what similarity measure was used in histograms and scatter plot?

The similarity was measured by the correlation coefficient between the responses to the chirp stimulus (r_{chirp} in the
revised manuscript). We have changed the labeling of the panel and legend in Figure 5g to explain it more clearly.

Line 401: “Relay motif example of an SC neuron receiving convergent inputs from a pool of RGCs with similar functional
responses. Receptive fields (left). Responses to the chirp stimulus (middle). Spike waveforms, CCGs, contours of RFs,
and the histogram of r_{chirp} between RGC-RGC (orange) (right). The magenta dot shows the average r_{chirp} of the
presynaptic RGC pool.”

Line 405: “Functional diversity of RGC convergent inputs to SC neurons. Histogram of the average r_{chirp} between RGCs
converging onto the same SC neuron. Note that some RGC input pools are very similar with r_{chirp} values close to 1
while others convey a mixed input with lower r_{chirp} values.”

• L. 396: “zebra finch OT neurons receive a limited pool of RGC afferents”. Unclear what the authors want to
express here as the pool must be limited rather than infinite.

Corrected.

Line 420: “Similar to the mouse SC, zebra finch OT neurons received a small number of RGC afferents (Figs. 6d and
S8f) ...”

• Fig 6D+G: it would be more appropriate to show actual data rather than only cartoons/fits in this main figure.
The fits can be plotted on top of histograms/the scatterplot.

We now show the actual data from the zebra finch in Figure 6. Furthermore, to unify this Figure with what is shown in
Figure 4 and 5 we also now report the correlation between the similarity and the connection contribution in Figure 6.

• Fig 6F: to show that there is little scatter in retinal input one needs to see the outlines of single retinal RFs
compared to the outline of the RF of the postsynaptic neuron. Also, it is difficult to compare a contour line to a RF.

We thank the reviewer for requesting this change, which helped us to improve that part of Figure 6. We now show
individual retinal RFs as orange outlines and the average of the SC RFs as one black outline. We only show the average
of the SC RFs because each SC RF was centered around the origin (0,0) in this analysis, to account for the different
retinotopic locations of all recorded RGC-SC pairs. Showing individual SC RFs would not provide additional information.
To integrate and visualize the synaptic strength we adjusted the alpha value for each RGC RF outline depending on
the connection efficacy, with strong connections having a high alpha value and weak connection a low alpha value.

• L. 495: Liang et al (Cell, 2018, A Fine-Scale Functional Logic to Convergence from Retina to Thalamus.)
shows a different picture and should be mentioned here. Another important paper on thalamo-cortical connections is
da Costa et al (J Neurosci, 2011, How thalamus connects to spiny stellate cells in the cat’s visual cortex)

The aforementioned papers are added.

Line 576: "Taken together, the efficient way SC/OT neurons integrate RGC inputs is reminiscent of the way neurons in
the dLGN integrate retinal inputs^{29,44} (but see⁸⁰)."

• L. 610: what is the insertion point in AP? Are the reported ranges the different insertion points of the coverage
of the probe within the brain? For reproducibility, insertion point is more important. Also, it is unclear what is meant by
the angle. As the probe can be rotated and tilted in 3D, please specify in which plane the angle is measured (as written
azimuthal) and what is 0 degrees?

We thank the reviewer for indicating this lack of clarity. We now provide more details on the insertion points and angles.

Line 739: "The Neuropixels probe was inserted either tangentially in the superior colliculus from the back (Figures
S1b/d, antero-posterior insertion: 15 to 25 deg, 500 to 1200 μm ML, -100 to -500 μm DV, -100 to -300 μm AP from
lambda) or from the side (Figs. S1b/h, medio-lateral insertion: 20 deg to 30 deg, - 100 to -500 μm DV, 0 to 900 μm AP).
The angles in the antero-posterior insertions were measured in reference to the azimuthal plane, with the probe initially
aligned to the brain midline so that it remained within a sagittal plane. Similarly, the angles in the medio-lateral insertion
were measured in reference to the azimuthal plane, with the probe being perpendicular to the brain midline in order to
stay within a coronal plane. In the zebra finch, the insertion was performed along the antero-posterior axis (within
sagittal planes) at 40 deg from the azimuthal plan (Figs. S8a/b, in reference to lambda: 3000 to 3800 μm ML, -4250 to
928 -5000 μm DV, 4000 to 4800 μm AP)."

• L. 641: what's the length and width of the bars?

The width was 10 deg and the length was larger than the screen/dome.

Line 716: "Moving bars: To measure the orientation and direction tuning, we presented moving white bars on a dark
background. The bars moved in 1 out of 12 directions (30 deg spacing between directions) on every trial at a fixed
speed of 90 deg/s. The bars were 10 deg in width and with a length that covered the entire projector image/screen."

• L. 642: provide details about the chirp stimulus (starting and ending frequency, speed of modulation, ...)

We have added the details about the chirp stimulus in the revised manuscript.

Line 710: "Full-field chirp: To characterize the contrast polarity, temporal frequency as well as contrast response
properties we presented a full-field chirp stimulus¹. The full-field stimulus varies in brightness: it starts with a gray
background and several light decrement and increment steps (~2.18 s black, ~3.28 s white, ~3.28 s black, 2.18 s gray)
followed by sinusoidal intensity modulations with increasing frequency (0.5 Hz to 11 Hz) at full contrast (8.75s) and
increasing contrast (0 to 100 %) at 0.4 Hz (8.75 s) and ending with 2.18 s gray background."

• L.642: "The timings... " This statement is unclear. What synchronizing signals? Marked where?

We now provide more details on the synchronizing signals.

Line 677: "Visual stimuli were generated in Python using the PsychoPy¹⁰⁸ toolbox. The onsets of the visual stimuli were
marked by a TTL signal that was generated and time-locked to the screen update on the stimulus computer and
recorded together with the neuronal signals from the Neuropixels probe."

• L.646: what was the extent of the removal of visual cortex? Are there histological records? We suggest to
mention the cortical removal in the main text as it is a major influencing factor.

We now mention the cortical removal in the main text and provide further information about the extent of the removal
in the methods.

Line 89: "To further test this hypothesis, we performed a series of in vivo pharmacological experiments in mice in
which we had removed most of visual cortex to ensure that the axonal signals do not originate from visual cortex"

Line 763: "To that end, the skull was open above visual cortex (1 mm to 3 mm lateral from midline and -2 mm to -4 mm from Bregma) and the underlying cortex was manually aspirated via a pipette."

Here we show a histological record of the cortical removal:

Figure R3: Illustration of the extent of the cortical removal. Shown is the mouse brain with the visual cortex highlighted in color and the approximate position of the coronal brain slice (right). The green fluorescence signal in the slice is from the Alexa 488 Conjugate that was added to the muscimol solution which was injected into SC and that spread to cortex.

• L.655: under which conditions were the experiments deemed as not successful?

Clarified.

Line 778: "Experiments with synaptic blocker mixture were considered unsuccessful when we encountered a problem in the second injection (n = 3 successful double-injections)."

• L.705: "The window was interpolated (101 times)". Please clarify. How is the window defined (space and time)? What does 101 times interpolating mean? In space and time? Was smoothing done in space also?

We apologize for the typo '101 times' which was corrected to be 10 times. The interpolation was done in the temporal domain to obtain more data points for the waveform characterization (Kaufman et al. 2010). Smoothing was done in time and space using a Gaussian filter (sigma time = 0.1 ms, sigma space = 2 recording sites along the probe).

Line 863: "This window was interpolated in time (10 times) and subsequently smoothed in time and space using a Gaussian filter (sigma time = 0.1 ms, sigma space = 2 recording sites along the probe)."

• L.707: "All slope measurements...". Unclear

Line 867: "All four slope measurements (S1-S4 in Fig. S3a) were defined as the 80th percentile values of the observed peaks."

• L.709: Fig S2B is probably the wrong reference

Corrected.

• L.755: how is synaptic contact field defined?

We have reformulated this definition in the main text.

Line 932: "Thus, we defined the area on the probe that contains the post-synaptic response of the SC dendrites the axonal synaptic contact field (AF)."

• L.791: do the CCG results depend on the stimulation protocol? Can it be measured during spontaneous
activity to exclude possible drive by visual stimuli?

The reviewer is right that the CCG peaks can vary with stimulation protocol. However, in order to avoid biases from
particular tuning of certain neuron types versus others we estimate the CCGs over the entire recording period. To
compensate for stimulus driven modulations of the CCG we have used the established jitter correction method. Due to
the low spontaneous activity in SC, at least in anesthetized experiments, measuring CCGs just from spontaneous
activity was, unfortunately, not possible. In the revised manuscript we now provide this information.

Line 994: "Spike times over the entire recording were used in the CCG analysis to avoid biases inherited from a
particular tuning following the exposure of a particular protocol."

• L.805-808: Clarify how contribution was determined: from retinal spikes of a single axon or from retinal
spikes of all recorded axons to a specific neuron? If the latter, is it somehow normalised to the number of detected
synapses?

The contribution was measured for single RGC axons. We now provide this information in the main text.

Line 290: "Next, we estimated the connection contribution, which characterizes the fraction of SC action potentials that
are driven by the activity of presynaptic RGCs and therefore provides a measure for how strong SC neurons are coupled
to the activity of individual RGC inputs."

• L.814: lag of -1 ms or -1 frame?

Corrected.

Line 945: "The spatial receptive fields were estimated via spike-triggered averaging (STA) and by using the receptive
field at lag -1 frame as the corresponding onset receptive field³⁴."

• Fig S1: Please use fewer abbreviations. The text is currently very hard to understand.

We apologize for using too many abbreviations. We modified the figure and legend to make it more understandable.

• Fig S1G: Left and middle panel look very different. Are they not the same example? What is meant by
"shank"?

The examples are from the same RGC. The waveforms appear slightly different because on the left we showed all
channels of the Neuropixels probe while on the right only one shank/column of the probe is shown. Because the
recording sites on the Neuropixels probe are arranged in a checkerboard pattern we often observed slight difference in
waveforms between the left and the right side/column of the probe and therefore we showed only one shank in the
zoom. With shank we meant the left and right column of recording sites on the Neuropixels probe. To avoid confusion
for the reader and to reduce the overall number of panels and figures we have removed the panel with the zoom in the
revised figure, which is now shown in the new Fig. S2a-c.

• Fig S1J: Right panel not clear. Are these sagittal sections? If so, what are the ML coordinates of each
section? What are we supposed to see? Please adapt brain atlas images to reflect histological images.

The brains of the medio-lateral insertions were sliced along the sagittal plane to better capture the staining of the
electrode track. We now provide the ML coordinates of the sections as white vertical lines in the atlas with corresponding
labels in the histology images. Furthermore, we now highlight the electrode track with a small circle in the histology
images to better visualize what this figure is supposed to show. The legend was also modified.

Line 24 in supplementary file: " Three consecutive sagittal slices (S1, S2, S3) with their coordinates marked by white
lines in the Allen Mouse Brain Atlas are shown (left). The electrode track is highlighted by dashed circles in the histology
images."

• Fig S1K: The green frame in Fig S1D shows channels in PrA. Where are the channels here? In the opposite
SC?

These channels are from the opposite SC in this example. We added a label on Figure S1I and provide extra information
in the legend.

Line 26 in supplementary file: "In the medio-lateral recording configuration, the probe can pass through the SC on the
opposite site, shown here in green."

• Fig S2A: Would considering more PCs improve classification?

We chose PC1 and PC2 for classification as the elbow method identified the optimal number of components to be 2.
Moreover, we have tried using more PCs in the classification and, although increasing the number of PCs used in the
Gaussian mixture model captures more variance of the dataset, moving from 2 to 3 PCs did not help with the
classification. We now added in the method more details on the justification of using PC1 and PC2 and show the PCA
scree panel in Figure S3b.

Line 859: "We used the first two principle components for classification because the elbow method identified the optimal
number of components to be 2 (Fig. S3b)."

• Fig S3A+C: under what conditions/stimulation were firing rates measured? Does it make a difference? What
is spontaneous rate?

We thank the reviewer for pointing out this ambiguity and we now provide additional information in the legend of Fig.
S4. The exact firing rate is stimulus dependent and the spontaneous firing rate is low, at least in anesthetized mice.
The aim of reporting the mean firing rate of the entire recording duration was to characterize basic properties of the
neurons.

Line 88 in supplementary file: "Mean firing rate (FR) across the entire duration (middle left)..."

• Fig S3B+D: how are these measures defined?

We now have added this information in the figure legend.

Line 93 in supplementary file: "RGC axons and SC neurons have similar quality measures. Quality metrics estimated
using the ecephys modules (https://github.com/AllenInstitute/ecephys_spike_sorting)."

• Fig S4C: What was the stimulation protocol?

The visual stimuli presented were sparse noise, moving bar and the chirp stimulus. We now provide this information
in the methods.

Line 759: "In each of these stages, a reduced test stimulus set (15 deg dark/light sparse noise sequences, chirp and
moving bar) was presented to assess the visually driven activity."

• L.944: should relate to panel (F). What is the correlation between non-coupled RGCs for comparison?

Corrected. The correlation between the chirp responses of uncoupled RGCs depends on the functional type of the
RGCs. For example, the correlation would be high for uncoupled RGCs from the same functional type and lower when
the functional types are different.

- Fig S7: Please check spelling and references to various panels (top, left, ...)

Corrected.

- Fig S7B: The left slice looks very different from the right slice. Is this really the correct match in the brain atlas? Please mark OT.

We apologize for showing the wrong slide from the finch atlas in the figure. We now corrected this mistake. We now also mark the optic tectum in the revised version of the panel, which is shown now in Fig. S8b.

- Fig S7D: What is the reason for the gap in the RFs?

We thank the reviewer for raising this question. Unfortunately, we cannot provide a conclusive answer. We see this gap in some recordings but not all. One example is shown in Fig. S8d. Supposedly, it reflects a sudden jump in retinotopy that we cannot fully explain. To illustrate this jump we provide an additional Figure R4. The jump occurs from site ~133 to site ~135. This sudden jump could be related to a gap of RGC axons in the optic tectum around the representation of the optic nerve head/pecten¹⁴ but more work is needed to confirm this hypothesis. We now raise this issue in the results.

Line 431: "Interestingly, we noticed that there was a gap in the receptive fields of the zebra finch (Fig. S8d), which could be related to a gap of RGC axons in the optic tectum around the representation of the optic nerve head⁶⁹."

Figure R4: Zebra finch MUA receptive fields of recordings sites surrounding the gap in retinotopy. The white contour shows the receptive field of recording site 127 and the red contour of recording site 140. The jump in retinotopy from ~133 to site ~135 is visible.

- Fig S8C: similarity measurements are not reported for these examples

This panel has been removed in the revised version to reduce the number of panels.

- State more clearly that most of the results were collected in anaesthetized mice, only Fig S1N,O are from awake mice.

We now state more clearly that the majority of the results are from anaesthetized mice and provide the number of anesthetized and awake mice in the main text and in the methods.

Line 130: "...both in anesthetized (n = 24 mice) and awake mice (n = 3 mice) ..."

Writing style (suggestions):

- Some paragraphs start with a conclusion of the previous paragraph. Other paragraphs start with a statement on what this paragraph is about to show. The authors may wish to stick to one style, preferably the latter.

Modified.

- “Paired recordings” in the title is misleading as only a single probe in one brain area is used for simultaneous axonal and neuronal recordings, which is the great advantage of this new approach.

Changed.

“High-density electrode recordings reveal strong and specific connections between retinal axon mosaics and midbrain neurons”

Conclusion

Sibille et al. present a novel recording technique that can be used in vivo, and is highly useful to the community, enabling to record synaptic input and the postsynaptic neuronal response of the retinocollicular circuit simultaneously. Using this technique, they shed new insights on a long-standing question: What is the connection pattern between RGCs and the SC/OT? While some of the claims need to be further substantiated, and more clarification is needed in parts of the work, the work itself is impressive and adds both to the visual neuroscience field, but also to neuroscience in general due to the novel technique. Accordingly, we highly recommend publishing this work once the issues we raised are addressed.

We appreciate the support and the suggestions from the reviewer. We believe that the reviewer’s comments and suggestions helped us to greatly improve the manuscript and we hope that we could address all raised issues in the revised version.

**Reviewer #3 (Remarks to the Author):**

Comments to the authors

Manuscript number: NCOMMS-21-38786-T

*Title: Strong and specific connections between retinal axon mosaics and midbrain neurons revealed by large scale*
*paired recordings*

*This is an interesting study investigating how the activity of retinal axons is paired with the activity of their target neurons*
*in the superficial layers of the superior colliculus of mice and optic tectum in zebra finches. The results are based on*
*extracellular recordings of single unit activities using pixel probes with many channels. Using thus a very modern*
*electrophysiological approach several experiments have been performed in anaesthetised and awake animals. Many*
*valuable findings are presented:*

*1. A method is presented on how of axonal waveforms, which are coming from the retinal ganglion cells (RGCs) can*
*be separated from the activity of superior colliculus neurons (SC) using extracellular recording. This section includes*
*additional validation experiments with pharmacological treatments.*

*2. Retinotopic organization of visual inputs in the superior colliculus is confirmed.*

*3. Monosynaptic connectivity between RGC axons and SC neurons is validated*

*4. How SC neurons integrate the inputs from RGC axons is investigated*

*5. The representation of spatiotemporal receptive fields in SC are investigated*

*6. The mammalian RGC-SC circuit is compared with the birds optic tectum recorded in zebra finches using a similar*
*approach.*

*Overall, I am truly impressed about the amount of work that the authors have done and I am thrilled by the abilities of*
*the authors to use different types of sophisticated analysis of a very large dataset. However, I also have some serious*
*concerns on the manuscript which need to be addressed. Given that the results contain several types of valuable*
*information my recommendation is a major revision. However, the manuscript should be really revised and partly*
*rewritten and not be published in its current state.*

*My major concern is that the paper lacks to define a main research question. Many experiments have been put together*
*according to the principle "more is more".*

We thank the reviewer for stating their enthusiasm for our work. We regret that the main research question was not
conveyed as clearly as we had thought. While we agree that the data presented in the manuscript is dense, the
manuscript has a defined overarching question: how do neurons in the midbrain integrate RGC inputs in vivo? This
question remained unanswered due to a lack of methodology for recording connected RGC-SC/OT neuron pairs in
vivo. In our work we developed a methodology that allows us to address this question and report the results. We have
now rewritten parts of the manuscript and the abstract to convey the overarching research question more clearly to the
reader.

*I believe that the presented results can be separated in at least three sophisticated research papers.*

*This would allow to describe properly what has been done in a way that a broader readership would understand. This*
*would also allow to address each research question separately, to present the findings accordingly and to discuss all*
*crucial details of these findings in light of existent literature.*

*At present many details are not explained and require a lot of thinking and scrolling back and forth through the*
*manuscript. The discussion of many aspects is short and only superficial. Please don't get me wrong, I truly believe*
*that everything that is presented in this manuscript is logical and that everything makes sense. However, I find that the*
*experiments are not presented efficiently.*

We are honored by the reviewer's suggestion which highlights the quantity of the gathered data and the quality of the
analyses. Based on the valuable suggestions of reviewer #3 we rewrote parts of the manuscript and changed some of
the figures in the main text and in the supplementary materials.

This is already apparent after a brief look at the figures (there are overall 6 Figures, which contains up to 7 subfigures
in the main manuscript. Furthermore, almost each sub-figure is divided in 2-3 additional sub-sub figures. Further 8
Figures with sub- and sub-sub-figures are in the supplement). This is an enormous amount of information. At the same
time the results and methods are not explained and discussed to the needed extend in light of the already existent
literature. The methods need to be reproducible. This is not given at present.

We now reduced the amount of information by removing several graphs and panels in a structured manner. Moreover,
as detailed below, we extensively rewrote the manuscript and extended the method section following the valuable
suggestions from all reviewers.

One suggestion could be to focus this manuscript mainly on the question how SC neurons integrate RGC inputs. In my
view this is the most novel and most interesting part of the study. Two very interesting hypothesis are proposed on how
superior colliculus neurons could integrate retinal ganglion cell inputs (Lines 21-25, summarised in Figure 1B). The
findings need to be discussed in light of these hypothesis. All the rest of the manuscript should be constructed around
this major question only.

We thank the reviewer for this valuable suggestion and we agree that the question how SC neurons integrate RGC
inputs is core to our study. We have now modified the manuscript to discuss the results in light of the hypothesis shown
in Fig. 1b. To that end we added a new section in the discussion "Functional specific retinotectal connection strength"
focusing on this question (Line: 562).

In order to address this question, it was of course needed first to validate methodologically that the activity in the retinal
ganglia cell axons can indeed be separated from the activity of superior colliculus neurons. This can be itself either a
purely methodological paper, that needs to be published first, or it can remain in the current paper as "experiment one".
However, it needs to be discussed very clearly and in light of existent literature to what degree such a separation of
waveforms based on extracellular recordings, without any morphological validation can be used to undoubtably identify
RGC axonal responses. An alternative interpretation would be that such separated waveforms are coming for axonal
activity of other SC internal neurons. They all would be visual and may respond faster than other SC neurons. This
interpretation needs to be excluded. At the moment I am not fully convinced that the used approach is reliable. The
pharmacological treatment for validation is also not very convenient to me. The pharmacological effects are clear, but
the interpretation is vague. The authors are welcome for a rebuttal :)! Explain please all your arguments against my
interpretation in the discussion section of your revised manuscript.

We thank the reviewer for raising this important point, which we are happy to discuss. Our conclusion is based on the
following results and arguments:

To verify that the triphasic waveforms do not originate from SC neurons we have injected muscimol into SC *in vivo*.
muscimol is a GABA_A receptor agonist that silences somatic spiking activity and thus waveforms that remain active
have to originate from outside the SC circuit and are hence axonal (see e.g.¹⁵ for how this approach was used to study
signals from thalamic axons in visual cortex of macaque monkeys *in vivo*). It could be that muscimol did not affect all
parts of the SC circuit and hence some of the waveforms could be from other internal SC neurons that are outside of
the area affected by muscimol. However, this is very unlikely as our muscimol injection suppressed a large area within
SC (Fig. S2). Moreover, if the triphasic waveforms had originated from internal SC neurons then muscimol application
should have significantly reduced the number of active single units with triphasic waveforms in the dataset, because
the SC neurons at the location of the muscimol injection would generate the triphasic waveform at a different location
in SC. This was not the case (Fig. 1f). In addition, the action potential streak that was visible in the antero-posterior
recordings matches well to the anatomy of RGC axons innervating SC^{16,17}, but not to axons of internal SC neurons and
less to axons from cortex¹⁸. Likewise, the spread of the axonal contact field matches well to the anatomy of RGC axonal
arbors in SC and the functional responses of the triphasic waveforms resembles what is known about retinal ganglion
cells, including putative electrical coupling between neighboring RGCs. Moreover, TTX injection into the eye silenced

the activity of the triphasic waveforms which further supports that they originate from the retina. Finally, the triphasic
waveforms that we measure in SC resemble in space and time the electrical signals from single thalamic afferents in
cortex, further^{19,20} supporting that what we measure are afferent axons making synaptic connections onto SC neurons.

We hope that these arguments convince the reviewer. We have added a dedicated section on this topic right at the
start of the discussion, entitled “Recording afferent axons with single high-density extracellular electrodes in vivo”. This
section summarizes the evidence and our reasoning for concluding that the tri-phasic waveforms are RGC axonal
afferents and not SC internal neurons.

Line 461: “We discovered that high-density electrodes capture the electrical activity of RGC axons in the midbrain of
mouse and zebra finch. Several lines of evidence support this conclusion. The pharmacological experiments in the
mouse revealed that the triphasic waveforms remained active after applying muscimol to the SC in vivo (Fig. 1f).
Therefore, the triphasic waveforms cannot originate from neurons within the SC circuit but are signals from long-range
afferent axons⁵⁵. Furthermore, the triphasic waveforms resemble the local field potential signature of single thalamic
axons in cortex measured via thalamic spike-triggered-averaging of cortical local field potentials using paired
recordings^{43,53,71}. Considering this data, we conclude that the triphasic waveforms originate from single afferent axons
making synaptic connections onto midbrain neurons.

Both retina and cortex provide long-range axonal inputs to SC and thus potentially both structures could be
the source of the axonal waveforms. We could observe the streak of the propagating action potential only in the antero-
posterior recordings but not in the medio-lateral recordings (Fig. S1). This observation matches well to the anatomy of
retinal axons innervating SC^{9,72} but less to the anatomy of cortical axons innervating SC⁷³. In addition, the spatial spread
of the axonal contact field is in the range of the anatomical spread of RGC axonal arbors in SC4, the visually evoked
activity of the axonal waveforms resembles what is known about RGCs (Fig. S5a/b), and applying TTX into the mouse
eye abolished the activity of the axonal signals (Fig. 1f). Taken together, we conclude that the triphasic waveforms
measured with the high-density electrode in SC/OT are RGC axons making synaptic connections onto midbrain
neurons.”

The section investigating the monosynaptic connectivity between RGC axons and SC neurons should also remain,
because it's an additional part dealing with connectivity of RGC axons and SC neurons. Thus it fits to the main story
line.

We agree that the monosynaptic connectivity between RGC axons and SC neurons is important for the main story.

The remaining parts of the manuscript should be left out of the present manuscript. They all can become other more
valuable papers. In the present manuscript there is not enough space for presenting and discussing all the findings.
Presenting them only superficially as they are present for now, is not a good solution in my view.

For instance, the finding of retinotopic organisation in optic tectum is nothing novel per se. This has been demonstrated
using optic imaging of intrinsic signals even in zebra finches (Keary et al., 2010, PlosOne). I agree of course that the
electrophysiological validation is needed. However, this is not so important for a high impact manuscript. Keep these
results for another solid paper in a decent journal, where all aspects and details would be discussed.

We thank the reviewer for highlighting that the midbrain of the zebra finch is retinotopically organized. We included this
valuable information in the main text. However, the key aspect of showing the axonal mosaic is that RGC axons
preserve the receptive field mosaic organization in the retina with “single cell precision”. This has not been reported
before in any species. We also included the axonal field organization part in the results as it provides the foundation
for the results of the synaptic organization of the retinotectal circuit. Showing that the RGC axons innervate the SC
circuit with such a high spatial precision is crucial since it allows us to interpret the results on the synaptic and functional
wiring of the retinotectal circuit.

Line 539: “While such an isomorphic representation of the retinotopic map on a larger scale is a known hallmark of the
visual system, including the superior colliculus in the mouse⁸⁹ and the optic tectum in the zebra finch⁹⁰, the single cell
precision of this mapping at the level of the RGC axons in the midbrain has not been shown before.”

The visual field analysis, directional tuning etc. can be also left out of this manuscript

Characterization of the functional connectivity is fundamental for developing mechanistic models of visual processing,
with the majority of studies conducted in the thalamo-cortical circuits, reviewed e.g. in²¹⁻²³. However, data on the
functional wiring of the retinotectal circuit is largely missing and therefore our mechanistic understanding of visual
processing in SC limited. Keeping the results on the functional architecture in the manuscript is, in our opinion, very
valuable for the community. Therefore, we respectfully decline this request.

(btw. what about orientation selectivity?).

We now provide an analysis about the orientation preference of connected RGC-SC neuron pairs in the text of the
results section.

Line 362: "and that connected orientation-selective RGC-SC pairs had similar preferred orientations (mean preferred
orientation difference = $10.50 \pm 8.22^\circ$, $n = 7$ connected pairs)."

At present this is a very superficial presentation of the findings. This part has a lot of potential in particular for a
comparative study of mice and zebra finches. I am a big fan of comparative study of brain functions and evolution of
visual processing. I believe that your data has a lot of potential for comparing between zebra finches and mice in proper
manuscript addressing only this issue. Here you could also consider, that there are some substantial differences in the
organisation of the retinas and optic tecta (e.g. in finches are more layers in the tectum compared to mice, finches have
more photoreceptors etc.). Thus, some differences in the activity in the optic tectum in these two vertebrate models
should be extractable from your data. Take a look at your data considering this and make a great separate paper out
of the data in the end.

We are grateful to the reviewer for seeing further potential in our data and approach. The main aim of the current study
was to show that our novel method is also applicable in zebra finches and reporting on the comparison between basic
properties of the retinotectal circuit in mice and zebra finches. In the current study the main difference we noticed
between mice and zebra finches is the higher spatiotemporal resolution of the zebra finch visual system compared to
the mouse. In the revised version of the manuscript we highlight these differences more prominently in the results and
discussion section.

Line 602: "Our results show that key observations in the mouse SC, e.g. the precise RGC axonal organization and the
functional specificity of connection strength, are also found in the zebra finch optic tectum. This is interesting given that
the spatial resolution of neurons in the optic tectum of zebra finch is higher compared to neurons in the mouse superior
colliculus (Fig. S4)."

Minor comments:

Abstract: please don't use abbreviations in the abstract.

Corrected.

Overall: please reduce the amount of abbreviation to a minimum. There is already so much information in the result
section, don't make it harder for the reader by adding additional abstraction level coded in abbreviations.

Corrected.

Introduction: there are too many aims. Remove paragraph two and specify the main aim in end of the manuscript

We have modified the introduction to improve the focus on the main aim of the study which is the question of how SC
neurons integrate RGC activity. However, we kept parts of paragraph two as its content is crucial for describing the
main objective of this study.

Results:

My suggestion as already mentioned above would be to remove all sections and leave only three following the order:

1. Recording afferent axons and local neurons simultaneously using high-density electrodes.
2. Synaptic organization of the retinocollicular circuit in vivo.
3. Measuring monosynaptic connectivity in vivo at a large scale

However, in any case, since the results are following the introduction, it should be made sure that the reader can understand the basic methodological approach without reading the methods first. A simple claim “see methods” is thus of little use for the reader here. A methodological figure, showing how the stimuli were presented and what kind of stimuli were used would be helpful.

We appreciate the suggestions from reviewer #3 and we have now updated the text to explain the basic methodological approach in more detail right at the start of the results section. Moreover, we also included a new schematic showing the visual stimulation setup and a graphic representation of the visual stimuli in Fig. 1 and Fig. S1.

Line 61: “To study the functional organization of the superior colliculus we used high-density electrodes (Neuropixels probes⁵⁰) to record extracellular neuronal activity in the mouse SC in vivo. The mouse was head-fixed, inside a visual dome⁵¹ that allowed us to present visual stimuli in a large part of the visual field⁵² (Fig. 1c). To record neuronal activity in the SC we targeted the visual layers of SC with a tangential recording configuration that places hundreds of recording sites within the optical layer and superficial gray layers of SC52 (Figs. 1c and S1n). To characterize the visual response properties of the recorded neurons, we presented light and dark sparse noise, a full-field chirp stimulus and moving bars (Figs. 1d and S1c).”

All needed details that would allow to understand the results should be provided. This would make the manuscript better accessible for a larger public.

We have integrated more details in the main text and method section in the revised version.

Overall to many graphs, and even more are in supplement as I already mentioned above. Moreover, some sub-sub figures are very small. See e.g. figure 1B, 1C or figure 5A. I am glad that I have a PDF and can zoom in on my computer monitor. I would not be able to see anything in a printed version. If you have less results sections, you would have more space for larger images.

We are very grateful for this criticism which helped us to make the manuscript and figures more concise. We now removed a considerable number of graphs from the figures.

Line 328: Please explain (or show in a figure) what kind of a sparse noise stimulus was used. I don’t want to read the Paper “15” to extract this information which would allow me to understand your paper.

We now show more details of the sparse noise stimuli in Figures 1d/S1c and provide more details about those stimuli in the method section.

Line 696: “Sparse noise for receptive field mapping: To characterize receptive fields, we presented sparse noise targets of varying size and contrast polarity for 100 ms in a pseudo random manner on a grid of 36x22 positions. The grid spacing was 5 deg and the grid covered 180x110 deg of the visual field. The sparse noise targets were either dark (on light background) or light (on dark background) to characterize the ON and OFF receptive fields. Because the number of possible grid positions was very high, we presented multiple sparse noise targets simultaneously but in non-overlapping positions at a given time to increase the number of repeats per grid position¹¹¹. We used three different target sizes presented in separate sequences with varying number of targets per frame and trials per position (5 deg targets = 6 targets per frame and 50 trials per position; 10 deg targets = 4 targets per frame and 30 trials per position; 15 deg targets = 2 targets per frame and 20 trials per position). The sparse noise sequences were generated once, saved and the same sequences reused across the different experiments.”

Line 348: What is a chirp stimulus? Please explain.

We now provide more details on the chirp stimulus in the main text and methods. We also show the stimulus more prominently in Fig. 1d.

Line 708: "Full-field chirp: To characterize the contrast polarity, temporal frequency as well as contrast response properties we presented a full-field chirp stimulus¹. The full-field stimulus varies in brightness: it starts with a gray background and several light decrement and increment steps (~2.18 s black, ~3.28 s white, ~3.28 s black, 2.18 s gray) followed by sinusoidal intensity modulations with increasing frequency (0.5 Hz to 11 Hz) at full contrast (8.75s) and increasing contrast (0 to 100 %) at 0.4 Hz (8.75 s) and ending with 2.18 s gray background."

Discussion:

Is the methodological validation your main finding? Then it should be a methodological paper. But then it would not be a suitable paper for nature communications. I would not put this part in front of the discussion and I would also not limit the discussion to only advertise your method so much here (Btw. pixel probes are commercially available, at least this part is not so novel). Instead I would suggest to discuss properly the validity of the method for measuring "axonal synaptic contact fields" in your extra cellular recording approach. I am not sure though, if such a conclusion can be made at all without a morphological validation study using e.g. calcium imaging and viral tracing. But you can try to convince also readers like me with a proper discussion.

We thank the reviewer for the helpful suggestion for restructuring the discussion. We agree with the reviewer that Neuropixels probes are commercially available and that the simple usage of these probes is not novel. However, a crucial part of our work is the discovery that one can record afferent axonal contact fields waveforms with these probes in vivo. Since the study crucially relies on the ability to measure the axonal contact fields in vivo with a single Neuropixels probe we have now added a new section "Recording afferent axons with single high-density extracellular electrodes in vivo" right at the start of the discussion to highlight the validity of the method. Please refer to lines 460:486 in the revised manuscript.

Methods:

It should be clarified why 95 mice were needed but only 7 zebra finches.

We apologize that the number of mice used in the study was wrongly reported in the methods. The correct number is $n = 24$ anesthetized mice and $n = 3$ awake mice. The number of zebra finches is lower ($n = 7$ zebra finches) because when we started the zebra finch experiments the Neuropixels method was already well established and tested in mice. The numbers have been corrected in the revised manuscript.

I still don't fully understand which setup was used for which experiments. For which of the experiments awake animals were needed and how many. Was the same setup used for presenting visual stimuli to zebra finches and mice?

The awake mice experiments were included in this study solely to show that it is possible to record RGC axons also under awake condition ($n = 3$ mice). The majority of mice and zebra finch experiments were conducted in the same visual dome setup. A subset of experiments was conducted using an LCD screen due to spatial constraints with additional experimental equipment (e.g. injector in the pharmacology experiments). In the revised version we have updated the method section and figures to explain the experimental setups and usage in a clearer way.

I think the methods should be organised in a more efficient way, presenting each experiment independently in a concise and clear way.

We modified the methods substantially in the revised version and now include independent sections for the mice and zebra finch experiments in case major differences in the experimental design exist, e.g. in the case of the properties of the visual stimuli.

Line 695: "Visual stimuli in the mouse experiments"

Line 719: "Visual stimuli in the zebra finch experiments"
Lines 570-571: I suppose this is an analgetic? Please add this information
We have added this information.
Line 655: "The analgesic metamizole (200 mg/kg, Zentiva-Novaminsulfon) was administered in drinking water after
head post implantation for a recovery period of 3 days."
Lines 580-589: "Recordings..." this part should go in the part "Electrophysiological recordings" starting from Line 601
We integrated lines 580-599 in the section "Electrophysiological recordings", starting line 725.
Lines 589-599: "Histology..." this part should be after the pharmacological application section before data analysis.
We moved the "Histology" section to the suggested place.
Lines 626-643: "Visual stimulation" this whole section needs to be overworked. Crucial details are missing. Was the
same setup used for anaesthetised zebra finches and mice?
The same visual dome setup was used for the experiments in mice and zebra finches. All mice experiments were
conducted at the Charité Berlin while the experiment on zebra finches were conducted at the MPI Seewiesen. For the
experiments on zebra finches we moved the setup from Berlin to Seewiesen. We have now rewritten this section to
explain the setups more clearly.
Line 674: "Visual stimulation..."
What does it mean either a calibrated screen or projector? For which of the experiments did you use a screen and for
which a projector? You need to be more specific.
We thank the reviewer for pointing to this unclarity. The majority of the experiments were conducted in the visual dome
setup using a projector. The pharmacology experiments were done with a regular LCD screen because the injector
system did not fit into the visual dome. The awake recordings were conducted using an LCD screen to track the pupil
via cameras. A subset of zebra finch experiments was conducted using an LCD screen because the visual dome setup
was not available at that time. Both the LCD screen and the projector in the visual dome setup were gamma corrected
using the ColorCALMKII sensor (Cambridge Research System) and the visual stimuli presented using the PsychoPy
software. We have modified the corresponding text in the revised manuscript.
Line 678: "Visual stimuli were presented in a spherical visual dome (EBrilliantAG, IP44, diam = 600 mm)⁵¹ using a
projector (NEC ME331W, refresh rate = 60 Hz, mean luminance = 110 cd/m², Gamma corrected) to cover a large part
of the visual field."
Line 688: "In a subset of experiments, we used an LCD display (Dell S2716DG, refresh rate = 120 Hz, mean luminance
= 120 cd/m², Gamma corrected) instead of the visual dome because additional equipment required more space, e.g.
the injector during the pharmacological experiments or the camera for pupil tracking in the awake experiments. "
It is not clear to me what kind of stimulation was presented for which species and under which conditions. A figure of
the setup/setups including images of the used visual stimuli would be very useful. Please keep in mind that the crucial
parts of the experiments have to be reproducible based on the information provided in the methods section.
We now included schematic/photos for the setups in the main and Figs. 1 and S1 and provide more details in the
method section.

Line 674: "Visual stimulation..."

Line 656: What do you mean by (n=3/6) ? Is it 3 or 6?

Corrected. It should have been n=3 successful double-injection pharmacological experiments.

Line 714-721: The logic of this approach for detection of axonal efferents needs to be explained better.

We modified the text in the paragraph "Detecting axonal contact field waveforms in Neuropixels datasets" to better explain the logic.

Line 865: "Detecting axonal contact field waveforms in Neuropixels datasets: The standard Kilosort2 parameters are sufficient to detect axonal contact field waveforms in Neuropixels datasets. Importantly, during the curation in Phy2, the rejection criteria such as "multiple spatial peaks" and "too large spread"⁴⁷ should be minimized to increase the number of identified axonal contact field waveforms in the dataset. A key factor for recording axonal signals is a well-placed Neuropixels probe in the SC/OT tissue. To optimize the targeting and the yield of axonal signals, we adapted a semi-online approach that allows the assessment of whether a given insertion contains axonal contact field waveforms. To that end, we recorded ~5 minutes of neuronal activity and spike-sorted this short dataset with Kilosort2. During the sorting process, Kilosort plots the detected waveforms using the function "make_fig.m", which allows visually inspection of the waveform types in the dataset. To facilitate the identification of axonal contact field waveforms in this plot, we modified the "make_fig.m" code such that the waveforms are sorted by the value around 1.5 ms (which is the time of the second trough in the RGC waveforms). This semi-online analysis allows assessment of whether axonal contact field waveforms are in the dataset, within a few minutes. It can thus be used during a recording session such that if no axonal waveforms are identified the Neuropixels probe can be relocated to a different position. The modified "make_fig.m" is available on our GitHub repository (<https://github.com/KremkowLab/Axon-on-Neuropixels-in-Kilosort>).

Line 746-760: This sounds really fascinating and I am really trying hard to understand how it is possible to separate signals coming from RGC axons from those of SC neurons. Are you sure that these are RGC and SC neurons without any morphological confirmation? I don't doubt that there is a reasonable logic behind this approach. However, this part needs to be described in a way, that also other people can understand.

We thank the reviewer for stating that our results are fascinating, and we apologize that our description was not clear. We now modified the results section "Recording RGC axons and SC neurons with high-density electrodes in the mouse SC" and added a new discussion section "Recording afferent axons with single high-density extracellular electrodes in vivo" to specifically address this question. Please refer to lines 460:486 for the discussion regarding this point.

Lines 762-786: this section is very hard to read because too many abbreviations were used. I would suggest in general to avoid abbreviation whenever it is possible through the whole manuscript. It is possible to write axonal field instead of AF and receptive field instead of RF etc. Your paper will become more readable.

We reduced the number of abbreviations in the entire manuscript. We mainly kept the abbreviations for retinal ganglion cells (RGC), superior colliculus (SC) and optic tectum (OT).

Lines 810-822: I think that this "Receptive fields" section should be better placed before "...retinal ganglion cells mosaics..." section in line 761. Moreover,... (you already probably know what I will say now :))... also this section needs a better explanation to make it understandable for more general public and to be reproducible.

We thank the reviewer for this valuable suggestion. We now placed this section before the section about retinal ganglion cell mosaics and we updated the text to make it more understandable.

Lines 826-827: What is a "Mises function" ?

A “von Mises function” is a circular normal distribution. It was introduced by Swindale et al. (2003)²⁴, to fit orientation
and direction tuning curves of neurons in visual circuits. We modified the methods to make this clearer to the reader.

Line 1023: “The von Mises function is a circular normal distribution and the sum of two von Mises functions allows fitting
direction and orientation tuning curves and extracting preferred direction (PD) or orientation (PO)¹²⁰.”

**Supplements:**

Figure S7B: consider that you penetrated several layers of optic tecta in zebra finches. While the outer layers are
retinotopically organized, the deeper layers, especially the output layers should be less retinotopic. Instead, several
types of functionally separated units should be more abundant in the deeper layers.

We thank the reviewer for this interesting comment about the difference between outer and deeper layers in the optic
tectum. Revealing difference between these different layers is very interesting and relevant for reaching a more detailed
understanding of the visual processing of the optic tectum. However, we feel that this question is beyond the scope of
this study.

**References**

- 1. Shamash, P., Carandini, M., Harris, K. D. & Steinmetz, N. A. A tool for analyzing electrode tracks from slice
histology. *Biorxiv* 447995 (2018) doi:10.1101/447995.
- 2. Steinmetz, N. A., Zatka-Haas, P., Carandini, M. & Harris, K. D. Distributed coding of choice, action and
engagement across the mouse brain. *Nature* 576, 266–273 (2019).
- 3. Rosón, M. R. *et al.* Mouse dLGN Receives Functional Input from a Diverse Population of Retinal Ganglion Cells
with Limited Convergence. *Neuron* 102, 462–476.e8 (2019).
- 4. Baden, T. *et al.* The functional diversity of retinal ganglion cells in the mouse. *Nature* 529, 345–350 (2016).
- 5. Chandrasekaran, A. R., Shah, R. D. & Crair, M. C. Developmental homeostasis of mouse retinocollicular
synapses. *The Journal of neuroscience : the official journal of the Society for Neuroscience* 27, 1746–1755 (2007).
- 6. Usrey, W. M., Reppas, J. B. & Reid, R. C. Specificity and strength of retinogeniculate connections. *Journal of*
*Neurophysiology* 82, 3527–3540 (1999).
- 7. Alonso, J.-M., Usrey, W. M. & Reid, R. C. Rules of connectivity between geniculate cells and simple cells in cat
primary visual cortex. 21, 4002–4015 (2001).
- 8. Lien, A. D. & Scanziani, M. Cortical direction selectivity emerges at convergence of thalamic synapses. *Nature*
558, 80–86 (2018).
- 9. Zhuang, J. *et al.* An extended retinotopic map of mouse cortex. *Elife* 6, e18372 (2017).
- 10. McInnes, L., Healy, J., Saul, N. & Großberger, L. UMAP: Uniform Manifold Approximation and Projection. *J Open*
*Source Softw* 3, 861 (2018).
- 11. Cleland, B. G., Dubin, M. W. & Levick, W. R. Simultaneous recording of input and output of lateral geniculate
neurones. *Nature: New biology* 231, 191–192 (1971).
- 12. Reid, R. C. & Alonso, J.-M. Specificity of monosynaptic connections from thalamus to visual cortex. *Nature* 378,
281–284 (1995).
- 13. Alonso, J.-M., Usrey, W. M. & Reid, R. C. Precisely correlated firing in cells of the lateral geniculate nucleus.
*Nature* 383, 815–819 (1996).
- 14. Puelles, L., Martinez, S. & Martinez-De-La-Torre, M. The locus of optic nerve head representation in the chick
retinotectal map lacks a retinal projection. *Neurosci Lett* 79, 23–28 (1987).
- 15. Chatterjee, S. & Callaway, E. M. Parallel colour-opponent pathways to primary visual cortex. *Nature* 426, 668–
671 (2003).
- 16. Huberman, A. D. *et al.* Architecture and Activity-Mediated Refinement of Axonal Projections from a Mosaic of
Genetically Identified Retinal Ganglion Cells. *Neuron* 59, 425–438 (2008).
- 17. Triplett, J. W. *et al.* Retinal input instructs alignment of visual topographic maps. *Cell* 139, 175–185 (2009).
- 18. Benavidez, N. L. *et al.* Organization of the inputs and outputs of the mouse superior colliculus. *Nat Commun* 12,
4004 (2021).
- 19. Bereshpolova, Y., Hei, X., Alonso, J.-M. & Swadlow, H. A. Three rules govern thalamocortical connectivity of fast-
spike inhibitory interneurons in the visual cortex. *Elife* 9, e60102 (2020).

- 20. Swadlow, H. A. & Gusev, A. G. The Influence of Single VB Thalamocortical Impulses on Barrel Columns of Rabbit
Somatosensory Cortex. *J Neurophysiol* 83, 2802–2813 (2000).
- 21. Kremkow, J. & Alonso, J.-M. Thalamocortical Circuits and Functional Architecture. *Annu Rev Vis Sc* 4, 1–23
(2018).
- 22. Niell, C. Cell Types, Circuits, and Receptive Fields in the Mouse Visual Cortex. *Annual Review of Neuroscience*
(2015) doi:10.1146/annurev-neuro-071714-033807.
- 23. Niell, C. M. & Scanziani, M. How Cortical Circuits Implement Cortical Computations: Mouse Visual Cortex as a
Model. *Annu Rev Neurosci* 44, 1–30 (2021).
- 24. Swindale, N. V., Grinvald, A. & Shmuel, A. The Spatial Pattern of Response Magnitude and Selectivity for
Orientation and Direction in Cat Visual Cortex. *Cereb Cortex* 13, 225–238 (2003).

REVIEWER COMMENTS

Reviewer #1:

The manuscript by Sibille et al. now titled “High-density electrode recordings reveal strong and specific connections between retinal axon mosaics and midbrain neurons” has undergone extensive changes, significantly improving the quality of the manuscript and depiction of the results. We especially appreciate the revision of the oversimplifying statements, adding of important references to the literature and the additional extensive analysis undertaken to address our concerns.

We approve the re-submitted manuscript with its additional analysis. However, there is one issue remaining that needs addressing in the discussion.

Major:

1. We still disagree with this statement in the discussion: Line 508: “Furthermore, we could identify multiple (3-5) converging RGC inputs to SC neurons (Fig. 3f), which is in the range (~5) of the reported number of converging RGC neurons onto SC neurons (Chandrasekaran et al. 2007). Thus, our approach can adequately sample the presynaptic RGC pool of individual SC neurons, although such a high sampling is achieved only in a subset of SC neurons (Fig. 3f).”

The average number of 5 presynaptic RGCs comes from a paper using slice electrophysiology of 7 cells with a minimal stimulation protocol to measure NMDAR-mediated events (Chandrasekaran et al. 2007). This provides a lower bound of the number of convergent inputs, which is not a good estimate of the total number of RGC inputs. In addition, anatomical evidence of the number of presynaptic RGCs does not yet exist. We still suggest rewriting this part of the discussion to make clear that under sampling is likely.

Minor:

1. Statistics for skewed distribution should have asymmetric measures of variance/confidence, unless there is a specific distribution being described (e.g. Poisson). We suggest reporting the interquartile intervals (or confidence interval) of the upper and lower bounds of the skewed distributions rather than +/- SD, e.g., for distances (fig 3) and efficacy & contribution (fig 4).

2. Figure S6: It is great to see the clustered responses of all recorded SC neurons. However, it would be easier to associate “d” with “e” if clusters were sorted in the same way as in “e”.

Reviewer #2:

Peer-review of “High-density electrode recordings reveal strong and specific connections between retinal axon mosaics and midbrain neurons” by Sibille et al.

Summary

The authors have addressed our comments very well and we think that the manuscript has improved significantly. The only outstanding issue we have is with regards to the use and analysis of “mosaics”, which we will address below. Other than that we have only minor comments, which should be easy to resolve. Therefore our recommendation is that the manuscript should be published after further relatively minor corrections.

Major comment

In our view, the evidence for retinal mosaics in the SC is still not convincing, however this is not necessary as it is already known that mosaics are a hallmark of retinal ganglion cell types. All the authors need to do here is to quantify how precise the mapping between RFs and AFs of RGC axons is in the SC.

To convincingly show that AFs of RGCs form mosaics in the same way as in the retina, one would need to determine the distance to the closest neighbours and the angles between the closest three neighbours for a sufficiently large sample of AFs of the same RGC type. Given that the sampling of AFs from neighbouring RGCs is highly incomplete using thin Neuropixels probes AND that AFs of RGCs from multiple functional types are pooled together (see l. 976), the expected distributions of distances and angles between neighbours change drastically. In this case, it is no longer expected

that angles cluster around 60 degrees, nor that distances are around the size of one RF, as RFs from multiple RGC types do not form hexagonal mosaics. That the authors still find mostly angles of 60 degrees and relatively large distances (Fig S5d) may point to limitations of the recording method (e.g. limited sampling of AFs that occupy the same space in SC) and analysis method (determining location of RF and AF centres).

To determine the precision of spatial mapping of RGC AFs in the SC, it seems unnecessary to perform separate analyses for each RGC type. On the contrary, the more RGCs are included, the more convincing the results will be.

We suspect that the median distance between RF and AF locations increases with increasing numbers of recorded RGC axons, so that recordings with fewer RGCs would underestimate the distance. We therefore suggest to plot the RF-AF distance versus the number of recorded RGC axons (one point per recording, or using subsampling of RGCs from the recordings). Does the median distance converge to a stable value with increasing numbers of RGCs?

In summary, we suggest to focus Fig 2 on the precise match between RFs and AFs (regardless of RGC type), rather than on retinal mosaics. Practically the results of Figure S5d-h could be incorporated into the main figure. We also suggest to remove "mosaic" from the title of the paper, which in our minds makes the message of the title much clearer as the paper shows specific connections between single RGCs and SC neurons, rather than between retinal mosaics and SC neurons.

Minor comments

- Some paragraphs are very long (e.g. starting l. 273). Please consider splitting.
- State which statistical test was used whenever reporting p-values
- Ll. 114: "Being able to ... is sufficient to ...". This statement is not convincing / not logically sound. Just because RFs of single spike clusters are close to each other doesn't mean that the spikes actually originate from individual RGCs nor that the RGCs are actual neighbours in the retina. Also neighbourhood and isolation of individual units seem to be unrelated issues.
- When using a monitor instead of the dome, was sphere mapping used to account for the change of distance between eye and screen across the extend of the screen?
- L. 171: what is meant by "electrode pitch"?
- Fig 3a: what do the 3 contour lines for each RF show?
- The authors seem to use two terms to refer to the same concepts: efficacy and connection strength, as well as contribution and coupling strength. It would be less confusing if only one of the two terms were used. (See for example ll. 308 and ll. 421)
- Ll. 321: the random sampling procedure is not clear. What are 1st data/shuffled versus 2nd data/shuffled? Instead of reporting how many shuffles were significantly different it may be better to: (1) determine median efficiency/contribution of strongest/2nd strongest connection in data, (2) determine distribution of median efficiency/contribution of 1st/2nd connection from 1000 times shuffling, (3) determine whether median of data falls into 2.5 to 97.5% percentile interval of shuffled distribution
- Two sentences starting in l. 365 ("Our results support...") sound like the direct opposites of each other. Please clarify what you mean (especially in the 2nd sentence).
- L. 382: use words instead of $r_{(SD)}$, ...
- Fig 5a: specify which data come from RGC, which from SC neurons
- Fig 5c+d: CCGs need to show where 0 is on x-axis
- Fig 5c: RFs of RGCs largely overlap, which seems to contradict previous claims that RFs of the same RGC type are not overlapping.
- L. 430: "we noticed..." unclear. Better: gap in RF positions along the probe (or similar)
- L. 511: more suitable reference to add here would be Schroeder et al. (2020, Neuron), which shows functional imaging of RGC boutons in the SC rather than the LGN
- L. 852: explain the "elbow method"
- Ll. 855 still unclear. What window are you referring to? Do you mean "upsampled" to 10 times the given sampling rate using linear interpolation?
- L. 858: what does "time-sliced" mean?
- Features used to classify spikes into retinal versus SC: As these features will be crucial for other researchers to replicate the method, it would be very helpful to describe how each feature was

determined in more detail (possibly in a table). For example, how were slopes determined? Is it: the slope between points of the waveform that cross: $A1-0.1*(A1-A2)$ and $A1-0.9*(A1-A2)$ for slope S1? How was half height of peaks determined? Height of W2 in SC waveform (Fig S3a) seems lower than half height of A3.

- Ll. 859: unclear how slopes were determined. “Percentile” refers distributions.
- Ll. 860: Does this mean that all 14 features were determined separately for each channel and then averaged across channels?
- Ll. 940: for a convincing argument, we would need to see the measures (RF for 5 degree versus 10 or 15 degree stimulus) for the whole population not just a single example. For large RF sizes, this method may underestimate the real size.
- Fig S1h: the opposite SC is not shown in green as stated in the caption
- Fig S6e: What do responses to bar stimuli show? Concatenated response vector for all bar directions? Or average across all directions?

How were classes sorted? Why not show the same order of classes in d and e?

Missing/wrong references:

- L. 84: add references to papers showing RGC axon innervation in SC
- L. 85: wrong ref. to Fig 1d
- L. 91: Fig S2l does not exist
- L. 375: Fig 5e
- L. 377: Fig 5f
- L. 889: Fig S6
- Ref 2 on l. 98 in suppl. material

Spelling/grammar:

- L. 113: RCGs
- Ll. 202: grammar at end of sentence seems off
- Ll. 315 should probably read: “only a few RGCs contributed strongly to the spiking of single postsynaptic SC neurons” (or similar)
- L. 355: similarly
- L. 557: pf

Reviewer #3:

The authors carefully addressed all the comments raised by the Reviewers. Now the manuscript reads like a completely different paper and I have no further comments. I really like the current version of the manuscript and recommend it for a publication.

**Reviewer #1:**

The manuscript by Sibille et al. now titled “High-density electrode recordings reveal strong and specific
connections between retinal axon mosaics and midbrain neurons” has undergone extensive changes,
significantly improving the quality of the manuscript and depiction of the results. We especially appreciate
the revision of the oversimplifying statements, adding of important references to the literature and the
additional extensive analysis undertaken to address our concerns.

We approve the re-submitted manuscript with its additional analysis. However, there is one issue remaining
that needs addressing in the discussion.

We are excited about the overall positive assessment and we have addressed the one remaining issue.

**Major:**

1. We still disagree with this statement in the discussion: Line 508: “Furthermore, we could identify
multiple (3-5) converging RGC inputs to SC neurons (Fig. 3f), which is in the range (~5) of the reported
number of converging RGC neurons onto SC neurons (Chandrasekaran et al. 2007). Thus, our approach
can adequately sample the presynaptic RGC pool of individual SC neurons, although such a high sampling
is achieved only in a subset of SC neurons (Fig. 3f).”

The average number of 5 presynaptic RGCs comes from a paper using slice electrophysiology of 7 cells
with a minimal stimulation protocol to measure NMDAR-mediated events (Chandrasekaran et al. 2007).
This provides a lower bound of the number of convergent inputs, which is not a good estimate of the total
number of RGC inputs. In addition, anatomical evidence of the number of presynaptic RGCs does not yet
exist. We still suggest rewriting this part of the discussion to make clear that under sampling is likely.

We agree with the reviewer that this number is likely a lower bound and that anatomical evidence of the
number of presynaptic RGCs is still missing. We now acknowledge this point in the discussion and provide
suggestions for future experiments.

Line 520: “Furthermore, we could identify multiple (3-5) converging RGC inputs to SC neurons (Fig. 3f),
which is in the range (~5) of the reported number of converging RGC neurons onto SC neurons estimated
electrophysiologically in vitro⁸⁰. Thus, based on this number our approach can sample a fair amount of the
presynaptic RGC pool of individual SC neurons, although such a high sampling is achieved only in a subset
of SC neurons (Fig. 3f). However, because the anatomical evidence of the number of presynaptic RGCs of
SC neurons is still an open question our numbers represent a lower bound and an under sampling is likely,
in particular for weak connections that do not reliably evoke spiking activity in SC neurons.”

**Minor:**

1. Statistics for skewed distribution should have asymmetric measures of variance/confidence, unless
there is a specific distribution being described (e.g. Poisson). We suggest reporting the interquartile
intervals (or confidence interval) of the upper and lower bounds of the skewed distributions rather than +/-
SD, e.g., for distances (fig 3) and efficacy & contribution (fig 4).

We now report the median and the interquartile range of the distributions in the revised version of the
manuscript.

2. Figure S6: It is great to see the clustered responses of all recorded SC neurons. However, it would
be easier to associate “d” with “e” if clusters were sorted in the same way as in “e”.

We now sorted the clusters in the same way in Figure S6d and S6e.

Reviewer #2:

Peer-review of “High-density electrode recordings reveal strong and specific connections between retinal axon mosaics and midbrain neurons” by Sibille et al.

Summary

The authors have addressed our comments very well and we think that the manuscript has improved significantly. The only outstanding issue we have is with regards to the use and analysis of “mosaics”, which we will address below. Other than that we have only minor comments, which should be easy to resolve. Therefore our recommendation is that the manuscript should be published after further relatively minor corrections.

Major comment

In our view, the evidence for retinal mosaics in the SC is still not convincing, however this is not necessary as it is already known that mosaics are a hallmark of retinal ganglion cell types. All the authors need to do here is to quantify how precise the mapping between RFs and AFs of RGC axons is in the SC. To convincingly show that AFs of RGCs form mosaics in the same way as in the retina, one would need to determine the distance to the closest neighbours and the angles between the closest three neighbours for a sufficiently large sample of AFs of the same RGC type. Given that the sampling of AFs from neighbouring RGCs is highly incomplete using thin Neuropixels probes AND that AFs of RGCs from multiple functional types are pooled together (see I. 976), the expected distributions of distances and angles between neighbours change drastically. In this case, it is no longer expected that angles cluster around 60 degrees, nor that distances are around the size of one RF, as RFs from multiple RGC types do not form hexagonal mosaics. That the authors still find mostly angles of 60 degrees and relatively large distances (Fig S5d) may point to limitations of the recording method (e.g. limited sampling of AFs that occupy the same space in SC) and analysis method (determining location of RF and AF centres).

We agree with the reviewer that performing the angle analysis on all recorded RGCs across all distances is not appropriate (Wässle et al., 1981) and we apologize that we did not communicate the details of our analysis appropriately, likely causing a misunderstanding.

The panel Figure S5d shows the analysis of the angles of only one example, the one shown in Figure S5d. In this example we had sufficient number of RGCs within the local neighborhood and with similar response properties such that analyzing the angles between neighboring RGCs was possible. Moreover, the angles were estimated by the Delaunay triangulation and Voronoi tessellations (Zhan and Troy, 2000), which characterizes the nearest neighbor angles. This step was done specifically to not include angles across all distances, for the reasons the reviewer mentioned. This information was provided in the original manuscript (main text and figure legend) but unfortunately only shown graphically in the schematic of Figure S5d but not incorporated into the legend of Figure S5 in the revised manuscript. We are very sorry for this lack of information and for the extra work this has caused.

We would like to keep the panel (Fig. S5d in revision 1) in the supplementary figure, if possible, because we think it will inspire future studies investigating the spatial mapping between the RGC receptive fields and RGC axons within SC in more detail. To explain these graphs better to the reader, we now incorporate these graphs into the part of Fig. S5 showing the example where the RF/AF data is from (new Fig. S5c). We also included the information on the Delaunay triangulation and Voronoi tessellations into the legend of Fig. S5c and we have modified the discussion to motivate future experiments.

Lines 109 in supplementary materials: “The nearest neighbor angles between RFs and AFs were estimated
by Delaunay triangulation and Voronoi tessellations.”

Lines 572: “Moreover, the small width of the Neuropixels probe only provides a narrow sampling of neuronal
tissue in two dimensions. While several important properties of neighboring RGC axons could be revealed
using this method (Figs. 2 and S5), characterizing the full complexity of the three-dimensional organization
of RGC axons within SC requires further investigations. Two-photon calcium imaging of RGC axons in SC
would be well suited to further deepen our understanding of the functional organization of RGC axons in
SC in 2D and potentially also 3D using multi-plane imaging⁹⁶, in particular when combined with transgenic
mouse lines that label genetically identified single RGC types².”

To determine the precision of spatial mapping of RGC AFs in the SC, it seems unnecessary to perform
separate analyses for each RGC type. On the contrary, the more RGCs are included, the more convincing
the results will be.

We suspect that the median distance between RF and AF locations increases with increasing numbers of
recorded RGC axons, so that recordings with fewer RGCs would underestimate the distance. We therefore
suggest to plot the RF-AF distance versus the number of recorded RGC axons (one point per recording, or
using subsampling of RGCs from the recordings). Does the median distance converge to a stable value
with increasing numbers of RGCs?

We apologize for not explaining the rational and details about this analysis clearly enough. We agree with
the reviewer that performing this analysis with only a few RGCs could be problematic because the alignment
between RF and AF with only a few RGCs could underestimate the RF/AF distances. Therefore, we have
performed this alignment step/analysis only with recordings for which we had a decent number of RGCs
available. This situation is similar to my previous work (Kremkow et al., 2016) where I studied the spatial
organization of cortical receptive fields from the left and right eye, which also required an aligning step, due
to differences in the positions of the eyes. Also, here, it was important to perform the alignment across a
large population of receptive fields to avoid underestimating the distances and differences between
receptive field from the left and right eye. Therefore, we had selected recordings with a decent number of
RGCs ($n \geq 20$ RGCs) for the RF-AF distance analysis. We unfortunately did not convey this rational and
information in the manuscript, which we now do in the revised version.

Line 1017: “Important to note: this alignment step requires a population of RGCs to avoid underestimating
the distances between RFs and AFs ($n \geq 20$ RGCs in this study) but otherwise does not change the
geometric organization of the axonal field mosaic, it only scales and rotates their positions.”

In addition, we have performed the suggested
analysis using random subsampling of RGCs from
the recordings (10 repeated sampling per
recording). This analysis confirms that the median
distance converges with increasing number of
RGCs (Figure 1 in this letter).

Figure 1. Median RF-AF distance as a function of included RGCs. RGCs were randomly subsampled. Black dots = individual samples. Magenta = mean +/- std.

In summary, we suggest to focus Fig 2 on the precise match between RFs and AFs (regardless of RGC
type), rather than on retinal mosaics. Practically the results of Figure S5d-h could be incorporated into the
main figure.

In the revised version we followed the suggestion and have focused on the precise match between RFs
and AFs and incorporated results shown in Figure S5d-h in the main Figure 2 as follows:

- - In Figure 2b we now show the RFs and AFs of an example irrespective of the RGC types.
 - - In Figure 2c we then zoom in and show the precise match between RFs and AFs from a few examples
from RGCs with similar response properties. The example on the left is a subset of RGCs from the
example we now show in Figure 2b. The purpose of showing these examples is to visually convey the
precise match between RFs and AFs to the reader, which is particularly easy to grasp in the examples
from RGCs with similar properties. We do not discuss these examples in the context of mosaics in the
revised manuscript.
 - - In Figure 2e-g we now provide the results of the analysis irrespective of the RGC type.

We also suggest to remove “mosaic” from the title of the paper, which in our minds makes the message of
the title much clearer as the paper shows specific connections between single RGCs and SC neurons,
rather than between retinal mosaics and SC neurons.

We have removed the term mosaic from the title. The title is now: “High-density electrode recordings reveal
strong and specific connections between single retinal ganglion cells and midbrain neurons”

**Minor comments**

- • Some paragraphs are very long (e.g. starting l. 273). Please consider splitting.

We have split several paragraphs in the revised version.

• State which statistical test was used whenever reporting p-values

We now provide the statistical test whenever reporting p-values.

• LI. 114: “Being able to ... is sufficient to ...”. This statement is not convincing / not logically sound.
Just because RFs of single spike clusters are close to each other doesn’t mean that the spikes actually
originate from individual RGCs nor that the RGCs are actual neighbours in the retina. Also neighbourhood
and isolation of individual units seem to be unrelated issues.

We have removed this sentence in the revised version of the manuscript.

• When using a monitor instead of the dome, was sphere mapping used to account for the change
of distance between eye and screen across the extend of the screen?

No sphere mapping was used in the LCD monitor because the LCD monitor was mainly used in the
pharmacological and awake experiments. We have added this information in the method section.

Line 713: “In a subset of experiments, we used an LCD display (Dell S2716DG, refresh rate = 120 Hz,
mean luminance = 120 cd/m², Gamma corrected but without sphere mapping)”

• L. 171: what is meant by “electrode pitch”?

The electrode pitch is the distance between recording sites. This term was provided in the Neuropixels user
manual. To improve the readability of the text we have now changed this term to “recording site distance”.
Line 174.

• Fig 3a: what do the 3 contour lines for each RF show?

The 3 contours show different levels of the RFs. In the revised version we have harmonized the way RFs
are shown and now only show one threshold level.

• The authors seem to use two terms to refer to the same concepts: efficacy and connection strength,
as well as contribution and coupling strength. It would be less confusing if only one of the two terms were
used. (See for example ll. 308 and ll. 421)

To improve the readability of the text we have modified the text in the revised version of the manuscript.
For example:

Line 284: “we estimated the strength of the connection by the efficacy measure and the coupling of the
connection by the contribution measure”

Line 314: “Across the population, we discovered a log-normal distribution of connection efficacy ($p = 0.295$
for testing the hypothesis that the log of the efficacies is not normally distributed using the D'Agostino's K2
test, n pairs = 1044), but not for connection contribution ($p < 0.001$, D'Agostino's K2 test).”

• LI. 321: the random sampling procedure is not clear. What are 1st data/shuffled versus 2nd
data/shuffled? Instead of reporting how many shuffles were significantly different it may be better to: (1)
determine median efficiency/contribution of strongest/2nd strongest connection in data, (2) determine
distribution of median efficiency/contribution of 1st/2nd connection from 1000 times shuffling, (3) determine
whether median of data falls into 2.5 to 97.5% percentile interval of shuffled distribution

We have followed the suggestion and determined whether the median of the data falls into the 2.5 to 97.5%
percentile interval of the shuffled data. We have modified the corresponding panels in Figure S7f/g, figure
legend and main text.

Line 327: “To test this prediction, we performed a permutation test by randomly sampling ($n = 1000$ repeats)
connection efficacy and connection contribution of divergent connections from the measured distributions
and analyzed those randomly generated divergent connections in the same way as the real data. This
permutation test showed that the median of the data fell within the 2.5% and 97.5% percentile interval of
the shuffled data for both efficacy (Fig. S7f) and contribution (Fig. S7g).”

• Two sentences starting in l. 365 (“Our results support...”) sound like the direct opposites of each
other. Please clarify what you mean (especially in the 2nd sentence).

We have modified these sentences to convey more clearly what we mean.

Line 374: “Our results support the notion that retinocollicular connections are organized in a specific manner
with functionally similar RGC-SC pairs being strongly connected, thus suggesting that a large fraction of
SC neurons receives limited convergent input from the retina. However, we also noticed cases with
relatively strong connections between RGC-SC pairs with low similarity, suggesting that some SC neurons
receive convergent input from a functionally more diverse pool of RGC afferents.”

- L. 382: use words instead of $r_{(SD)}$, ...

We now use words in the revised manuscript.

Line 391: "similarity of the responses to the dark/light sparse noise and moving bars"

- Fig 5a: specify which data come from RGC, which from SC neurons

We now provide this information in the figure legend.

- Fig 5c+d: CCGs need to show where 0 is on x-axis

The 0 ms mark was added.

- Fig 5c: RFs of RGCs largely overlap, which seems to contradict previous claims that RFs of the same RGC type are not overlapping.

We apologize but we forgot to account for overestimating the receptive field size by using the 10deg sparse noise in this example. In the revised version we corrected this mistake.

- L. 430: "we noticed..." unclear. Better: gap in RF positions along the probe (or similar)

Thank you for this suggestion. We now write in line 443: "we noticed that there was a gap in the receptive fields' positions along the probe in a zebra finch recording"

- L. 511: more suitable reference to add here would be Schroeder et al. (2020, Neuron), which shows functional imaging of RGC boutons in the SC rather than the LGN

Corrected.

- L. 852: explain the "elbow method"

We have added text to the method section to explain the elbow method in more detail.

Line 876: "The optimal number of principal components (PC) that capture sufficient variance in the dataset was estimated heuristically, using the elbow method¹¹⁶ illustrated by the scree plot representation (Fig. S3b). A scree plot represents the percentage of the variance contained in each PC, ordered by descending values (Fig. S3b). The "elbow" point in such a graph is identified as the PC number where the curve changes from a steep slope descent, to a linear, gradually descending slope – defining thus an optimal balance between the lowest number of components used and the cumulative variance explained between. In our case, beyond $n = 2$ components, the curve resorts to a linear slope descent, thus, the lowest number of components that could explain the maximum variance of the dataset was chosen as 2."

- LI. 855 still unclear. What window are you referring to? Do you mean "upsampled" to 10 times the given sampling rate using linear interpolation?

Yes, indeed the waveform was up sampled 10 times. We change the paragraph as written below.

- L. 858: what does “time-sliced” mean?

We apologized for the inexact use of language. We repositioned the waveform in time aligning to the trough. We have corrected the paragraph, see below.

- Features used to classify spikes into retinal versus SC: As these features will be crucial for other researchers to replicate the method, it would be very helpful to describe how each feature was determined in more detail (possibly in a table). For example, how were slopes determined? Is it: the slope between points of the waveform that cross: $A1-0.1*(A1-A2)$ and $A1-0.9*(A1-A2)$ for slope S1? How was half height of peaks determined? Height of W2 in SC waveform (Fig S3a) seems lower than half height of A3.

We now provide more information on how the features were determined including a new table.

- LI. 859: unclear how slopes were determined. “Percentile” refers distributions.

We apologized for the inexact use of language. Here the slopes were measured as described below which is now updated in the corresponding paragraph in the current manuscript.

- LI. 860: Does this mean that all 14 features were determined separately for each channel and then averaged across channels?

Yes, the features were determined separately for each channel. In the revised version we now provide a more detailed explanation how the features were determined, including a new table (Table 1 in the revised manuscript).

Line 889: “The interpolated and smoothed waveform was trough-aligned for more reliable characterizations keeping a pre-trough period of 0.6 ms and post-trough period of 3 ms. For further quantification the waveforms were re-normalized. 14 features were measured on each channel individually (Table 1), and averaged across the channels of the previously defined spatial spread (Fig. S3a). For example, all four slope measurements (S1-S4 in Fig. S3a) were computed between two concurrent peaks/troughs and were calculated from time points where the waveform crosses peak/trough1 (0.8 x peak/trough1) to peak/trough2 (0.2 x peak/trough2).”

- LI. 940: for a convincing argument, we would need to see the measures (RF for 5 degree versus 10 or 15 degree stimulus) for the whole population not just a single example. For large RF sizes, this method may underestimate the real size.

We agree that this method might underestimate the receptive field size of all RGC/SC neurons, in particular for neurons with large receptive fields. However, the aim of this analysis was to characterize the spatial positions of the receptive field centers and relate those to the spatial positions of the axonal field centers. This revealed that the receptive field centers are almost identical when mapped with different sparse noises. The receptive field size measurements were mainly used for illustration purposes in the figures and were not used for further analysis in the manuscript.

- Fig S1h: the opposite SC is not shown in green as stated in the caption

Corrected

• Fig S6e: What do responses to bar stimuli show? Concatenated response vector for all bar
directions? Or average across all directions?

In the revised version we provide more information on how the response vector for the moving bars was
calculated.

Line 948: “The evoked responses to the moving bars (light bar on dark background, 12 directions) were
calculated following the method described in1. Briefly, in the first step the times at which the bar entered
the receptive field (onset response) and the moment when the bar left the receptive field (offset response)
were calculated. The trial averaged PSTHs for each direction were then aligned and centered around the
onset-response, with a 0.1ms pre-stimuli, and 0.7ms post-stimuli time window. The final response array [12
(directions) x 2700 (time points in ms)] was decomposed using singular value decomposition, to obtain a
temporal component that represents an averaged response of all directions over time, and an orientation
component that represents its tuning preference. This temporal component obtained for each neuron, which
could uncover its polarity preference (ON/OFF/ON-OFF), and kinetics preference (sustained/transient) to
the bar, was concatenated with its corresponding responses to the chirp and the sparse noise stimuli.”

How were classes sorted? Why not show the same order of classes in d and e?

We now sorted the classes in the same way in Fig. S6d and Fig. S6e.

Missing/wrong references:

• L. 84: add references to papers showing RGC axon innervation in SC

We have added references to papers showing the RGC axon innervation patterns in SC.

• L. 85: wrong ref. to Fig 1d

Corrected to “Fig S1d”

• L. 91: Fig S2l does not exist

Corrected to “Fig S2i”

• L. 375: Fig 5e

Corrected to “Fig 5c”

• L. 377: Fig 5f

Corrected to “Fig 5d”

• L. 889: Fig S6

Corrected to “Fig S5”

• Ref 2 on l. 98 in suppl. Material

Reference 2 refers to Baden et al. 2016 which is correct in this context.

Spelling/grammar:

• L. 113: RCGs

Corrected

• LI. 202: grammar at end of sentence seems off

This sentence has been modified in the revised manuscript.

• L1. 315 should probably read: “only a few RGCs contributed strongly to the spiking of single
postsynaptic SC neurons” (or similar)

Corrected

• L. 355: similarly

Corrected

• L. 557: pf

Corrected

**References in this letter**

Kremkow, J., Jin, J., Wang, Y., and Alonso, J.M. (2016). Principles underlying sensory map topography in
primary visual cortex. *Nature* 533, 52–57. <https://doi.org/10.1038/nature17936>.

Wässle, H., Boycott, B.B., and Illing, R.B. (1981). Morphology and mosaic of on- and off-beta cells in the
cat retina and some functional considerations. *Proceedings of the Royal Society of London. Series B,*
*Containing Papers of a Biological Character.* Royal Society (Great Britain) 212, 177–195. .

Zhan, X.J., and Troy, J.B. (2000). Modeling cat retinal beta-cell arrays. *Visual Neurosci* 17, 23–39.
<https://doi.org/10.1017/s0952523800171032>.

REVIEWER COMMENTS

Reviewer #2 (Remarks to the Author):

Peer review of "High-density electrode recordings reveal strong and specific connections between single retinal ganglion cells and midbrain neurons"

We'd like to congratulate the authors to a great paper! All of our concerns have been addressed and we're looking forward to seeing this work published.